# Cryo-EM structures of *Trypanosoma brucei gambiense* ISG65 with human complement C3 and C3b and their roles in alternative pathway restriction

Hagen Sülzen [1,2], Jakub Began [1,7], Arun Dhillon[1], Sami Kereïche[1,3], Petr Pompach[4], Jitka Votrubova[1], Farnaz Zahedifard[5], Adriana Šubrtova [1], Marie Šafner[1], Martin Hubalek[1], Maaike Thompson [1,6,8], Martin Zoltner [5] & Sebastian Zoll[1] ✉

African Trypanosomes have developed elaborate mechanisms to escape the adaptive immune response, but little is known about complement evasion particularly at the early stage of infection. Here we show that ISG65 of the human-infective parasite *Trypanosoma brucei gambiense* is a receptor for human complement factor C3 and its activation fragments and that it takes over a role in selective inhibition of the alternative pathway C5 convertase and thus abrogation of the terminal pathway. No deposition of C4b, as part of the classical and lectin pathway convertases, was detected on trypanosomes. We present the cryo-electron microscopy (EM) structures of native C3 and C3b in complex with ISG65 which reveal a set of modes of complement interaction. Based on these findings, we propose a model for receptor-ligand interactions as they occur at the plasma membrane of blood-stage trypanosomes and may facilitate innate immune escape of the parasite.

The ability of pathogens to survive and proliferate in the host's hostile environment is inextricably linked to their capacity to evade and manipulate its immune responses. African trypanosomes are extreme examples of such competence. The blood-stage parasites are exposed to the full pressure of both branches of the immune system and must employ a plethora of countermeasures to avoid clearance[1]. Many of the molecular mechanisms underlying these elaborate immune evasion strategies have been identified and, particularly with regard to the adaptive immune response, described in detail: variant surface glyco-proteins (VSGs) constituting a protective surface layer, antigenic variation of exposed VSG epitopes, VSG O-glycosylation and flow-

mediated clearance of surface-bound antibodies[2–6]. In contrast, defence mechanisms against the innate immune response have so far mostly been described by the example of trypanolytic factors, toxin-bearing higher-order protein complexes that are unique to humans and some primates[7,8]. *T. b. gambiense*-specific glycoprotein TgsGP (together with haptoglobin-haemoglobin receptor HpHbR) in *T. b. gambiense*[9–12] and serum-resistance-associated protein SRA in *T. b. rhodesiense*[13,14] have evolved as adaptations to overcome the action of these factors.

In comparison, our understanding of other innate immune defence mechanisms in human-infective African trypanosomes, parti-cularly during the early stage of infection, is incomplete.

[1]Institute of Organic Chemistry and Biochemistry of the Czech Academy of Sciences, Flemingovo namesti 542/2, 16000 Prague, Czech Republic. [2]Faculty of Science, Charles University, Albertov 6, 12800 Prague 2, Czech Republic. [3]First Faculty of Medicine, Charles University, Albertov 4, 12800 Prague, Czech Republic. [4]Institute of Biotechnology of the Czech Academy of Sciences,  25250 Vestec, Czech Republic. [5]Department of Parasitology, Faculty of Science, Charles University Prague, Biocev, 25250 Vestec, Czech Republic. [6]University of Antwerp, Antwerp, Belgium. [7]Present address: Department of Immuno-biology, University of Lausanne, Chemin des Boveresses 155, 1066 Epalinges, Switzerland. [8]Present address: Agidens, Industrial Machinery Manufacturing, Zwijndrecht, Antwerp, Belgium. ✉e-mail: sebastian.zoll@uochb.cas.cz

The complement system is an essential part of the host's innate immune response against invading pathogens. It involves 3 distinct pathways, the classical (CP), the lectin (LP) and the alternative pathway (AP)[15] (Fig. 1). While CP and LP are triggered by recognition of specific structures on the pathogen surface, typically proteins and glycans, the AP can also be initiated spontaneously. Each pathway comprises a series of proteolytic events, resulting in the generation of convertases, multi-component enzyme complexes that serve as central hubs within the cascades. The first convertase of the CP and LP is formed at the junction of both pathways by the covalent binding of C4b to the plasma membrane and subsequent interaction with the serine protease C2a. The emerging CP/LP C3 convertase C4bC2a cleaves native C3 to C3b, which can then associate with C4bC2a to create the CP/LP C5 convertase C4bC2aC3b, the last enzyme of the pathway. Cleavage of C5 initiates the terminal pathway (TP), the point of convergence of CP/LP and AP, during which C5b sequentially binds C6, C7, C8 and C9. C9 polymerises and subsequently forms a lytic pore in the membrane of the invading pathogen, the membrane attack complex (MAC).

While it mainly functions as an amplification pathway of the CP and LP, shortly after an infection, the AP can also act as a first line of defence against invading pathogens, before an antibody response is mounted[16,17]. According to the prevailing hypothesis, it is in a constant state of low-level activation (tick-over), resulting in the formation of the fluid-phase C3 convertase $C3(H_2O)Bb$[17–19]. In the presence of foreign microbial patterns, this can lead to swift activation of the AP cascade. Analogue to the CP and LP, central events of the AP are the formation of C3 and C5 convertases with identical substrate specificities as their counterparts, but different structures and compositions. The AP C3 convertase C3bBb comprises C3b and Bb, a cleavage product of factor B, and the AP C5 convertase C3bBbC3b is most likely established following the binding of a second molecule of C3b. The role of the second C3b molecule has, however, not been fully clarified.

Recently, it was also suggested that surface-bound C3b only primes C5 for cleavage by C3bBb rather than being part of a trimolecular complex[20].

In humans, sleeping sickness or HAT (Human African trypanosomiasis) is caused by only two *Trypanosoma* species, *T. b. rhodesiense* and *T. b. gambiense* of which the latter is responsible for 95% of all cases. Earlier, it was shown that *T. b. gambiense* does not succumb to cell lysis in human serum despite specifically activating the AP, pointing to a dedicated mechanism of complement inactivation at this stage. No complement-binding receptor was, however, identified[21,22].

In this work, we report the identification and characterisation of invariant surface glycoprotein 65 (ISG65) from *T. b. gambiense* as a receptor for C3 and its activation products in human-infective bloodstream forms. We show that only convertases of the AP but not of the CP/LP can form on the surface of *T. b. gambiense* and that ISG65 is capable of specifically inhibiting the activity of the AP C5 convertase in vitro. This prevents initiation of the TP and, thereby, cell lysis. Moreover, we present the structures of *T. b. gambiense* ISG65 in complex with native C3 as well as C3b determined by cryo-EM, describe the differences in the structural basis of complement binding and propose a plausible model for receptor-ligand interactions as they occur in the context of the VSG layer and how they likely facilitate the innate immune escape of the parasite.

## Results

### ISG65 is a C3b receptor in bloodstream trypanosomes

Initially, the interaction between *T. b. gambiense* ISG65 and C3 was identified using pull-down assays with the extracellular domain of ISG65 as bait and human serum (Supplementary Fig. 1a). To probe the stability of the complex, an excess of ISG65 was incubated with C3b and subjected to size-exclusion chromatography. Two distinct peaks corresponding to the ISG65:C3b complex and free ISG65 were

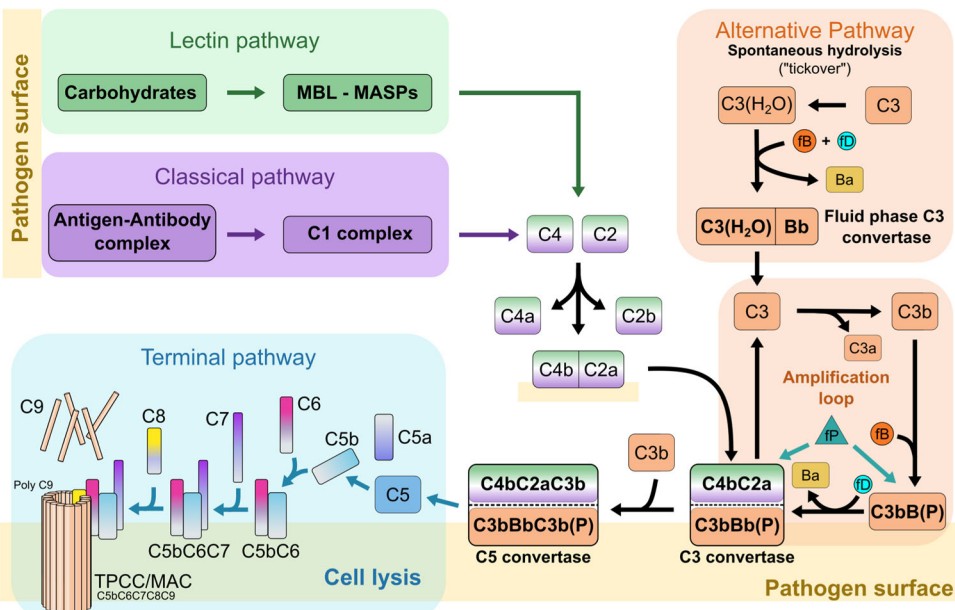

**Fig. 1 | Overview of the three complement pathways and their components.** The classical (CP) and the lectin pathway (LP) are triggered by specific antigens on pathogen surfaces. The alternative pathway (AP) can start spontaneously by a nucleophilic attack of water on metastable native C3, resulting in the generation of the fluid-phase C3 convertase, but it also serves as an amplification pathway after CP and LP. CP and LP share the same C3 and C5 convertases CP/LP C4bC2a and CP/LP C4bC2aC3b, respectively. Analogue to C3b of the AP C3 convertase, CP/LP C3 convertase formation is initiated by the covalent binding of thioester-containing C4b to the pathogen membrane. Both CP/LP convertases are structurally distinct from the AP C3 and C5 convertases AP C3bBb(P) and AP C3bBbC3b(P), but have the same substrate specificities. Properdin stabilises the AP convertases. Cleavage of C5 initiates the terminal pathway. C5b sequentially binds C6, C7, C8 and poly-C9 and inserts into the plasma membrane where it forms a lytic pore, the membrane attack complex. MBL Mannan-binding lectin, MASP MBL-associated serine proteases, fB factor B, fD factor D, Ba factor B fragment a, Bb factor B fragment b, fP factor P (properdin), MAC membrane attack complex, TPCC terminal pathway complete complex.

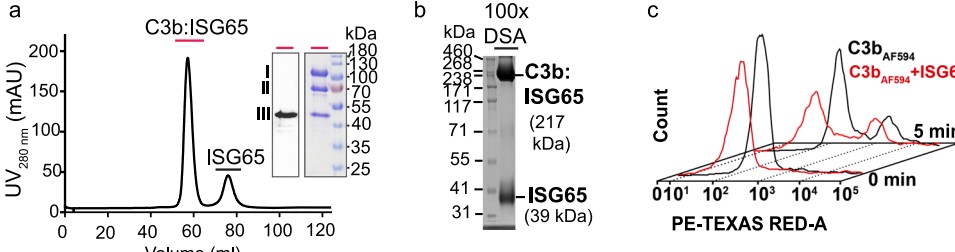

**Fig. 2 | ISG65 is a C3b receptor in *T. b. gambiense*. a** Chromatogram showing purification of the ISG65:C3b complex (217 kDa) using size-exclusion chromatography. Prior to injection, ISG65 was incubated with C3b in threefold molar excess. The complex formation of ISG65 with C3b was confirmed by analysing fractions underneath the main peak (indicated by a red bar) using reducing SDS-PAGE and Western blotting. Detection with an anti-His antibody confirmed the presence of His-tagged ISG65 in the complex peak; I, a-chain C3b, II, b-chain C3b, III, ISG65. Displayed Western blot and SDS-gel are representatives of three independent experiments. **b** SDS-PAGE analysis of cross-linked ISG65:C3b complex (100x excess of Di-(N-succinimidyl)adipate (DSA)). The displayed SDS-gel is representative of three independent experiments. **c** Representative scatter plots from FACS analysis of AF$_{594}$ labelled C3b surface binding and uptake by *T. b. gambiense* in the presence (red) and absence (black) of ISG65. Measurements were done in triplicates at two different timepoints. Source data are provided as a Source Data file.

observed. Analysis of peak fractions by SDS-PAGE and Western blotting revealed three bands that corresponded to ISG65 as well as to the α- and β-chains of C3b, demonstrating that a stable complex was formed (Fig. 2a). To further characterise the interaction, the complex was cross-linked. SDS-PAGE analysis showed two bands migrating at approximately 217 and 39 kDa, consistent with the expected sizes of cross-linked complex (at 1:1 stoichiometric ratio) and free ISG65 (Fig. 2b).

No complete ISG65 knock-out cell line was available during the time of this study. To investigate if ISG65 functions as a C3 receptor in *T. b. gambiense*, we compared surface binding and uptake of free C3b to ISG65-saturated C3b using fluorescence-activated cell sorting (FACS). FACS revealed that Alexa Fluor 594 (AF$_{594}$) labelled human C3b is efficiently internalised via endocytosis in *T. b. gambiense* cells and delivered to the lysosome, similar to AF$_{594}$ labelled human transferrin (Supplementary Fig. 2). The addition of recombinant ISG65 at a 1.5-fold molar excess led to 60% reduction of C3b surface binding and a 40% reduction of AF$_{594}$ positive cells after 5 min of uptake, as compared to treatment with C3b alone (Fig. 2c, Supplementary Fig. 3, and Supplementary Data 1).

Incubation of labelled C3b with another member of the *T. b. gambiense* invariant surface glycoprotein family, ISG75, had no significant effect on surface binding and uptake (Supplementary Fig. 3a). These findings confirmed the suspected role of ISG65 as a C3 receptor. Residual uptake of ISG65-saturated C3b can either be attributed to fluid-phase endocytosis, partial disintegration of the complex or the presence of another C3 receptor.

## ISG65 restricts the alternative but not the classical and the lectin pathway

Previous results indicated that *T. b. gambiense* activates the alternative but not the classical pathway, and that a specific molecular mechanism is likely to exist that blocks its progression and prevents cell lysis[21]. Using pathway-specific, fluorescently-labelled antibodies and depleted human sera, we could show that only Bb, a component of AP convertases, but not C4b, an essential component of both CP and LP convertases, is accumulated in the flagellar pocket (Fig. 3a, b and Supplementary Fig. 4a, c) as a consequence of deposition on the surface of *T. b. gambiense*. In contrast, we could readily detect C4b on the surface of sensitised sheep erythrocytes, which were incubated in C6-depleted human serum to avoid premature cell lysis (Fig. 3b). Using C2-depleted serum, we could also confirm that Bb is deposited as a consequence of AP activation and not via the amplification loop through a minor deposition of the CP/LP C3 convertase which contains C2. In agreement with this observation, C3b could only be detected on cells that were incubated in C2-depleted, but not fB-depleted serum (Fig. 3c, d and Supplementary Fig. 4b, d). Since trypanosomes are not affected

by complement-mediated lysis, we investigated whether the TP is initiated at all by testing for binding of C5b, the first complement factor of the TP, using a fluorescently-labelled anti-C5b antibody.

No flagellar pocket signal could be detected on trypanosomes while sheep erythrocytes, when incubated in C7-depleted human serum (allowing membrane-binding of C5b via C6[23], but preventing premature lysis) bound C5b efficiently and presented a robust fluorescent signal (Fig. 3e). This strongly suggests that the TP is not initiated on *T. b. gambiense* and that restriction of the complement cascade occurs upstream.

To ascertain that ISG65 binding to C3 is responsible for the observed restriction of the alternative pathway, we carried out functional haemolytic assays using ISG65 as an inhibitor. The addition of soluble ISG65 to human serum resulted in an almost complete abrogation of rabbit erythrocyte lysis (up to 90% decrease) following activation of the AP (AP50 test, Haemoscan) (Fig. 4a). In agreement with the prior observation that no components of CP/LP convertases could be detected on the parasite surface, we also could not observe any ISG65-mediated decrease in lysis of sensitised sheep erythrocytes after induction of the CP in vitro (CH50 test, Haemoscan) (Fig. 4b). In both assays ISG75 had no measurable impact on erythrocyte lysis, confirming the specificity of the observed effect for ISG65.

Modelling suggests that ISG65 does not interfere with the assembly or activity of the AP C3 convertase (Supplementary Fig. 5a). Additionally, AP C3 pro-convertase formation in the presence of ISG65 was shown by SPR (Supplementary Fig. 5b). Therefore, inhibition via binding to C3b must take place at the stage of the AP C5 convertase, either by preventing its assembly or by diminishing the ability of the assembled convertase to cleave C5 substrate. To investigate how exactly ISG65 interferes with AP C5 convertase activity in vitro, we used a modified AP50 assay (Haemoscan) to separate convertase deposition on the cell surface from the assembly of the membrane attack complex (MAC) in the TP, leading to haemolysis. Erythrocytes were hereby incubated with dilutions of human complement C5-depleted serum, which allowed for C5 convertase assembly on the cell surface, but stalled the progression of the cascade due to a lack of C5 substrate. After a washing step, ISG65 and ISG75 (negative control) were added together with dilutions of human complement factor C3-depleted serum, providing the C5 substrate without introducing competing C3.

The modified AP50 assay was validated by efficient lysis of erythrocytes when both depleted sera were used consecutively (reconstituted NHS). Exchange of either serum with buffer prevented lysis. The addition of ISG65 to the assembled AP C5 convertase revealed a significant effect on TP progression. In comparison to reconstituted NHS, haemolysis was decreased by up to 40% in the presence of ISG65 (Fig. 4c). This demonstrated that binding of ISG65 to pre-assembled AP

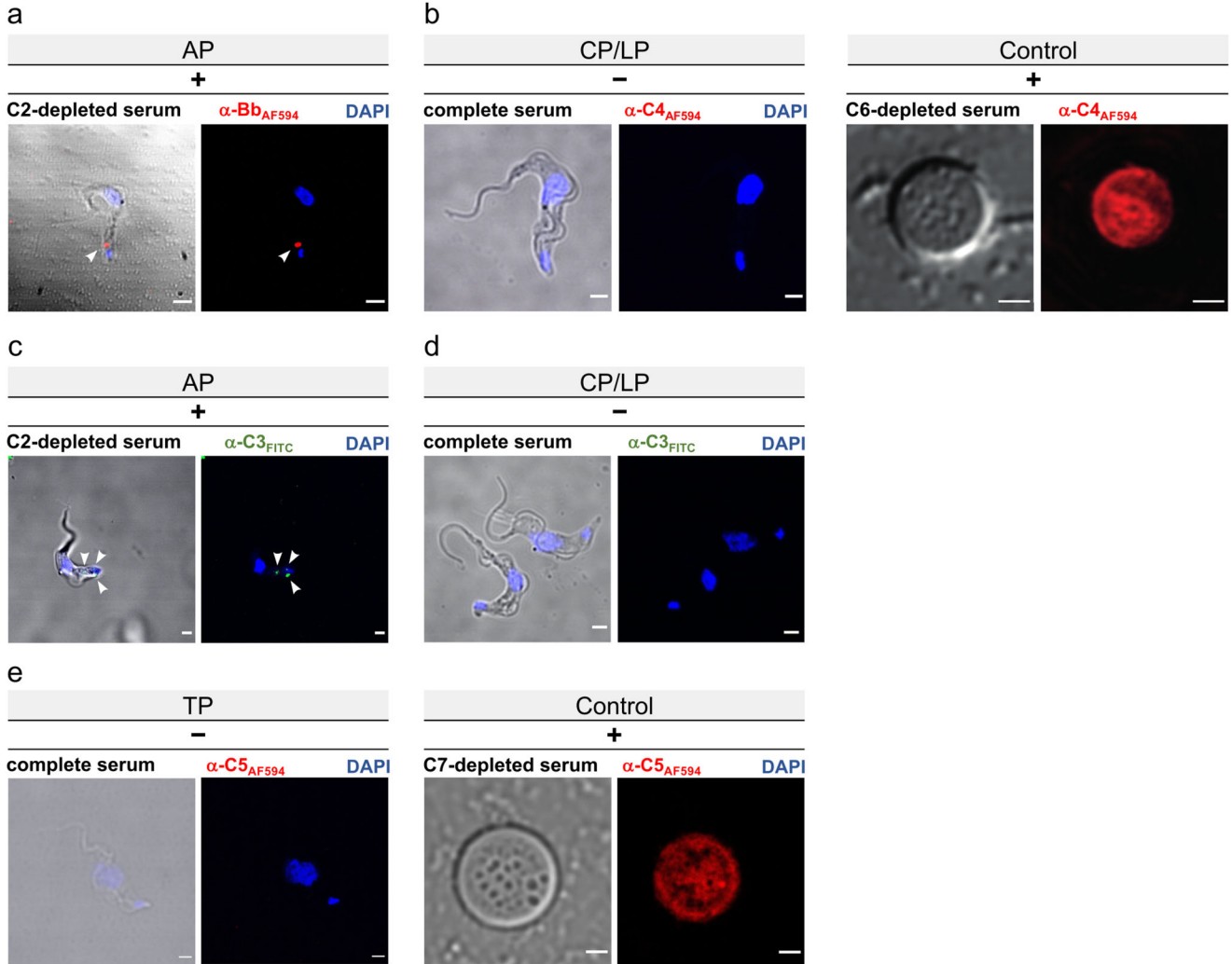

**Fig. 3 | Bb and C3b are deposited on the surface of *T. brucei gambiense* in a C2-independent manner, while C4b and C5b are not detected by immunofluorescence.** Representative fluorescence micrographs of trypanosomes incubated in depleted or complete human serum tested for deposition of complement factors using fluorescently-labelled antibodies as indicated (red = AF594, green = FITC; blue = DAPI stain; left images are merged with brightfield channel; scale bar = 2 µm). Independent experiments were performed at least in triplicate. Note, that even at low temperatures, molecules bound to the trypanosome surface are rapidly internalised, accumulating in the flagellar pocket before delivery to the early endosome. Flagellar pocket, early endosome and lysosome are highlighted by white arrows on selected images. **a** *T. b. gambiense* in C2-depleted serum (no CP/LP) stained with anti-Bb. Bb was robustly detected at the flagellar pocket. **b** *T. b. gambiense* (left) in complete serum and sensitised sheep erythrocytes (right, positive control) in C6-depleted serum (no MAC assembly) stained with anti-C4. No C4b-specific signal was detected on trypanosomes. **c** *T. b. gambiense* in C2-depleted serum (no CP/LP) stained with anti-C3. C3b was robustly detected at the flagellar pocket. **d** *T. b. gambiense* in fB-depleted serum (no AP) stained with anti-C3. No C3b-specific signal was detected. **e** *T. b. gambiense* (left) in complete serum and sensitised sheep erythrocytes (right, positive control) in C7-depleted serum (no MAC assembly, but C5b-6 membrane deposition) stained with anti-C5. No C5b-specific signal was detected on trypanosomes. For additional experiments, replicates and controls, see Supplementary Fig. 4.

C5 convertase can at least partially abrogate cleavage of C5 substrate in vitro.

### ISG65 binds C3 fragments with varying affinities

During the activation of the AP, C3 undergoes multiple proteolytic cleavages that result in an array of fragments, each with a specific role within the AP (Fig. 1). To understand whether ISG65 can distinguish between these naturally occurring fragments, we determined dissociation constants ($K_D$) using surface plasmon resonance (SPR) (Fig. 5). The identity of all C3 species was confirmed by intact mass spectrometry (MS), N-terminal sequencing, Western blotting and ion-exchange chromatography (Supplementary Fig. 6). ISG65 has clear preferences for certain C3 fragments. Native C3 ($K_D = 130$ nM), the most abundant C3 species in human blood, has the lowest binding affinity, followed by C3b ($K_D = 81$ nM) and C3MA, a mimic for C3($H_2O$)[24] ($K_D = 18$ nM). The interaction with C3d (the proteolytically liberated

thioester domain of C3, TED) takes place at an even higher affinity ($K_D = 7$ nM), likely due to unimpeded movement in the absence of the remaining C3 scaffold (i.e. C3c).

Dissociation constants for C3 and C3b are in line with the high serum concentrations of both fragments (C3: 5–11 µM, C3b: 152–888 nM)[25]. With reported concentrations ranging from 3–100 nM[18,26,27] for C3($H_2O$), the measured affinity of ISG65 to C3MA also appears to be in a physiologically relevant range. Notably, the amount of C3b depends on the complement activation status. Together with a short half-life of C3b, actual concentrations may be lower than reported values. No binding of C3c to ISG65 could be observed, indicating that TED is the main interacting domain of C3. Lack of steric interference from the C3 scaffold could also be the reason for the several-fold higher affinity to C3MA compared to C3b and native C3. In C3MA, the TED has been suggested to be more accessible or more flexibly tethered to the C3 scaffold than in the other fragments[28,29].

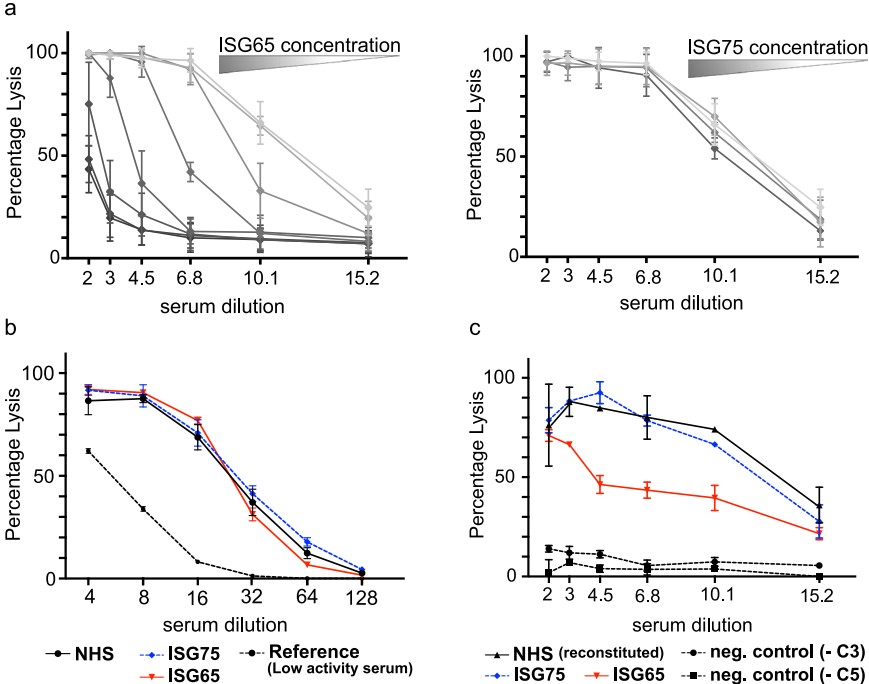

**Fig. 4 | ISG65 prevents haemolysis following activation of the alternative pathway, but not the classical pathway. a** Haemolysis assay (AP50 test) measuring the lysis of rabbit erythrocytes in human serum in the presence of ISG65 (left) and ISG75 (right) following activation of the alternative complement pathway ($n = 3$ technical replicates; data points are presented as mean values ± SD). ISG65, but not ISG75, considerably reduced haemolysis in a concentration-dependent manner. Colour gradients indicate protein dilutions ranging from 7.5 to 0 μM. **b** Haemolysis assay (CH50 test) measuring the lysis of sensitised sheep erythrocytes in human serum in the presence of ISG65 (red solid line), ISG75 (dashed blue line) and without added inhibitors (NHS, normal human serum; solid black line) following activation of the classical complement pathway ($n = 4$ technical replicates; data points are presented as mean values ± SD). At 24 μM, neither ISG65 nor

ISG75 showed notable inhibition of erythrocyte lysis. Low complement activity reference ($n = 3$ technical replicates; data points are presented as mean values ± SD) is depicted with a dashed black line. **c** Adapted AP50 haemolysis assay using C5- and C3-depleted human sera. Binding of ISG65 to assembled AP C5 convertase C3bBbC3b reduced haemolysis (red solid line). No reduction in haemolysis could be detected for ISG75 (dashed blue line). Human sera depleted for either C5 (-C5) or C3 (-C3) did not cause haemolysis (dashed black lines). Incubation of erythrocytes with C5-depleted serum followed by incubation with C3-depleted serum, reconstituted haemolytic activity (NHS (reconstituted), solid back line). ($n = 3$ technical replicates; data points are presented as mean values ± SD). Source data are provided as a Source Data file.

## Cryo-EM structures of ISG65 in complex with C3b and native C3

To gain structural insights into the interaction, we employed single-particle cryo-EM and determined the structures of ISG65:C3 and ISG65:C3b. The ISG65:C3 complex could be reconstructed to a reported $FSC_{0.143}$ of 3.58 Å, while ISG65:C3b was reconstructed to $FSC_{0.143}$ of 3.59 Å (Supplementary Table 1). However, the local resolution range across the reconstructions varies significantly, especially for ISG65:C3b. The resolution of the reconstruction corresponding to the rather rigid MG-ring is in the near-atomic range of 3.5 Å. The local resolution in the CUB and TED region varies between 4 and 5.5 Å, while at the interface and in the remainder of ISG65, it ranges from 5 to beyond 10 Å (Supplementary Fig. 7c). Secondary structure features like β-sheets in the CUB domain and α-helices in ISG65, however, could still be identified (Fig. 6a). For ISG65:C3, the local resolution predominantly ranges from 3 to 5 Å, with lower resolutions in the disordered head region of ISG65, the flexible C345c domain, and glycans in the two glycosylation sites (Supplementary Fig. 8c). The higher variance in local resolution across the ISG65:C3b reconstruction is likely caused by increased flexibility of the extended domains CUB and TED relative to the MG-ring. In contrast, C3 adopts a more compact conformation with TED wedged between MG2, MG8 and CUB. This results in reduced flexibility and thus variance within the molecule, which is reflected by the sloped shape of the FSC of ISG65:C3b compared to a rather vertical drop of the FSC in the ISG65:C3 reconstruction (Supplementary Figs. 7, 8).

C3b could be modelled except for the flexible regions Asn93-Gly101, Gln312-Leu314, Pro665-Ala667, Ser749-Leu751, Glu1372-

Ala1380, and Ser1523-Asp1524 (Fig. 7). C3b resembles the previously published structure of C3b[30] (PDB 2I07), to which ISG65-bound C3b can be aligned with an RMSD = 2.0 (11,068 atoms). Minor rearrangements of the TED and the CUB domain, especially in residues Gln1161-Pro1287, can be observed in C3b and may be caused by ISG65 binding. The lack of near-atomic resolution in ISG65 of the ISG65:C3b reconstruction prevented us from drawing conclusions about the details of the binding interface. However, the superposition of the atomic model of ISG65 with its counterpart in the C3 complex suggests an identical interface with TED (Supplementary Fig. 9). In addition, the electron density corresponding to the head domain of ISG65 also suggests contacts with the CUB domain of C3b (Fig. 7). Should this interaction occur in a physiological context, the second attachment point could restrict the free movement of the TED.

ISG65 adopts a three-helix bundle fold (Helix 1, Lys36-Lys84; Helix 2, Asp100-Asn144; Helix 4, Thr259-Glu300) with a 20° curvature along its longest axis (Figs. 6, 7). The disordered N-terminal residues Leu18-Leu34, as well as residues Arg146-Ser202 and Lys227-Met255, constitute intrinsically disordered loops and thus were not resolved. Instead of distinct conformations, a cloud of electron density was observed, describing the average volume occupied by the loops (Supplementary Fig. 10d). Using MS-based disulphide mapping, we identified three disulphide bridges in the disordered head domain of ISG65 (Cys43-Cys200, Cys168-Cys181 and Cys240-Cys251) (Fig. 8a and Supplementary Fig. 10). The atomic model of the disordered regions was predicted using template-guided modelling in AlphaFold2 using real-space constraints from the reconstruction and disulphide

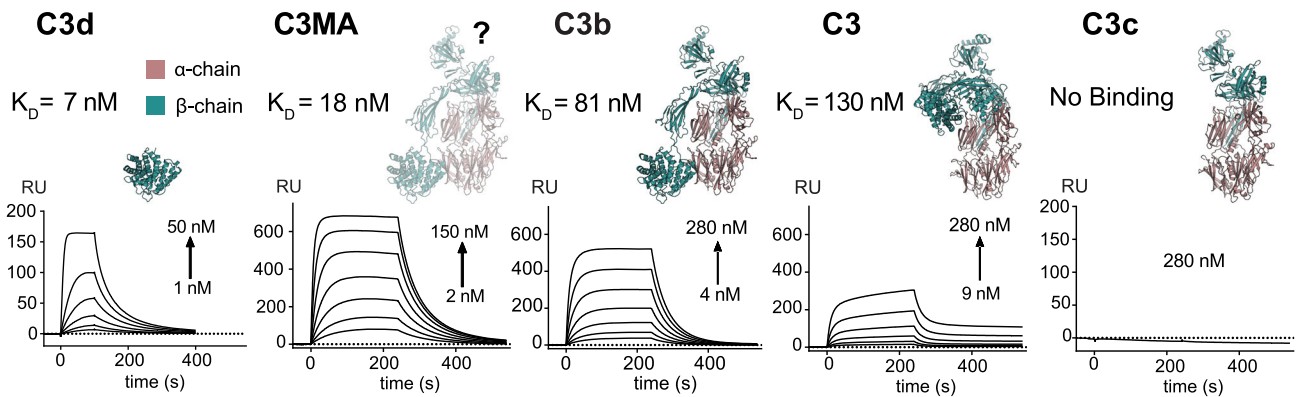

**Fig. 5 | ISG65 binds proteolytic fragments of C3 with different affinities.** Surface plasmon resonance sensorgrams showing the kinetics of ISG65 binding to C3d, C3MA, C3b, native C3, and C3c (from left to right). Increasing concentrations of C3 analytes were injected into the sensor chip, to which C-terminally biotinylated ISG65 (ligand) was immobilised. Structures of C3 fragments are shown above the corresponding sensorgrams. The atomic structure of C3MA (a mimic of C3(H$_2$O) is unknown, but its overall architecture has been suggested to be similar to C3b[28], which is shown here instead, indicated with a question mark. Source data are provided as a Source Data file.

bonding as selection criteria for the models (Supplementary Fig. 10). Although resolved at a lower local resolution, the loop connecting helix 1 and helix 2 (Lys85-Asp100) close to the C-terminus could be modelled. Due to a well-resolved electron density, we could confidently place the short helix 3 (residues Gln217 to Val226) and the preceding loop (Asp210-Leu216) near the head domain. Residues Thr203-Asp210 were not resolved to atomic resolution but appeared to have a 3$_{10}$ helical conformation (Figs. 7, 8). Near the C-terminus of the model, Gln301-Leu305 form a loop connecting helix 4 with the short helix 5 (Thr306-Ala312), which points towards the N-terminus of ISG65, away from the parasite surface (Fig. 8b). Beyond these residues, only poorly resolved electron density was observed which precluded further model building. With the exception of the disordered regions Ala666-Ser672, Arg740-Leu751 and Glu1634-Asn1642, C3 was well-resolved and could be modelled in its entirety (Fig. 7). It has high overall similarity (RMSD = 2.0 Å, 11,878 atoms) with the previously published structure of C3[31] (PDB 2A73) (Supplementary Fig. 9c). Positioned above the loop (Gly684-Lys692) connecting the two N-terminal helices of the ANA domain and the N-terminal side of the longest helix in the domain (Ala719-His738) (Fig. 7), binding of ISG65 to C3 is predominantly mediated via hydrophobic contacts and four hydrogen bonds with the TED. Three hydrogen bonds are mediated by Tyr293 (Tyr293$_{ISG65}$-Glu1110$_{C3}$, Tyr293$_{ISG65}$-Glu1111$_{C3}$ and Tyr293$_{ISG65}$-Glu1119$_{C3}$) and one by Arg77 (Arg77$_{ISG65}$-Leu1109$_{C3}$) (Supplementary Table 2). The interaction between the proteins is further facilitated by the almost perfect complementary fit between the convex tip of TED and the concave 3-helix bundle of ISG65.

**Validation of the ISG65:TED interface**
Based on the higher-resolution structure of ISG65:C3, contacting interface residues (Supplementary Fig. 11a, b), were individually mutated to alanines. The binding affinities of the recombinantly expressed C3d and ISG65 mutants were determined by SPR and compared to the wild-type proteins (Fig. 7). Correct protein folding was assessed via circular dichroism spectroscopy (Supplementary Fig. 12). ISG65 residues across the entire interface contributed to the binding as indicated by the decrease in dissociation constants (Fig. 7 and Supplementary Table 3). Mutation of Tyr293, which is responsible for almost half of all contacts, completely abrogated ISG65 binding to C3d. Notably, mutation of Trp211 caused a decrease in K$_D$ to 42%. This is in agreement with the well-resolved electron density of the adjacent loop (Asp210-Leu216) as well as the side chain of Trp211. Although only contributing two hydrophobic contacts, this interaction might be crucial for the correct alignment of C3d at the upper end of the interface. Similarly, mutation of two of the four hydrogen bond-

forming residues in C3d, Leu1109, and Glu1110 also leads to a decrease in binding affinity. This can also be attributed to the fact that the side chains of Leu1109 and Glu1110 are located at the core of the interface and engage in multiple hydrophobic contacts (Fig. 7). Mutation of Arg77 of ISG65 was not included in the analysis due to misfolding of the mutated protein. Further validation of the main interface was carried out by hydrogen-deuterium exchange mass spectrometry (HDX-MS) and small-angle X-ray scattering (SAXS) using C3d instead of C3 (Supplementary Fig. 11c, d and Supplementary Table 4). Both methods unequivocally confirmed our identification of the interface as described here. All interface residues are located in areas of high protection from deuterium uptake and the model of ISG65:C3d is in good agreement with solution scattering data of the same complex. Surprisingly, despite its localisation outside the interface, Arg1134 of C3d was found in a 'protection hotspot', indicating limited solvent accessibility (Supplementary Fig. 11c and Supplementary Data 2). In line with this finding, the mutation did indeed cause a decrease in K$_D$ to 30%. Considering that Arg1134 is located within a long (26 aa) loop close to the interface, it seems conceivable that a transient interaction with ISG65 during binding may occur.

**ISG65 interacts with ANA/C3a**
A second, smaller interface is formed between loops in the membrane-proximal region of ISG65 and the ANA domain in C3 (Fig. 7). Within the interface, we could identify two possible hydrogen bonds, Lys97$_{ISG65}$ – Glu689$_{C3}$ and Glu302$_{ISG65}$ – Gly717$_{C3}$ (Fig. 7). Although the reconstruction in this flexible region suffers from poor side chains density, the identified residues are well within bonding distance (Supplementary Table 2). To investigate whether the interface between ANA and ISG65 may be physiologically relevant, we tested the binding of ISG65 to recombinant C3a (the proteolytically liberated ANA domain), which revealed a K$_D$ of 18 µM (Fig. 7 and Supplementary Table 3). In agreement with the lack of clear electron density, kinetic profiles showed high on- and off-rates, indicative of a rather short-lived interaction. Specificity of the interaction was further confirmed by alanine mutations in ISG65, each resulting in reduced affinity with dissociation constants of 80 µM for Lys97Ala and 78 µM for Glu302Ala, respectively (Fig. 7, Supplementary Fig. 11b, and Supplementary Table 3).

**Mechanisms of C3b binding on the parasite surface**
Unlike other *Trypanosoma* receptors described so far, ISG65 is a type-I transmembrane protein that is spread over the entire cell surface where it is embedded into the VSG layer[32]. The C3-binding domain of ISG65 is hereby tethered to its membrane anchor via a 46 amino acid long, disordered linker. In an effort to understand how this linker could

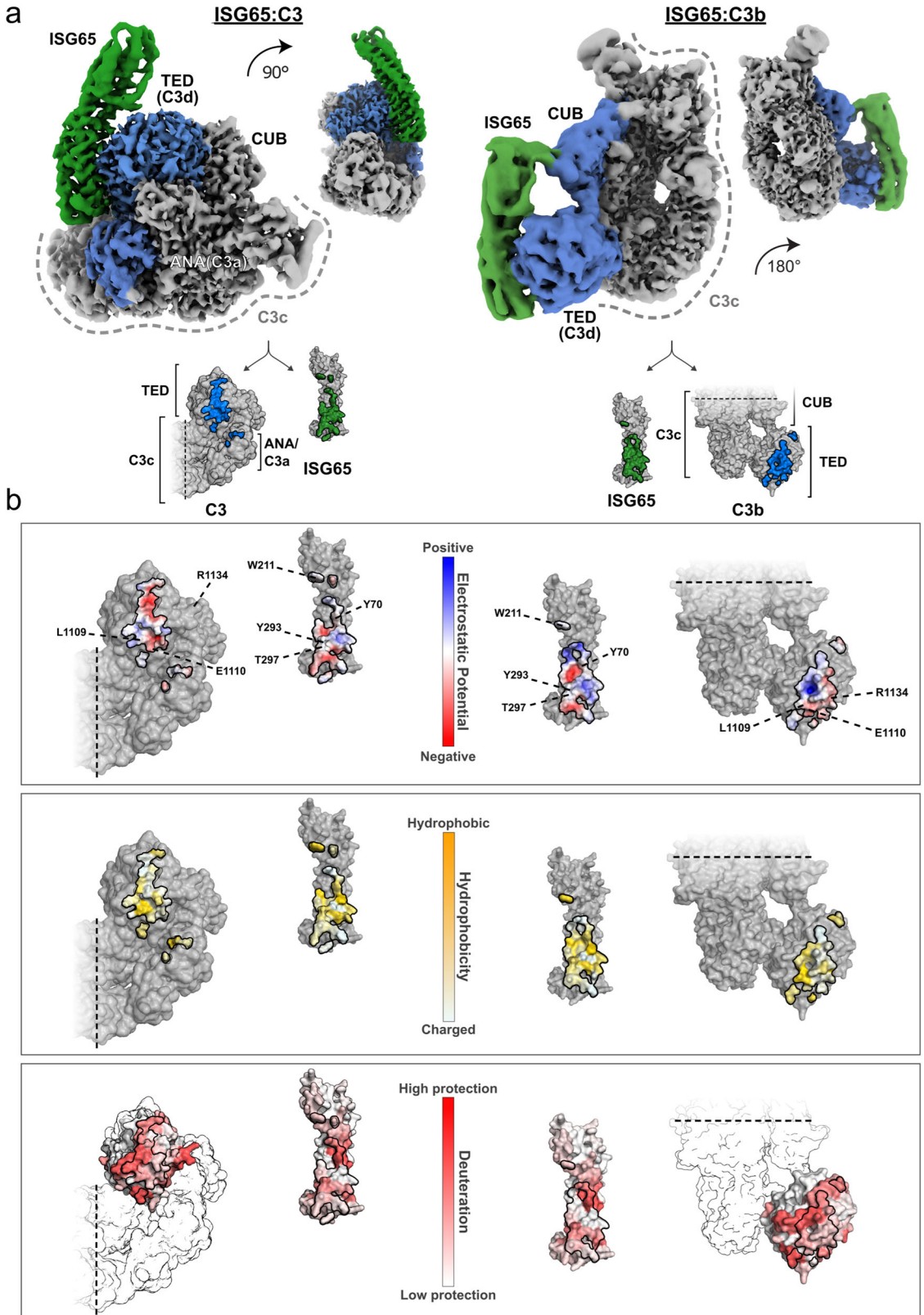

**Fig. 6 | Cryo-EM structures of ISG65 in complex with native C3 and C3b. a** Cryo-EM density maps showing side views of ISG65:C3 (left), and ISG65:C3b (right) at two different angles. ISG65 is represented by map regions coloured in green. Interacting domains in C3 and C3b are depicted in blue. In both C3 conformations, TED provides the primary interface. ISG65 and TED show a high degree of shape complementarity. Smaller, secondary interfaces are located in ANA (native C3) and CUB (C3b). The remaining scaffold (C3c, grey) shows no additional contact points

**b** Open book representations of the interaction interface coloured according to electrostatic potential, hydrophobicity and protection from deuterium uptake as determined by HDX-MS (from top to bottom). HDX-MS measurements were performed with ISG65 and C3d. Positions of contacting residues that contribute to binding as determined by alanine mutagenesis are indicated. Interfaces are delimited by a black line.

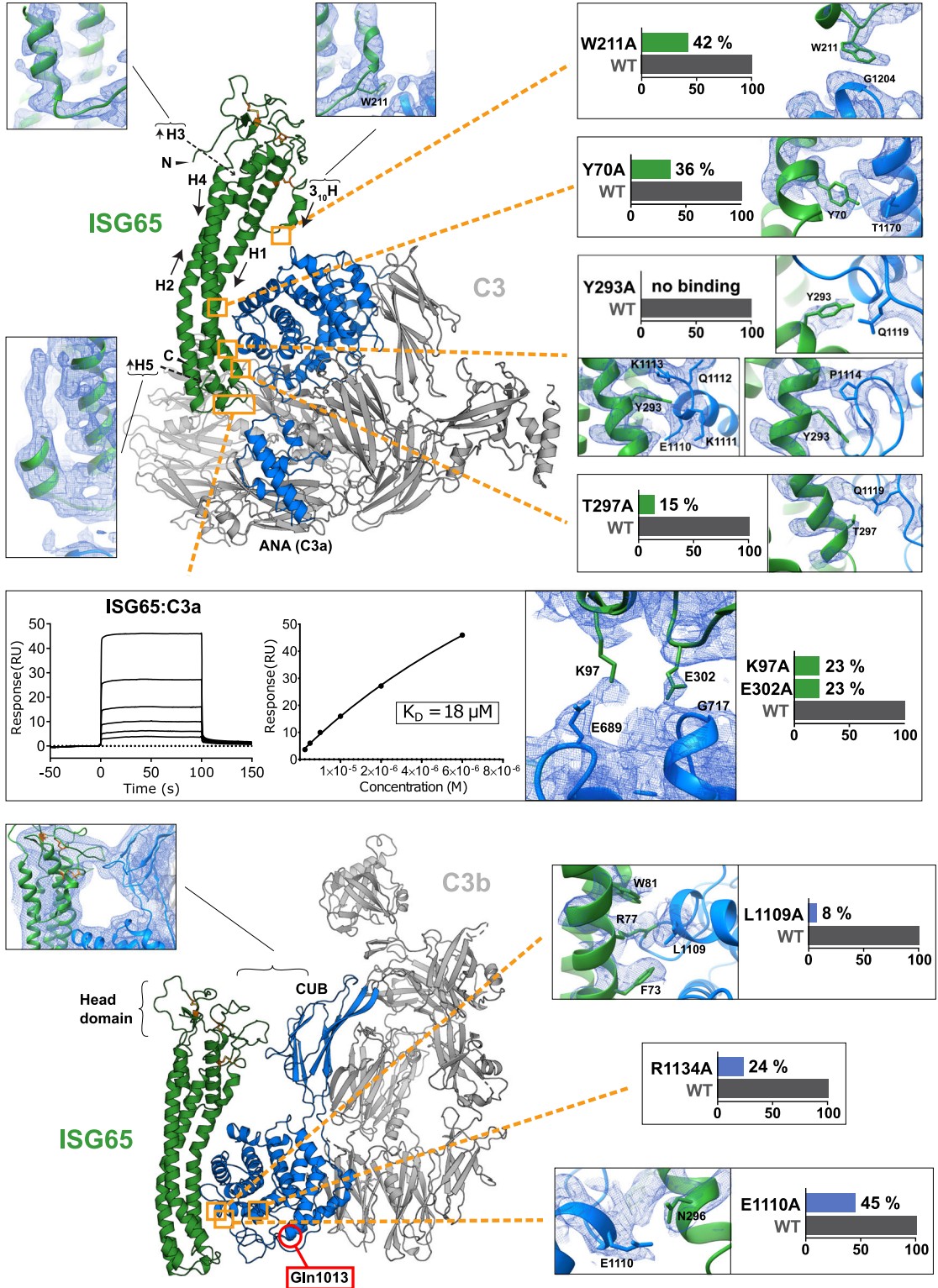

**Fig. 7 | Close-up views of different regions in ISG65 and C3/C3b.** Models of ISG65:C3 and ISG65:C3b are shown in cartoon representation, the directionality of helices in ISG65 is indicated by arrows. Gln1013 of the broken thioester bond is marked by a red circle. Orange rectangles indicate the positions of ISG65 and TED interface residues that were mutated. Presented bar diagrams show the relative effects of alanine mutations in ISG65 and TED on the overall binding affinity. Close-ups of relevant areas and interacting residues in the ISG65:C3 complex are shown in the unsharpened density map. Arg1134 of TED is not an interacting residue in the presented structures but was found to be protected from deuterium uptake in HDX-MS. Sensorgrams of the ISG65:C3a interaction are shown next to the respective close-up. Source data are provided as a Source Data file. The head domain of ISG65 was modelled using AlphaFold2 guided by experimental constraints (PDBDev: PDBDEV_00000201).

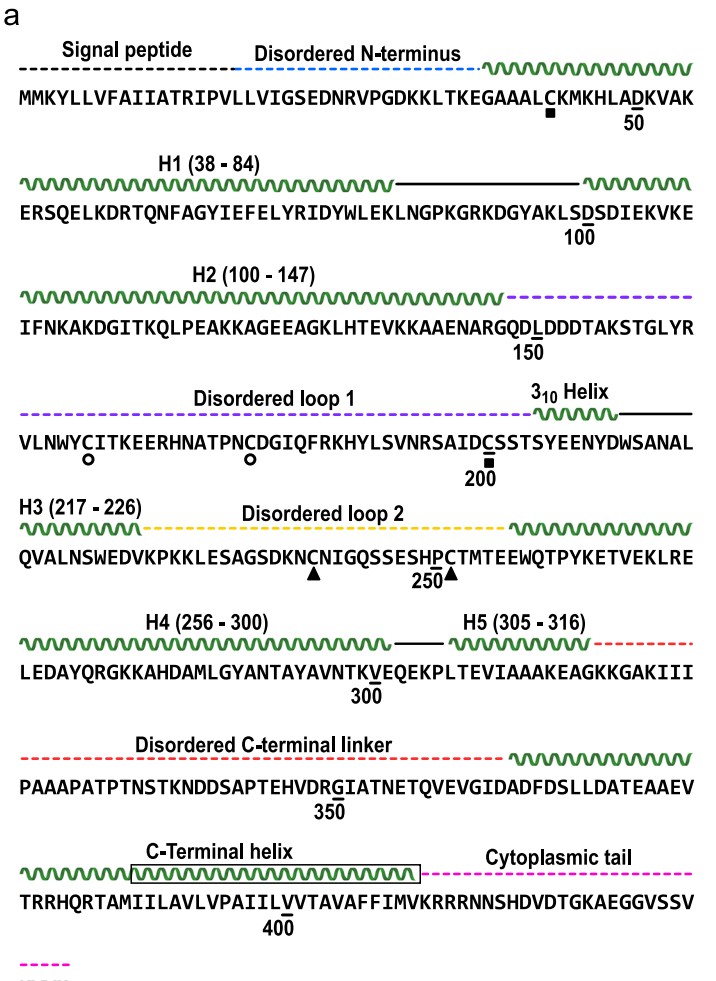

**Fig. 8 | Schematic illustration of ISG65 primary and secondary structure.**
**a** Amino acid sequence of full-length ISG65. Secondary structure elements are indicated above the sequence. Helices are displayed as wavy lines; loops present in the experimental model are displayed with solid black lines and loops absent from the experimental model are displayed as dashed lines. The colour scheme corresponds to panel **b**. Disulphide-bonded cysteine residues are indicated by squares, circles and triangles, paired residues are marked with the same geometric shape. The transmembrane region of the C-terminal helix is highlighted by a black box around the stylised secondary structure. **b** Overview of the connectivity and arrangement of ISG65 secondary structure elements. Disordered regions are indicated as dashed lines. These regions did not give rise to traceable electron density and were amended to the final model based on an experimentally constrained AlphaFold2 model. The colouring scheme is the same as in panel **a**.

possibly affect the orientation of ISG65 on the cell surface, we determined its likely conformations in solution using SAXS and ensemble optimisation (EOM)[33]. EOM allows for the calculation of several co-existing conformational states of a molecule in solution, taking inter-domain flexibility into account. For ISG65, two distinct conformational pools could be identified (Supplementary Fig. 13). These constitute the ensemble with the most likely conformations of the disordered C-terminal linker that connects helix 5 with the partially membrane-embedded helix 6. While most of the models in the ensemble were compact (~100–115 Å), a small subset adopted an extended conformation of ~170 Å (Supplementary Fig. 13e).

Using AlphaFold-assisted molecular modelling[34], we subsequently predicted a model of C-terminal helix 6 of ISG65 and the cytoplasmic domain that we appended to the models of the two most-populated conformational states of ISG65 (Supplementary Fig. 13d). The extracellular part of helix 6 spans ~30 Å in length. On the trypanosome surface, the compact conformation of ISG65 would therefore extend approx. 130–145 Å from the plasma membrane of the parasite, while in the extended conformation, it could span more than 200 Å, thereby protruding beyond the boundaries of the VSG layer. Based on these findings, we suggest a plausible model for the interaction of ISG65 with

human complement C3b on the cell surface. It has previously been reported that, in *T. b. brucei*, the VSG coat extends about 140–155 Å from the cell membrane[35]. We propose that in its compact conformation, the 'down' position, ISG65 sits flush with the VSG coat, exposing only the disordered head domain to the outside, while other epitopes are hidden underneath the protective VSG umbrella (Fig. 9). Upon extension into its 'up' position, ISG65 protrudes the VSG coat by about 60 Å, enabling it to interact with C3b which is either near or bound to the VSG layer (Fig. 9). While ISG65-binding likely serves to keep C3b away from the plasma membrane, it cannot be ruled out that some fraction of complement escapes the interception, penetrates the VSG layer and covalently attaches to the plasma membrane where it might form convertases. Although the exact conformation of the covalently attached complement remains unknown[36] it is conceivable that ISG65 in its compact conformation might be able to interact due to the same relative position within the VSG layer (Fig. 9).

## Discussion
Here we have presented the identification and structure of an invariant, immunogenic[37,38] yet abundant surface protein[38] of the human parasite *T. b. gambiense* that escapes neutralising antibody detection

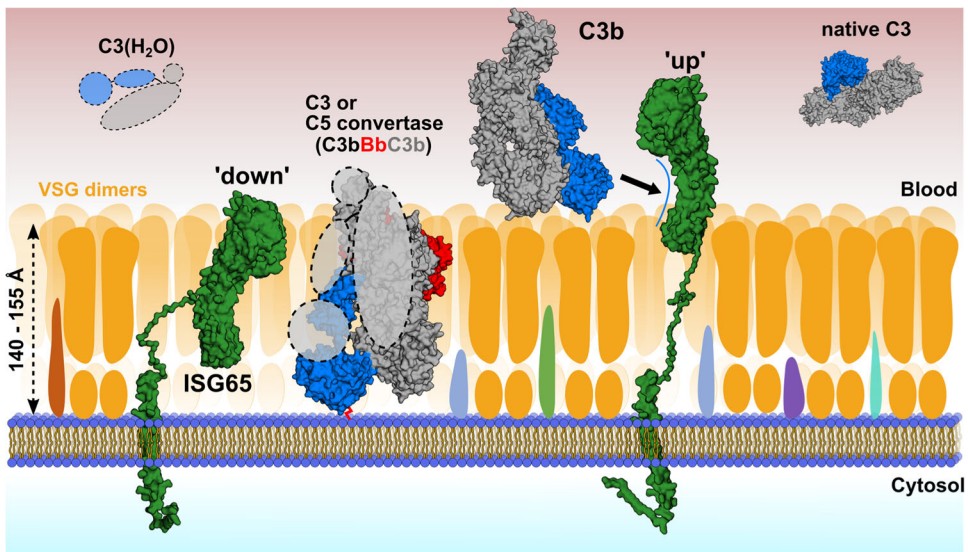

**Fig. 9 | Model of complement binding on the surface of *T. b. gambiense*.** In its compact 'down' position, ISG65 would remain embedded within the VSG coat, with its C3-binding epitopes being concealed and only the disordered head domain exposed to the outside. In this state, ISG65 may bind and inactivate AP convertases via C3b that may pass through the VSG coat, where it could covalently bind to the plasma membrane via Gln1013 of the former thioester (red connecting line). In its extended 'up' position, ISG65 would be to able intercept C3b outside the VSG coat and transport it to the flagellar pocket for uptake and subsequent lysosomal degradation. C3b would hereby either be present in close proximity to the coat or attached covalently or non-covalently to VSGs. C3(H$_2$O) and native C3 are depicted in the solution, as it is currently unknown whether they bind to the trypanosome surface. Surface receptors embedded within the VSG coat are depicted in different colours. The atomic structures of C3(H$_2$O) and the AP C5 convertase have not been determined and are therefore shown as dashed lines. Their overall architectures are presented as inferred earlier[15,28,68].

and likely takes over a role in facilitating innate immune escape through its interaction with human complement C3b. We show that ISG65 is competent to restrict the alternative complement pathway via C5 convertase, but that it does not play a role in the control of the classical and lectin pathway.

ISG65 is evolutionary related to VSGs and characterised by a similar, membrane-distal head domain. Intriguingly, unlike in the structures of GPI-anchored trypanosome receptors, the head domain has not been 'repurposed' for ligand binding[39–41]. It likely acts as an immuno-dominant decoy analogous to the same region in VSGs[42–44]. In ISG65, the ligand binding site is located at the membrane-proximal side of the helix bundle, near the C-terminus. The TED of C3 is the main site of interaction with ISG65, whereby its helix bundle follows the curvature of the globular TED. This strictly complementary conformation likely accounts for the high affinity in the absence of many hydrogen bonds or salt bridges. Additional contact points are present between the membrane-proximal end of ISG65 and ANA in native C3 as well as the head domain of ISG65 and CUB in C3b. Both interactions occur between less structured regions and may only be of transient nature, possibly contributing to correctly position ISG65 in different C3 conformations. For ANA, this is corroborated by the fast on- and off-rates that were measured between ISG65 and C3a.

While this study was under review, the crystal structure of ISG65 from the animal parasite *T. b. brucei* in complex with human C3d was published[45]. *Tbb* ISG65:C3d (PDB 7PI6) and *Tbg* ISG65:C3(TED) can be aligned with an RMSD = 1.7 Å (1791 atoms), confirming high similarity of the overall assembly (Supplementary Fig. 9a). Interface residues are present in similar positions in both structures with the exception of Arg1134 in TED, which is located in a long, flexible loop, and Trp211 in ISG65. Although likewise positioned in a loop region, the electron density for *Tbg* Trp211 is well-resolved and the region is protected from deuterium exchange. The importance of Trp211 for the interaction is further corroborated by affinity measurements using an alanine mutant, resulting in significantly reduced K$_D$. In contrast, in *Tbb* ISG65:C3d, Trp211 is not engaged with any C3d residues.

Using SPR, the authors of the aforementioned study[45] reported a comparatively low affinity for C3d and additionally claimed to have identified a second binding site located within human C3c (C3b lacking CUB and TED). This contradicts our own findings. *Tbg* ISG65, the receptor that encounters human C3 under natural infection conditions, showed a two magnitude higher affinity for C3d and revealed no interaction with C3c. This is in agreement with our cryo-EM structures which show no evidence for additional contact points with C3c.

As blood-dwelling parasites, African trypanosomes are in constant contact with toxic serum proteins that need to be swiftly removed from the cell surface before they can exert their damaging effects. The underlying mechanism has been described for VSGs and immunoglobulins. Motion-induced drag forces transport VSG:IgG complexes to the flagellar pocket at the posterior of the cell, where they are endocytosed[5]. Analogous to VSGs, ISG65 could mediate immune evasion by serving as a specific, high-affinity attachment point for C3b above the VSG layer, efficiently delivering a toxic cargo first to the flagellar pocket and then to the lysosome for degradation. This is in agreement with the high recycling rate, similar to VSGs, that has been shown for ISG65[46]. Efficient surface removal could thus help in preventing a critically high density of C3b on the surface that has been associated with C5 convertase formation[47]. While we did not demonstrate the surface binding of native C3 and C3(H$_2$O), it is possible that ISG65 can interact with these fragments in solution after being released from the surface. ISG65-shedding was reported to occur during the differentiation of *T. brucei*[38].

Although ISG65 is the likely site for C3b binding on the parasite surface, we cannot rule out that a fraction of C3b covalently attaches to the cell membrane, where it could serve as a nidus for convertase formation. Therefore, any capacity of ISG65 to block pathway progression, besides surface removal of C3b, could display an additional mechanism to avoid complement-mediated lysis.

Intriguingly, abrogation of haemolysis by soluble IGS65 and thus TP inhibition was only observed following initiation of the AP, but not of the CP. In the CP/LP, C3b is only part of the C5 convertase

C4bC2aC3b. Therefore, if ISG65 is binding to the CP/LP C5 convertase via C3b, it does not affect the cleavage of the C5 substrate and thus could not serve to restrict pathway progression.

Moreover, our immunofluorescence experiments (Fig. 3) indicate that in *T. b. gambiense* the CP/LP does not even progress to the formation of the CP/LP C3 convertase C4bC2a as no C4b was detected in the flagellar pocket. It is, however, unclear if the CP/LP is not triggered, as suggested earlier[21], or if it is inhibited upstream of C4b deposition. In contrast, we could identify Bb, a convertase component of the AP, but not C5b, the first factor of the TP, on the parasite surface, suggesting that the complement cascade is restricted upstream of TP initiation. These results are in agreement with an early study that showed deposition of AP convertases on the surface of *T. b. gambiense*, but found no evidence for activation of the CP[21]. They do, however, contradict the aforementioned recent study[45], which proposes the CP as the likely target of ISG65 based on mouse infection experiments which showed a decrease in parasitaemia progression of an ISG65 knock-out strain, temporally coinciding with the induction of infection-specific IgGs.

Here we provide detailed molecular insight, demonstrating that an effect of ISG65 on the CP/LP via binding to C3b is indeed unlikely and postulate the AP to be the most probable target.

However, how exactly does ISG65 restrict the AP via binding to C3? ISG65 binding to C3b and native C3 does neither prevent assembly of the C3 convertase (C3bBb) and the enzyme-substrate complex (C3bBb:C3), nor does binding to C3 substrate prevent its cleavage by C3bBb (Supplementary Fig. 5). It is, therefore, conceivable that inhibition of the cascade takes place at the level of the AP C5 convertase C3bBbC3b. Here, we could show that binding of soluble ISG65 to C3bBbC3b, pre-assembled on erythrocytes surfaces, significantly reduces cell lysis. This could be due to binding of ISG65 to either one or both C3b molecules of the convertase, thereby possibly reducing C5 substrate affinity and turnover. When ISG65 is additionally allowed to interfere with the assembly of the convertase, as we demonstrate by adding erythrocytes to serum preincubated with ISG65 (Fig. 4), haemolysis is almost fully abrogated. Therefore, in vitro, full inhibition of the AP C5 convertase appears to require ISG65 interference with both, convertase assembly as well as substrate cleavage. Due to the lack of structural information on the AP C5 convertase architecture, it remains to be elucidated which specific C3b interaction of ISG65 could hereby be rate-limiting for C5 turnover. Additional studies will thus be needed to obtain this crucial insight, but also to unravel which further molecular countermeasures trypanosomes employ to withstand other innate immune responses or prevent their activation entirely.

## Methods

### Cloning, expression and purification

**ISG65**. A DNA fragment (Genewiz) encoding amino acids (aa) 18–363 of ISG65 from *T. b. gambiense* (Tbg.972.2.1600) was codon optimised for bacterial expression and cloned into the pET15b plasmid using the Gibson assembly method (New England Biolabs). For recombinant expression in *E. coli* T7 shuffle cells (New England Biolabs), a plasmid encoding an N-terminal hexahistidine-tag (His-ISG65$_{18\text{-}363}$) and another with an additional C-terminal Avi-tag (His-ISG65$_{18\text{-}363}$-Avi) were generated. Amino acids substitutions were introduced into the His-ISG65$_{18\text{-}363}$ construct using the Q5 site-directed mutagenesis kit (New England Biolabs). For all ISG65 variants, the same expression and purification protocol was employed. Proteins were produced overnight at 22 °C after induction of protein expression with 1 mM iso-propyl-β-D-thiogalacto-pyranoside (IPTG).

Cells were harvested by centrifugation at $6000 \times g$, 4 °C for 30 min. Cell pellets were resuspended in Buffer A (20 mM Tris pH 8.0, 500 mM NaCl, and 10 mM imidazole) and phenylmethylsulfonyl fluoride (PMSF) was added to a final concentration of 1 mM immediately before cell lysis using an EmulsiFlex-C3 (AVESTIN Europe) with

1000 to 1100 bar lysis pressure. The supernatant was cleared from cell debris by centrifugation at $40.000 \times g$, 4 °C for 1 h.

Soluble protein was subsequently purified from the cleared lysate by immobilised metal-affinity chromatography (IMAC) using nickel-nitrilotriacetic acid (NTA) beads (Qiagen), pre-equilibrated with Buffer A, and gravity-flow columns (Bio-Rad).

After application of the lysate, the beads were washed twice with 10 column volumes (CV) of Buffer A before eluting the immobilised proteins with 10 CV Buffer A, supplemented with 400 mM imidazole. The eluate was fractionated and collected. Fractions containing the protein of interest, as identified by SDS-PAGE, were pooled and dialysed overnight into 20 mM Tris (pH 8.0), 150 mM NaCl at 4 °C, using SnakeSkin Dialysis Tubing (Thermo Fisher Scientific). After dialysis, the purified proteins were concentrated using Amicon Ultra centrifugal filters (Merck Millipore) and subjected to size-exclusion chromatography using a Superdex 200 Increase 10/300 GL (GE Healthcare) equilibrated with 20 mM HEPES (pH 7.5) and 150 mM NaCl. 0.5 mL fractions were collected throughout, and fractions containing the protein of interest, as identified by SDS-PAGE, were pooled and flash-frozen in liquid nitrogen.

**ISG75**. The gene fragment encoding the extracellular domain of ISG75 (*T. b. gambiense* LiTat 1.3, accession number DQ200220.1) comprising residues 29–462 of the full-length protein was cloned into the mammalian expression vector pHLsec[48]. It encodes an N-terminal secretion sequence and a thrombin-cleavable, C-terminal hexahistidine-tag. Transfection of Expi293F cells (Thermo Fisher) was carried out according to manufacturer instructions. Briefly, 100 ml of exponentially growing cells were transfected with 100 µg of the plasmid. About 5 µM Kifunensine was added to the cell suspension immediately after the transfection to produce a homogenous glycosylation pattern. Cells were harvested 2 days after transfection by centrifugation (10 min, $2000 \times g$, 20 °C). The supernatant was filtered using a 0.45 µm bottle top filter unit and dialysed twice for at least 4 h against 20 volumes of IMAC equilibration buffer (20 mM Tris, 500 mM NaCl, 10 mM Imidazole, and pH 8.0). The dialysed cell supernatant was applied to Nickel-NTA agarose beads (Qiagen) using a gravity-flow column (Bio-Rad), washed with 20 column volumes of IMAC wash buffer (20 mM Tris, 500 mM NaCl, 30 mM Imidazole, and pH 8.0) and the bound protein was eluted with five column volumes of IMAC elution buffer (20 mM Tris, 500 mM NaCl, 400 mM Imidazole, and pH 8.0). Cleavage of the His-tag was carried out by the addition of thrombin (GE Healthcare) to the eluate, which was immediately dialysed against IMAC equilibration buffer at 20 °C overnight. The cleaved protein was separated from the uncleaved protein by applying it again to Nickel beads. The flow-through was subsequently concentrated to 500 µL using an Amicon centrifugal ultrafiltration device (Merck Millipore) with a 10 kDa molecular weight cut-off ($3300 \times g$, 4 °C) and injected onto a Superdex S200 10/300 increase size-exclusion column (GE Healthcare) previously equilibrated with SEC running buffer (20 mM HEPES, 150 mM NaCl, and pH 7.5). The fractions containing *Tbg*ISG75 were pooled, flash-frozen in liquid N$_2$ and stored at −80 °C. Each purification step was monitored by SDS-PAGE for the absence of contaminations as well as degradation and >95% purity of the target protein in the final step.

**Human complement factor 3**. Native C3 was purified from normal human serum (Sigma-Aldrich)[49].

In short, 15% PEG4000 in 0.1 M sodium phosphate (pH 7.4) was added to serum to a final concentration of 5% PEG4000. The precipitate was removed by centrifugation ($6000 \times g$, 4 °C, 30 min). 26% PEG4000 in 0.1 M sodium phosphate (pH 7.4) was added to the supernatant to a final concentration of 12% PEG4000. After centrifugation ($9000 \times g$, 4 °C, 40 min), the supernatant was discarded and the pellet was immediately resuspended in 0.1 M sodium phosphate (pH 7.4). PMSF was added to a final concentration of 0.5 mM.

Using a gravity-flow column (Bio-Rad), the sample was passed over 25 mL of L-lysine-ceramic hyper D F Hydrogel resin (Sigma-Aldrich), pre-equilibrated with 0.1 M sodium phosphate (pH 7.4) and the flow-through was collected. The resin was washed with 100 mL of cold EDTA (5 mM, pH 8) and the wash fraction was collected. Flow-through and wash were combined and diluted to a specific conductance of 2.5 mS/cm with cold EDTA (5 mM, pH 8). 6-aminohexanoate and PMSF were added to a final concentration of 50 mM and 0.5 mM, respectively. The sample was loaded onto a hand-packed Source 15Q (Amersham) anion exchanger column (10 mL bed volume), pre-equilibrated with Buffer A (25 mM sodium-potassium phosphate, 10 mM EDTA, 50 mM 6-aminohexanoate, 0.5 mM PMSF, pH 7.4). After washing with 5 column volumes of Buffer A, a linear gradient to 100% Buffer B (25 mM sodium-potassium phosphate, 10 mM EDTA, 50 mM 6-aminohexanoate, 0.5 mM PMSF, 500 mM NaCl, pH 7.4) was applied over 30 column volumes, 1 mL fractions were collected throughout the elution. Fractions containing C3 were identified using SDS-PAGE, pooled and concentrated using an Amicon centrifugal ultrafiltration device (Merck Millipore) with a 50 kDa molecular weight cut-off (3300 × $g$, 4 °C) before being subjected to size-exclusion chromatography using a Superdex 200 16/600 column (GE Healthcare), pre-equilibrated with SEC running buffer (20 mM HEPES, 150 mM NaCl, pH 7.5). Fractions containing the protein of interest, as identified by SDS-PAGE, were pooled and flash-frozen in liquid nitrogen.

Purification and generation of the fragments C3b and C3MA were carried out as described[24].

In short, the thioester in C3 was hydrolysed using methylamine (MA) and deactivated by incubation with iodoacetamide. C3c and C3a were purchased from Complement Technology, Inc. C3d (His-C3d) was produced recombinantly in *E. coli* as described[50]. Amino acid substitutions were introduced using the same method as for ISG65. For SPR assays, a C3d construct (His-C3d-Avi) with an N-terminal His-tag and a C-terminal Avi-tag was generated. Cloning of this construct into pET15b was done by the Gibson assembly method. Expression and purification of all C3d constructs were carried out in the same way as described for ISG65.

## Pull-down assay
About 400 µL of settled Ni-NTA resin (Qiagen) were incubated with 200 µg of His-tagged ISG65, washed with 2.5 ml IMAC wash buffer (20 mM Tris, 500 mM NaCl, 20 mM Imidazole, pH 8.0) and incubated with 2.5 ml of human serum. Subsequently, the beads were washed again using the same buffer, before the bound protein was eluted in a total volume of 2 ml IMAC elution buffer (20 mM Tris, 500 mM NaCl, 400 mM Imidazole, pH 8.0). Four fractions were collected and analysed by SDS-PAGE and Coomassie-staining. All steps were carried out in gravity-flow columns (Bio-Rad). ISG65 (without serum) and serum (without ISG65 as bait protein) were used as negative controls.

## Western blotting
All Western Blots were performed using the iBlot Dry Blotting System (Invitrogen) and iBlot PVDF Mini Gel Transfer Stacks (Invitrogen) as per the manufacturer's instructions. Blotting was conducted using 'Programme 3' with default settings (20 V, 7 min). After the transfer, membranes for Western Blots of His-tagged proteins were blocked (1 h, RT, shaking) using a Blocking buffer (Penta-His HRP Conjugate Kit; Qiagen), prepared according to the manufacturer's instructions. The membrane was washed three times with PBS-T (0.05% Tween 20) for 5 min, before incubating in a working solution (Blocking Buffer, 0,05% Tween 20, Anti-His HRP conjugate [1/2000 dilution]) for 1 h. The membrane was washed three times for 10 min with PBS-T before removing the wash and developing the Western Blot using Pierce ECL Western Blotting Substrate (Thermo Fisher Scientific) according to the manufacturer's instructions. Blots were imaged within 30 min of substrate addition.

After Western Blotting of C3a-containing proteins, membranes were blocked in a Blocking solution (25 mg/mL bovine serum albumin (BSA) in PBS-T) (1 h, RT, shaking).

After rinsing with PBS-T, the membrane was incubated in a working solution (12.5 mg/mL BSA in PBS-T, Mouse anti-human C3a (MAB3677, R&D systems) [1/500 dilution]). After washing three times for 10 min with PBS-T, the membrane was incubated with the detection antibody (12.5 mg/mL BSA in PBS-T, Goat Anti-mouse HRP conjugate (A28177, Invitrogen) [1/10000 dilution]) for 1 h. After washing three times for 10 min with PBS-T, the wash was removed and the Western blot was developed using Pierce ECL Western Blotting Substrate (Thermo Fisher Scientific) according to the manufacturer's instructions. Blots were imaged within 30 min of substrate addition. All Chemiluminescence and digitised images were acquired on Image-Quant LAS 4000 and ImageQuant LAS 4000 mini camera systems (GE Healthcare).

## In-gel digestion and LC-MS/MS analysis
Single gel bands (three in total), as depicted in Supplementary Fig. 1, were cut, chopped into small pieces, reduced with dithiothreitol, alkylated with iodoacetamide and digested with trypsin overnight. Peptides were extracted from each gel piece, lyophilised in speed-vac, dissolved in 0.1% formic acid and 1/5th of the dissolved sample was separated on an UltiMate 3000 RSLCnano system (Thermo Fisher Scientific) coupled to a Mass Spectrometer Orbitrap Fusion Lumos (Thermo Fisher Scientific). No normalisation was carried out since bands were cut manually. The peptides were trapped and desalted with 2% acetonitrile in 0.1% formic acid at a flow rate of 5 µL/min on an Acclaim PepMap100 column (5 µm, 5 mm by 300 µm internal diameter (ID); Thermo Fisher Scientific). Eluted peptides were separated using an Acclaim PepMap100 analytical column (2 µm, 50 cm × 75 µm ID; Thermo Fisher Scientific). Using a constant flow rate of 300 nL/min a 65 min elution gradient was started at 5% B (0.1% formic acid in 99.9% acetonitrile) and 95% A (0.1% formic acid). The gradient reached 30% B at 52 min, 90% B at 53 min, and was then kept constant until 57 min before being reduced to 5% B at 58 min. For the first minute, nanospray was set to 1600 V, 275 °C source temperature, measuring the scans in the range of m/z 350–2000. An orbitrap detector was used for MS with the resolution 120,000, AGC target value was set as custom with a normalised AGC target of 250%. Maximum injection time was set to 50 ms, MSMS was acquired also using orbitrap with resolution 30,000, the data were acquired in a data-dependent manner, ions were fragmented by HCD collision energy set to 30% with dynamic exclusion set to 60 s.

The software Proteome Discoverer 2.3 (Thermo Fisher Scientific) was used for peptide and protein identification using Sequest and MS Amanda as search engines and databases of *Homo sapiens* (downloaded from Uniprot combining reviewed and unreviewed sequences on November 12 2018) and common contaminants (downloaded from Proteome Discoverer software on December 21 2018). The protein sequence of ISG65 was included in a database for the search. Mass tolerance for MS was 10 ppm and MSMS 0.6 Da for Sequest, tolerance in MS 5 ppm and MSMS 0.06 Da in MS Amanda. The fixed modification included carbamidomethylation on cysteine, variable modifications were set to methionine oxidation and deamidation of asparagine and glutamine, two missed cleavages were allowed, FDR target was set to 1%.

## Cross-linking
ISG65:C3b complex was cross-linked using homobifunctional cross-linker DSA (Di-(N-succinimidyl)adipate). For cross-linking, proteins were transferred into a buffer containing 20 mM HEPES (pH 7.5) and 150 mM NaCl. ISG65 and C3b were mixed in a 2:1 molar ratio, and freshly prepared DSA was added to the proteins at a hundred molar excess. The reaction mixture was incubated at room temperature for

30 min and quenched with 10 molar excess of ethanolamine. For EM-grid preparation, the cross-linked complex was also gel-filtered on Superdex 200 Increase 10/300 GL (GE Healthcare).

## C3b surface binding and uptake analysis

About $4 \times 10^5$ *T. brucei gambiense* cells were harvested from a mid-exponential grown culture and resuspended in 200 μl serum-free HMI-11 supplemented with 1% bovine serum albumin. Uptake assays were carried out with AF$_{594}$ labelled C3b at a concentration of 50 μg/ml or AF$_{594}$ labelled human transferrin (T13343, Thermo Fisher Scientific) as a positive control at a concentration of 50 μg/ml, with or without the addition of recombinant ISG65 or ISG75 extracellular fragment at a 1.5-fold molar excess (preincubated with C3b for 10 min). Samples were incubated at 37 °C and harvested at different timepoints. To analyse the contribution of surface binding, samples were pre-cooled on ice for 10 min before mixing with test proteins and harvested immediately. To demonstrate specific, receptor-mediated uptake, we compared C3b$_{AF594}$ surface binding and uptake at a concentration of 10 μg/ml in the presence and absence of non-labelled C3b at a concentration of 40 μg/ml (adding 80 μg/ml ISG65 or ISG75, respectively). Cells were harvested by centrifugation ($1400 \times g$, 4 °C), washed twice with ice-cold PBS and resuspended in 500 μl PBS. Cells were fixed for 30 min at RT after the addition of an equal volume of 4% formaldehyde in PBS, then washed once with PBS and resuspended in a final volume of 300 μl PBS. An aliquot of each sample was mounted on a glass slide with Vectashield Antifade mounting medium with 2-(4-amidinophenyl)−1H-indole-6-carboxamidine (DAPI) for fluorescence microscopy imaging on a Leica TCS SP8 WLL SMD-FLIM inverted confocal microscope. Images were processed in Fiji[51]. Samples were analysed by flow cytometry on an LS Fortessa (BD Biosciences) using an excitation wavelength of 560 nm in combination with a PE-Texas Red filter ($\lambda_{emission} = 610$ nm, bandpass = 20 nm) and counting $10^4$ cells. Data were processed using BD FACSDiva software v8.0.1 using a refined gate based on size versus granularity to exclude cell debris. Untreated cells were subjected to the same procedure in parallel as negative control and used to define the gate for AF$_{594}$ positive cells. Flow cytometry standard (FCS) files used in this study are available in the Flow-Repository database[52] under accession code FR-FCM-Z5XC.

## Detection of complement factor surface deposition by immunofluorescence

About $1 \times 10^5$ *T. brucei gambiense* cells were harvested from a mid-exponential grown culture ($800 \times g$, 8 min, RT), washed once in 1 ml serum-free HMI-11 supplemented with 1% bovine serum albumin and then resuspended in 100 μl human serum (complete serum, C2-depleted serum (C0913, Sigma) or Fb-depleted serum (A506; Quidel)) containing 5 μg (5% [w/v] dilution) AF594 (complement factor Bb monoclonal antibody (10-09; Invitrogen); complement C4 polyclonal (ab47788; Abcam); complement C5 polyclonal antibody (PA596933; Invitrogen) or fluorescein isothiocyanate (FITC) labelled antibody (complement C3 polyclonal, (PA1-28933, Invitrogen)). Unlabelled antibodies were fluorescently labelled using the Alexa Fluor 594 Conjugation Kit (Fast)−Lightning-Link (Abcam).

After 2 min at RT, 10 mM sodium azide was added and cells were sedimented ($800 \times g$, 8 min, 4 °C). Cells were resuspended in 100 μl PBS containing the respective antibody (at the same dilution) and incubated on ice for 2 min before addition of 0.2% formaldehyde, followed by two washes in ice-cold PBS and mounting on microscopy slides. For testing complement deposition on sensitised sheep erythrocytes, 2 μl of concentrated cells (CH50 Test, K002-1; Haemoscan) were diluted in 1 ml diluent (K002-2, Haemoscan), sedimented by centrifugation ($400 \times g$, 5 min, 4 °C) and then resuspended in 100 μl human serum (C6-depleted serum (C1288, Sigma) or C7-depleted serum (C1413, Sigma)). Erythrocytes were incubated for 15 min at 37 °C, then sedimented ($400 \times g$, 5 min, 4 °C) and resuspended in 100 μl cold PBS containing 5 μg (5% [w/v] dilution) labelled antibody. After 15 min incubation under gentle tumbling, formaldehyde was added to 0.2% and erythrocytes were washed twice in ice-cold PBS. An aliquot of each sample was mounted on a glass slide with Vectashield Antifade mounting medium with DAPI for fluorescence microscopy imaging on a Leica TCS SP8 WLL SMD-FLIM inverted confocal microscope. Images were processed in Fiji[51].

## Haemolysis assays

**Normal human serum.** Inhibition of complement haemolytic activity by ISG65 and ISG75 (negative control) was assessed using the AP50 and CH50 test kits (HaemoScan), according to the manufacturer's instructions. Dilutions (2–15.2-fold (AP50) & 4-128-fold (CH50)) of non-heat inactivated human serum (Sigma-Aldrich) were incubated with 24 μM ISG65 and ISG75 (CH50 assay) or serially diluted ISG65 and ISG75 (7.5−0 μM) (AP50 assay).

Complement inhibition was determined from the extent of lysis of rabbit (AP50) or antibody-sensitised sheep erythrocytes (CH50) measured at 415 nm. The concentration of free haemoglobin is hereby directly proportional to complement activity. The measurements were performed in technical triplicates ($n = 3$, AP50) or quadruplets ($n = 4$, CH50). Graphs were created using GraphPad Prism.

**Complement factor depleted serum.** Inhibition of complement haemolytic activity by ISG65 and ISG75 (negative control) was assessed using the AP50 test kit (HaemoScan), using an adapted version of the manufacturer's instructions. Rabbit erythrocytes were incubated with dilutions (2–15.2-fold) of human complement C5-depleted serum (Sigma-Aldrich). After incubation for 30 min at 37 °C, the erythrocytes were centrifuged ($400 \times g$, 10 min, RT), the supernatant removed, and the erythrocytes washed twice with dilution buffer (HaemoScan). The erythrocytes were resuspended and dilutions (2–15.2-fold) of human complement factor C3-depleted serum (Sigma-Aldrich), were incubated with 7.5 μM ISG65 or ISG75 added. Lysis was performed for 30 min at 37 °C.

The modified protocol was validated by efficient lysis of erythrocytes when both sera were consecutively used (reconstituted NHS). Exchange of either serum with buffer prevented lysis. Complement inhibition was determined as described for normal human serum.

## Intact mass measurement

Samples were diluted to 0.2 mg/ml with 0.1% formic acid. About 5 μl were injected into a desalting column (MassPREP desalting, Waters) and desalted using a fast gradient (4 min) of acetonitrile in water supplemented with 0.1% formic acid. The separation was carried out by an LC system (I-class, Waters) that is online coupled to a mass spectrometer (Synapt G2, Waters) to acquire protein masses by electrospray ionisation. Prior to sample measurement, the system was calibrated with sodium formate in the range of 100−1500 m/z. The calibration was checked by measuring myoglobins ions in the same range. For myoglobin ions, the ppm error was less than 1 ppm. For the measurement of samples of interest, the instrument was set up to accept a calibration error of less than 10 ppm. The raw spectrum was processed in MassLynx (Waters). The deconvolution was performed in MaxEnt1 with the mass range 160,000−200,000 m/z, resolution 0.5 Da/channel and uniform Gaussian used to calculate the damage model (Waters). The final spectrum was produced in the mMass software (mmass.org)[53].

## N-terminal protein sequencing

Purified proteins were run on reducing SDS-PAGE, electroblotted onto PVDF membrane and stained with Coomassie Brilliant Blue R-250. The N-terminal amino acid sequence of the α-chain of C3b was determined by Edman degradation using Procise 494 cLC Protein Sequencing

System (Applied Biosystems), following the manufacturer's recommendation. Briefly, in each sequencing cycle, the N-terminus of the protein was treated with phenylisothiocyanate, and specifically released phenylthiohydantoin amino acid derivative was analysed by reverse-phase HPLC in 3.5% tetrahydrofurane and 12% n-propanol/acetonitrile mobile phases, respectively.

## Surface plasmon resonance

**Binding-impaired mutants.** Binding-impaired mutants of ISG65 and C3d were subjected to kinetic binding analysis using SPR. For interrogation of ISG65 mutants, His-C3d-Avi was used as a ligand, and for C3d mutants, His-ISG65$_{18-363}$-Avi was the ligand. Site-specific Avi-tag biotinylation for immobilisation on the SPR chip was carried out as previously described[54]. SPR experiments were performed at 25 °C using Series S sensor chip CAP (Cytiva) on a BIAcore T200 system (GE Healthcare). All binding analyses, as well as dilutions, were performed in SPR running buffer (20 mM HEPES pH 7.5, 150 mM NaCl, 3 mM EDTA, 0.005% (v/v) TWEEN-20). Biotinylated ligands were coupled to flow path 2 at 10 µl min$^{-1}$ for 120 s. Dilution series (1:1) of both analytes (ISG65, 1.5–100 nM; C3d, 0.78–50 nM) were applied to flow paths 1 and 2 at 30 µl min$^{-1}$ for 120 s, followed by 300 s of dissociation time. The chip surface was regenerated in between titrations with 6 M guanidinium hydrochloride dissolved in 0.25 M sodium hydroxide. The binding data was reference subtracted, and kinetic and steady-state affinity parameters were evaluated using BIAcore T200 evaluation software (GE Healthcare).

**C3 fragments.** Binding analyses of native C3, C3b, C3MA and C3c to immobilised His-ISG65$_{18-363}$-Avi was carried out analogous to the method described above for the binding-impaired mutants. The binding and dissociation times of analytes were altered depending on the kinetic profiles. For binding analyses of C3a, His-ISG65$_{18-363}$ was covalently immobilised via amine coupling according to the manufacturer's instructions (amine coupling kit, Cytiva). The reference channel was activated and subsequently quenched without immobilisation of a ligand. C3a (8, 4, 2, 1, 0.5, 0.25 µM) was flown over both channels at 30 µl min$^{-1}$, allowing 100 s for association and 300 s for dissociation. Between kinetic runs, the chip surface was regenerated with 1 M sodium chloride for 30 s at 30 µl min$^{-1}$.

## Disulphide mapping

For detection of disulphide bonds under low pH conditions to avoid disulphide-bridge scrambling, 100 pmol of ISG65 were diluted into 0.5 M glycine, pH 2.3 and online digested using a Nepenthesin-2 column (AffiPro)[55]. Generated peptides were trapped and desalted on a micro-trap column (Luna Omega 5 µm Polar C18 100 Å Micro Trap 20 × 0.3 mm) for 3 min at a 200 µL/min using an isocratic pump delivering 0.4 % formic acid in water. Both the protease column and trap column were placed in an icebox. After 3 min, peptides were separated on a C18 reversed-phase column (Luna Omega 1.6 µm Polar C18 100 Å, 100 × 1.0 mm) with a linear gradient 5–35% B in 26 min, where solvent A was 2% acetonitrile/0.4% formic acid in water and solvent B 95% acetonitrile/5% water/0.4% formic acid. Peptides were measured by a timsToF Pro PASEF mass spectrometer (Bruker Daltonics) operated in a positive data-dependent mode in the m/z range 300–1600. The scheduling target intensity was set at 20,000. The intensity threshold was 2500 and the PASEF charge range was 1–6. The mass spectrometer was externally calibrated using an ESI low-concentration tuning mix (Agilent Technologies). The data were processed by Data Analysis 5.0 software (Bruker Daltonics) and ProteinScape 4.0 (Bruker Daltonics). Non-specific cleavage and dehydro-cysteine were used as variable modifications in the peptide search. The peptide tolerance was 15 ppm and MS/MS tolerance was 0.1 Da. The data were searched using MASCOT against the database containing ISG65 protein. The MASCOT ions score cut-off was 2, the peptide rank cut-off was set at 3, and the minimal peptide length was 4.

## Cryo-electron microscopy

### Grid preparation and data collection

**ISG65:C3.** About 3 µl (0.16 mg/ml) of ISG65:C3 were applied to freshly glow-discharged copper C/Flat 1.2/1.3 300 mesh grids (Protochips) and vitrified by being plunged into liquid ethane using a Thermo Fisher Scientific Vitrobot Mark IV system (4 °C, 100% relative humidity, no wait time, 2 s blotting time). The grids were then transferred to a Titan Krios G3i microscope (Thermo Fisher Scientific) equipped with a K3 detector (GATAN Inc.) and operated at 300 kV. Images were recorded at a magnification of 105,000X while tilting the stage by 25°, yielding a pixel size of 0.86 Å. Multi-frame movies (40 frames with a total dose of 41.10 e$^-$/Å$^2$ were recorded using a nominal defocus (dF) range of −2.5 to −1.5 µm. Data were collected using the automated data collection software EPU (Thermo Fisher Scientific).

**ISG65:C3b.** To prepare grids used in the collection of datasets at a 25° tilt, 3.5 µl (0.1 mg/ml) of ISG65:C3b complex was applied to glow-discharged copper C/Flat, 300 mesh 1.2/1.3, and 2/2 TEM grids (Protochips). The grids were vitrified by being plunged into liquid ethane using the Thermo Scientific Vitrobot Mark IV system (4 °C, 100% relative humidity, 30 s waiting time, 4 s blotting time) and then transferred to a Titan Krios microscope (Thermo Fisher Scientific) for data acquisition.

For data collections at a 0° stage tilt, Au, 300 mesh, R1.2/1.3 TEM grids (Protochips) were coated with a graphene monolayer using an in-house developed protocol. About 3.5 µl (0.15 mg/ml) of ISG65:C3b complex was applied to freshly plasma-cleaned TEM grids and vitrified as explained for grids used to obtain the 25° tilt dataset. The grids were subsequently transferred to a Titan Krios microscope (Thermo Fisher Scientific) for data acquisition.

The data were collected at 300 kV using SerialEM[56] software. The data were collected on a K2 direct electron detection camera positioned behind a Gatan Imaging Filter (Bioquantum 967, Gatan). The camera was operated in the electron counting mode, and the data were collected at the calibrated pixel size of 0.818 Å/px. The data from a 5.0 s exposure were split into 40 frames comprising an overall dose of 60 e$^-$/Å$^2$.

### Image processing

**ISG65:C3.** A total of 10,380 movies were acquired during data collection. Motion correction, as well as CTF correction (CTFFIND4 wrapper), were done using cryoSPARC[57]. About 5,508,854 particles were picked using the reference-free Gaussian Blob picker in cryoSPARC. Particles were extracted with a 400 px box with twofold binning applied, which resulted in a pixel size of 1.72 Å/px. Iterative panning using reference-free 2D classification in cryoSPARC was performed until a subset of 432,224 particles was identified. The particles were re-extracted using a 400 px box with no binning applied. The re-extracted particles were submitted to another round of reference-free 2D classification in Relion 3.1[58]. About 406,545 particles were selected and used for ab initio modelling and subsequently subjected to the 3D classification job in Relion. Particles were classified into five classes, all yielding well-defined reconstructions of the desired complex. All particles selected after 2D classification in Relion were later used for the creation of an ab initio model in cryoSPARC, which was subsequently refined using the non-uniform refinement procedure (cryoSPARC v3.2 and later) and local CTF refinement to yield the final reconstruction.

**ISG65:C3b.** A total of 18,062 movies were collected at a 25° stage tilt in two separate data collections (for two different grid types), and 8960 movies were collected at a 0° stage tilt on graphene-coated grids. The processing of each dataset was performed independently before

 

merging selected particles with previous data collections. Motion correction and CTF correction for all 27,022 movies were performed using cryoSPARC's patch motion and patch CTF correction implementations, respectively. On images acquired at a 25° stage tilt, particles were picked using the convolutional neural network-based particle picker, crYOLO[59]. Particles were initially identified using the supplied general model. The resulting particle picks were manually inspected, and reference-free 2D classification was performed in cryoSPARC. Particles constituting the 2D classes which represented the protein (regardless of whether they were the desired complex or not) were selected and used to re-train the outer layers of the general model on a subset of micrographs at varying defocus. crYOLO identified a total of 2,836,445 particles (1,414,868 and 1,421,577). The processing of the two datasets followed the same overall processing pipeline, using cryoSPARC. Particles were extracted using a box with 400 px and twofold binning, resulting in a pixel size of 1.656 Å/px. The iterative, reference-free 2D classification was performed until satisfactory 2D classes were identified. The selected particles were subjected to ab initio modelling using 3 classes. Models representative of the desired ISG65:C3b complex and C3b were chosen for subsequent Heterogenous Refinement. Particles constituting the reconstruction representing ISG65:C3b were selected, re-extracted using a 400 px box (no binning applied resulted in a pixel size of 0.828 Å/px), and subjected to Non-uniform Refinement.

For images acquired at a 0° stage tilt, the Gaussian particle picker in cryoSPARC was utilised to identify particles. The remaining processing was performed as described for data collected at a 25° stage tilt.

For the final reconstruction, 204,946 combined particles were re-subjected to ab initio modelling using 2 classes. The class representative of the desired complex was subjected to Non-uniform Refinement and local CTF refinement. A total of 145,172 particles, of which 54,453 particles came from datasets at 25° tilt (22,266 and 32,187, respectively) and 90,719 particles came from a 0° tilt dataset, contributed to the final reconstruction.

### Model building and refinement

**ISG65: C3**. The initial step in the model building was performed by docking starting models (the crystal structure of C3 (PDB 2A73)) and an ISG65 model predicted by AlphaFold2[34] into the obtained electron density map using Phenix dock_in_map[60]. After an initial real-space refinement using Phenix real_space_refine, both models were independently and iteratively refined using a combination of manual refinements in Coot as well as automated real-space refinements using Phenix and REFMAC5[61]. Model validation was performed using Mol-Probity (http://molprobity.biochem.duke.edu)[62]. After the individual models had reached satisfactory validation metrics and map correlation, they were manually and automatically refined together in Coot[63] and Phenix[60] to model interactions between the two proteins.

**ISG65: C3b**. Due to large variations in the local resolution of the obtained electron density map, the modelling of ISG65 at atomic resolution was not possible. Therefore, the atomic model obtained from the ISG65:C3 data was docked in the electron density map using Phenix dock_in_map[60]. For C3b, the same was done using the crystal structure of C3b (PDB 2I07). After an initial real-space refinement using Phenix real_space_refine[60], the model was iteratively refined using a combination of manual refinements in Coot[63] as well as automated real-space refinements in Phenix and REFMAC5[61]. Model validation was performed using MolProbity[62].

### Molecular modelling

The model of the full-length mature ISG65 (aa 18–436), excluding the signal sequence but including the transmembrane domain and the cytoplasmic tail, was predicted by using the cryo-EM structure of ISG65

(aa 35–316) as a template into AlphaFold2. The obtained molecular model was used to complete the experimentally determined structure of the C3-binding domain of ISG65. The model was previously incomplete due to disorder in the N-terminal portion (aa 18–34), in the loop-rich membrane-distal head domain (aa 147–201 and 227–252), and in the C-terminal portion. A model was selected based on the presence of disulphide bonds that were determined experimentally prior to modelling as well as by the general fit of their head domains to the experimental electron density that delineated this region. No constraints were imposed on the selection of the C-terminal portion that is composed of a long-disordered C-terminal linker (aa 317–363), a C-terminal alpha-helix with an extracellular (aa 364–387) as well as membrane-embedded part (aa 387–410), and the cytoplasmic tail.

### Circular dichroism spectroscopy

For ISG65, far-UV CD experiments were carried out on a Jasco J-1500 spectropolarimeter with a 0.2 mm path cell. ISG65 was dissolved in 10 mM HEPES pH 7.5, 150 mM NaF at a concentration of 0.4 mg/ml. Spectra were recorded between 195 and 260 nm wavelength at an acquisition speed of 10 nm/min and corrected for buffer absorption. For the calculation of melting curves at 222 nm, spectra were recorded every 5 degrees between 5 °C and 80 °C with a slope of 0.16 °C/min. During measurements, the temperature was kept constant. The raw CD data (ellipticity θ in mdeg) were normalised for the protein concentration and for the number of residues, according to the equation below, yielding the mean residue ellipticity ([θ] in deg cm² mol⁻¹), where MM, n, C and l denote the molecular mass (Da), the number of amino acids, the concentration (mg/mL), and the cuvette path length (cm), respectively.

$$[\theta] = \frac{\theta \cdot MM}{n \cdot C \cdot l}$$

### Hydrogen-deuterium exchange mass spectrometry

**Peptide mapping.** 300 pmol of ISG65 or C3d were mixed in a 1:1 (v/v) ratio with 1 M glycine, 200 mM Tris(2-carboxyethyl)phosphine (TCEP) at pH 2.3 and injected onto a Nepenthesin-2 column (Affipro). Generated peptides were trapped and desalted using a micro-trap column (Luna Omega 5 µm Polar C18 100 Å Micro Trap 20 × 0.3 mm) for 3 min at a flow rate of 400 µl/min using an isocratic pump delivering 0.4% (v/v) formic acid in water. Both the protease column and the trap column were placed in an icebox. After 3 min, the peptides were separated on a C18 reversed-phase column (Luna Omega 1.6 µm Polar C18 100 Å, 100 × 1.0 mm) with a linear gradient of 5–35% B over 26 min, where solvent A was 2% (v/v) acetonitrile/0.4% (v/v) formic acid in water and solvent B was 95% (v/v) acetonitrile/4.5% (v/v) water/0.4% (v/v) formic acid. The analytical column was placed in an icebox. A 15 T solariX XR FT-ICR mass spectrometer (Bruker Daltonics) operating in positive MS/MS mode was used for the detection of peptides. Data were processed by Data Analysis 4.2 software (Bruker Daltonics) and Protein-Scape 4.0 (Bruker Daltonics). The peptide ion tolerance was 5 ppm and MS/MS tolerance was 0.05 Da. No enzyme specificity was set for peptide search. MASCOT ion score cut-off was set at 2, peptide rank cut-off at 3 and minimal peptide length was 4. The MASCOT search engine was used for the identification of peptides, using a database containing the sequence of ISG65.

**Hydrogen-deuterium exchange.** Hydrogen-deuterium exchange (HDX) was initiated by a tenfold dilution of 160 µM ISG65: C3d complex into a deuterated buffer (20 mM HEPES, 150 mM NaCl, pD 7.5). The hydrogen-deuterium exchange reaction was performed at RT. 50 µl aliquots (100 pmol) were taken after 20, 120, 1200 and 7200 s of incubation in deuterated buffer and quenched by the addition of 50 µl of 1 M glycine, 200 mM TCEP at pD 2.3, followed by immediate freezing

using liquid nitrogen. Aliquots were quickly thawed and analysed as described for peptide mapping. Peptides were separated by a linear gradient of 10–30% B over 18 min. The mass spectrometer was operated in positive MS mode in the mass range 350–2000 m/z. Data were measured with 1 Mb acquisition with two selective accumulations. The capillary voltage was set at 3900 V, the drying gas temperature was 180 ˚C and the nebuliser gas was 2.0 bar. Spectra of partially deuterated peptides were processed by Data Analysis 4.2 (Bruker Daltonics, Billerica, MA) and the in-house programme DeutEx[64].

In the DeutEx software calculation of the envelope width intensity was set at 25, the error for deuteration was 10 ppm and the error for isotopes was 10 ppm. An intensity filter was set at a value of 25 and a rate filter was set at 5. No back exchange correction was applied, because quantification of the absolute amount of exchange was not desired.

### Small-angle X-ray scattering and ensemble optimisation

Small-angle X-ray scattering (SAXS) data were collected at ESRF, Grenoble (France), using SAXS beamline BM29 with a wavelength of 0.99 Å on a Pilatus 2 M detector (DECTRIS) at 20 °C. For SEC-SAXS, 50 µl of His-ISG65$_{18-363}$ at 11 mg/ml were injected onto a Superdex 200 3.2/300 column (equilibrated in 20 mM HEPES pH 7.5, 150 mM NaCl, 3% (v/v) glycerol) at a flow rate of 75 µl/min. Scattering data were acquired as components eluted from the column and passed through the SAXS measuring cell. The ATSAS[65] software package was used to normalise the data to the intensity of the incident beam, to average the frames, and to subtract the scattering contribution from the buffer. In detail, 10 frames corresponding to the void volume of the column were averaged and subtracted from ten averaged frames of the main elution peak. The radius of gyration ($R_g$), maximum particle dimension ($D_{max}$), and distance distribution function ($p(r)$) were evaluated using the programme PRIMUS[66] as part of the ATSAS package.

The ensemble optimisation method (EOM 2.0)[33] as part of ATSAS online was used to determine the flexibility and conformational dynamics of the disordered C-terminal linker region (aa 317–aa 366) connecting the structured, C3-binding domain of ISG65 (aa 18–316) to the C-terminal, partially membrane-embedded, alpha-helix that was predicted using AlphaFold2. The EOM 2.0 web-based application was run with default settings using scattering data obtained for ISG65$_{18-363}$ in combination with the cryo-EM structure of ISG65. To generate the final model of the full-length ISG65$_{18-436}$, the C-terminal helix and the cytoplasmic tail (aa 364–436) from the ISG65 AlphaFold model were appended to SAXS models representative of compact and extended conformations within the EOM obtained ensemble.

### *T. b. gambiense* culturing

Bloodstream form *T. b. gambiense* DAL972 was cultured in HMI-11 complete medium (HMI-11 supplemented with 10% (v/v) fetal bovine serum (non-heat-inactivated), 100 units/ml penicillin and 100 units/ml streptomycin)[67] at 37 °C with 5% $CO_2$ in a humid atmosphere, in culture flasks with vented caps.

### Reporting summary

Further information on research design is available in the Nature Portfolio Reporting Summary linked to this article.

## Data availability

Models of the presented complexes have been deposited in the Protein Data Bank under accession codes 7ZGJ (ISG65:C3) and 7ZGK (ISG65:C3b). The associated cryo-EM density maps have been deposited in the Electron Microscopy Data Bank under accession codes EMD-14707 (ISG65:C3) and EMD-14708 (ISG65:C3b). The hybrid model of ISG65 has been deposited in PDBDev under accession code PDBDEV_00000201. Starting models of C3b and C3 used in the modelling of the complexes are deposited in the Protein Data Bank under accession

codes 2I07 (C3b) and 2A73 (C3). The SAXS data generated in this study is available in SASBDB under accession codes SASDP99 (ISG65) and SASDPA9 (ISG65:C3d). The mass spectrometry proteomics data have been deposited to the ProteomeXchange Consortium via the PRIDE partner repository with the dataset identifiers PXD036611 (C3 and proteolytic fragments) and PXD033606 (Analysis of ISG65 disulfides and HDX-MS). Flow cytometry standard (FCS) files used in this study are available in the FlowRepository database (https://flowrepository.org/) under accession code FR-FCM-Z5XC. Complete Flow cytometry traces are provided in Supplementary Data 1. Peptide mapping, HDX-MS analysis of the ISG65:C3d complex, and HDX-MS summary table are provided in Supplementary Data 2. All other source data are provided in the Source Data file. Source data are provided with this paper.

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

## Acknowledgements

We thank Josef Houser for assistance during SPR measurements and acknowledge CF Biomolecular Interactions and Crystallisation of CIISB, Instruct-CZ Centre, supported by MEYS CR (LM2018127), for providing access to the SPR instrument. We thank Jiri Novacek for assistance during cryo-EM data collection and acknowledge cryo-electron microscopy and tomography core facility CEITEC MU of CIISB, Instruct-CZ Centre supported by MEYS CR (LM2018127). We also acknowledge CMS-Biocev ('Biophysical techniques, Crystallisation, Diffraction, Structural mass spectrometry') of CIISB, Instruct-CZ Centre, supported by MEYS CR (LM2018127) and CZ.02.1.01/0.0/0.0/18_046/0015974 with regard to MS measurements carried out by Petr Pompach. This publication was developed under the provision of the Polish Ministry of Education and Science project: 'Support for research and development with the use of research infrastructure of the National Synchrotron Radiation Centre SOLARIS' under contract nr 1/SOL/2021/2. We thank Anton Popov at the ESRF (BioSAXS beamline BM29) for outstanding beamline support, Lucie Bednarova and Zdenek Voburka from IOCB for performing CD measurements and N-terminal sequencing, Philippe Büscher from the Institute of Tropical Medicine in Antwerp, Belgium for the donation of the *T. b. gambiense* strain that was used in this study and Ondrej Honc and Marie Olsinova from the Imaging Methods Core Facility at Biocev for excellent technical support. Finally, we thank Alzbeta Kadlecova from Sebastian Zoll's group at IOCB for her invaluable administrative support throughout the project. Research in S.Z.'s lab is supported by the Czech Science Foundation (project 22-21612 S). H.S. was supported by the Grant Agency of Charles University (project no. 383821/2600). A.D. was supported by a post-doctoral fellowship from IOCB. M.Z. was supported by a grant from the Czech Ministry of Education (project OPVVV/0000759).

## Author contributions

H.S. and S.Z. conceptualised the study. Initial pull-down experiments were performed by SZ. J.V., M.S. and J.B. cloned the recombinant expression constructs. H.S. and J.B. performed protein expression, purifications and SPR measurements, assisted by A.S. and M.T. Haemolysis assays were performed by H.S. S.K. prepared and subsequently screened for suitable TEM grids for ISG65:C3 data collections. Fluorescence microscopy and FACS measurements were performed by M.Z. and F.Z. P.P. performed HDX-MS, XL-MS and SS-bond mapping experiments. M.H. performed intact MS and LC-MS/MS experiments. Cryo-EM data processing, subsequent model building, and molecular modelling was done by H.S. SAXS measurements were evaluated by H.S. and S.Z. H.S. and S.Z. wrote the manuscript with input from M.Z. A.D., H.S. and S.Z. revised and edited the manuscript.

## Competing interests

The authors declare no competing interests.
