## [Peer Review File · Nature Communications]

REVIEWER COMMENTS

Reviewer #1 (Remarks to the Author):

The manuscript establishes ISG65 as a complement C3 binding protein. The primary binding site of ISG65 is shown to be on the thioester domain of C3, which enables its binding to C3, C3water, C3b and the C3d fragment. Two cryo-EM structures are presented that convincingly establishes the binding site of ISG65 on C3 and C3b. The structural work is solid although minor issues need to be addressed. In contrast, the functional characterization of how ISG65 interfere with complement activation and the downstream effector functions is incomplete and the description of complement biology is too simple minded.

Major issues:

Abstract “we provide evidence that ISG65 not only plays a crucial role in the inhibition of C3, but also interferes with phagocytosis and the activation of immune cells, thus bridging the gap to adaptive immunity”.

Authors perform a single assay, hemolysis, to show that ISG65 interfere with complement. This demonstrates that ISG interfere with either the C5 convertase or assembly of the MAC complex. It does not show that ISG65 interfere with C3 cleavage and C3b deposition by the alternative pathway (AP) or the classical pathway (CP) C3 convertase. It is throughout assumed that it is the AP that is inhibited, but this is not demonstrated in any way, e.g. with serum depleted for either of the components C4, C2 (CP) or C3, FB, FD, FP (AP). C3b plays a role in both C5 convertases beyond being the Bb binding subunit in the AP C5 convertase, the effect of ISG65 on hemolysis could be through interfering with the role of C3b in the C5 convertases.

In the discussion it is stated “While the precise mechanism by which C3 convertase interacts with C3 remains unknown, Rooijackers et al., 200933 proposed a plausible structural model for this event which is incompatible with our structure of the ISG65:C3 complex as it likely presents itself on the cell surface.”

It is not true that it remains unknown how the C3 convertase recognizes the C3 substrate. Recent reviews on structural studies of the complement system may be inspected to get an improved understanding of C3 convertase models. In combination with the alphafold2 model of ISG65 available at uniprot and the figures in the manuscript, it is straightforward to predict the following:

1) ISG65 binding to C3b does not prevent binding of substrate C3 to the AP C3 convertase where C3b is the C3 binding subunit. ISG65 may prevent assembly of the AP proconvertase C3b-FB, but that needs to be experimentally tested.

2) Whether ISG65 binding to native C3 prevents binding to CP or AP C3 convertase appears to depend completely on how the C3 binding part of ISG65 is oriented relative to the cell membrane and the VSG coat. There is no strong experimental evidence in the manuscript for how the C3/C3b binding part of ISG65 is oriented and accessible on the parasite, the SAXS data is at best indicative and overinterpreted. Interaction of trypanosome ISG65 bound C3 and C3b with their known interaction partners are crucial to make a more qualified model of how accessible C3 and C3b are when bound to ISG65 on the parasite.

The prediction from the established C3 convertase model is that if only the fragment used for structural studies here, ISG65 may in fact not at all interfere with C3 cleavage by either of the two C3 convertases. This should be also tested experimentally in standard C3b deposition assay in vitro.

Authors do show that ISG65 interfere with FH binding and therefore possibly prevents C3b→iC3b conversion, but this is not confirmed with a simple C3b/FI/FH degradation assay. This is actual crucially for the understanding of the biological consequences since iC3b is the receptor for CR3 and CR4 on phagocytes.

If FI degradation is prevented by ISG65, and the C3b is active in forming an AP C3 convertase on the parasite also, rather than inhibiting C3 cleavage, C3b could actually be accumulated on the parasite! Apparently the C3b is taken up by the parasite, but a clear model for why it is advantageous for the parasite to express ISG65 and bind C3 and its degradation products is completely lacking and many fundamental experiments regarding how ISG65 interfere with complement are required to fully understand the implications of the finding that ISG65 interacts with both C3 and C3b

Considering the binding constants of ISG65 for the various functional states and the in vivo concentrations of fluid phase C3, it is very likely that accessible ISG65 will be saturated with C3. The in vivo fluid phase concentration of C3water and other degradation fragments are very low, see below. The implication of this need to be discussed carefully.

Minor issues

Line 35-38. The AP of complement activation serves as a first line of defence against invading pathogens

This is a very simplified view of complement. The AP is mainly an amplification pathway after initiation in the CP and LP. There are ongoing discussions in the complement field how relevant initiation through AP by C3water is in vivo. The CP and LP should be introduced. There are multiple

recent reviews on the complement cascade that could be referenced here, even reviews w focus on AP. The text regarding the terminal pathway is confusing and far too simplistic. A figure is needed to describe the multiple different states of C3 (C3, C3b, C3water, C3dg, iC3b) that are encountered in the Results.

Line 140-141. "The high affinity for C3(H₂O), the initiator of the AP, is likely to be a response to its low abundance in serum." Unclear what is meant here. Please relate to PMID 31019515 for concentrations of C3water also in discussion.

Line 186-190. ". Should this interaction occur in a physiological context, the second attachment point could lock the TED domain in a conformation that further restricts its free movement. This lock would thereby also prevent the movement of the hydrolysed thioester bond of Gln1013 towards the membrane."

Is there prior data showing that the C3b is NOT covalently linked to the parasite membrane or any other molecular structure on the surface of the parasite ? As written here, it sounds like all C3b bound to ISG65 is noncovalently associated through ISG65. The role of the thioester in nascent C3b needs to be clarified, if C3 is turned over at the parasite membrane by C3 convertase, it could well become covalently linked through the thioester

Line 194. How different was the experimental ISG65 structure from the alphafold2 structure now at uniprot? Were the disulfides correctly predicted by AF?

Line 227. How should the reader know what the top of the ANA is ? Mention residues or use N-term/C-term of the four helices in ANA to guide the reader. Refer to Fig 3A here.

Line 248-260. Suppl table 3 and the SPR data in figure 2 should be redone. Why are the C3d profiles so different from WT for L1109A, E1110A and likewise some of the ISG mutants? Also, if steady state was used for K_d calculation, please show all the curves and the plot from which the K_d is estimated. An orthogonal biophysical interaction technique might be considered. Mass photometry might be a better technique in this case. A simple pulldown comparing WT and mutants may also be informative.

Line 262. Why was Arg1134 mutated if not a contact residues in the structure ? No rationale is given

Line 281. C3a is a very charged protein and the SPR experiment may not reflect the interaction that maybe is present in the structure, although the evidence is very limited for this in the figures. A control experiment is needed. Mutate ISG65 or C3a at the putative interacting residues or use C5a as control. Otherwise, remove the paragraph

Line 309. It unclear from Fig 3B what is ligand and what is analyte in the FH experiment.

Line 320. This argument is unclear. Why can C3b perhaps bind while native C3 cannot bind?

Line 464-470. Purification of the proteins need to be much more detailed described. The reader should be able to reproduce what is done

Line 478. If C3 is purified from serum it may contain significant amount of C3water. Cation exchange chromatograms documenting that the native C3 used for functional studies is really native C3 should be presented.

Line 491. Why use the recombinant FH fragment and not full FH commercially available?

Figures and tables

Nowhere is close-up of functionally interesting residues shown together with the experimental cryo-EM map. Examples from both the C3 and C3b complexes should be provided.

An overlay of ISG65-TED from BOTH structures is needed, most likely the interface is similar but no information is given regarding this.

The manuscript needs a single figure with ISG65 sequence, secondary structure with names, disulfide bridges, contacts to C3/C3b etc. As it is now, this information is spread over multiple poorly designed suppl figures. This figure should be in the main text.

Suppl table 2 needs formatting. Two of the hydrogen bonds in the table are VERY short. It is worth inspecting the map at these hydrogen bonds, perhaps something needs to be adjusted or the weight on nonbonded interactions raised substantially during refinement?

Suppl fig 1. Is the gel from reducing or non-reducing conditions? Mw of marker proteins should be added. The C3 peptides, were they from the beta-, alpha- or both chains? Were peptides in ANA present?

Supp Fig 3. Needs to be better explained for the non-experts and results summarized in table or plots

Supp table 1. Standard deviations on bonds is very large. No information is given regarding correlation between map and model anywhere for the two structures, this is needed. A correlation-by-residue plot would be useful, especially if it was indicated on such a plot where interacting residues were present.

Supp 7+8+9 should be merged to a single figure.

Supp table 3. It is unclear why different models for fitting the data are chosen for fitting the data for the various data. This must be explained if the SPR data is kept

Reviewer #2 (Remarks to the Author):

Trypanosoma brucei gambiense are exclusively extracellular in their mammalian hosts and are therefore subject to a constant onslaught of both innate and adaptive immune responses. As a result they have developed multifaceted strategies of evasion and subversion in order to survive.

In this manuscript the authors identify the *Trypanosoma brucei gambiense* surface protein ISG65 to be a receptor of the human complement factor C3 and present data in support of this binding interfering with the killing of the parasite by the alternative pathway. They use an extensive array of different molecular and structural biology techniques to identify and verify binding and to

define the binding interface. In particular, they present the cryo-EM structures of ISG65 in complex with C3 and C3b and use SAXS and AlphaFold modeling to augment the structure models. Mutagenesis analysis was employed to confirm involvement of specific residues in the binding interface. SPR experiments show binding of different C3 fragments to ISG65 suggesting the surface protein interferes at various stages of the complement cascade. Interestingly the authors also find by competition SPR experiments that binding of factor H to C3b can be abrogated by ISG65. This work reveals another layer to the multipronged evasion strategies used by this African trypanosome.

Minor points

- 1) For the reader to fully appreciate the extent of known immune evasion strategies used by African trypanosomes it might be helpful to add a few keywords to at least give a broad overview of these mechanisms (cited in references 2-11). A few other papers might warrant inclusion here (Macleod et al., Nat. Commun. 2020 – FHR receptor; Pinger et al., Nat. Microbiol. 2018 – immune evasion by O-glycosylation of the VSG)
- 2) In Fig. 1D the arrows labelled with ISG65/ISG75 concentration seem confusing. Might it not be better to add a legend to clearly state the ISG concentration corresponding to each curve? Alternatively add this information in the caption.
- 3) In line 109 you reference the FACS data with Supplementary Fig. 2 this data is shown in Supplementary Fig. 3 though. The fluorescence microscopy imaging shown in Supplementary Fig. 2 appears not to be mentioned in the text.
- 4) You appear to mean excess not access in line 559
- 5) In the 'C3b surface binding and uptake analysis' paragraph in the supplementary Data section several volumes are stated as ml, but should surely be μl .

Reviewer #3 (Remarks to the Author):

This paper describes primarily the structure determinations of ISG65 in complex with C3 and C3b. By itself these are relevant data. However, the paper falls short in respect of functional data that demonstrates what is bound to the trypanosomal surface and what processes are at play on the surface. Since the primary data is only structural, the discussion should be phrased along the lines of possible or plausible structural interpretations and avoid attributing function to molecules.

The paper describes two protein complexes of ISG65 bound to C3 and C3b. It is unclear what the relative importance of either complex in the defense of the trypanosome against complement activation. At what stage complement activation is efficiently inhibited is unclear.

Overall, the writing style is one characterized by bold statements or overstatements and is rather non-academic in my opinion. At least for me this manuscript is very hard to read. As an example, the first paragraph consists of a set of bold statements of which only the last sentence was informative.

Line 61-64, "We demonstrate that ISG65 interferes with the release of anaphylatoxin C3a, as well as with the binding of fH to C3b, thereby inhibiting iC3b formation and consequently halting downstream opsonisation and phagocytosis of *T.b. gambiense*." These claims are not substantiated by the data. Neither C3a release nor effects on phagocytosis are directly experimentally shown.

Line 76, cryo-EM applied to complement proteins is not new, as suggested by the authors. A great example is the structure of the MAC pore by Bubeck and co-workers.

Line 145, the work by Chen et al. and Rappsilber *Mol Cell Proteomics* 2016, 15, 2730-2743 revealed the structural features of C3(H2O).

Line 157, the phrase 'near-atomic range' or 'near-atomic resolution' is misleading for 3.5 Ang data.

Line 358, the authors claim to have presented the function of ISG65. However, no knock-out data is available, thus the claim is unsubstantiated. The word 'function' needs to be replaced by 'role'.

Line 361, 'manipulates innate immunity at several critical intervention points using a multipronged strategy to avoid complement mediated clearance' is unsubstantiated. The paper lacks functional assays to support this statement. Earlier papers (refs 14 and 15) state that complement activation does not progress beyond formation of the C3 convertase (line 52/53). It is not clear what fragments occur at the surface of trypanosomes and what defense mechanisms are at action. It is not shown that factor H binds to trypanosomes.

Line 364, 'master function of ISG65-mediated complement inhibition is to keep the TED away from the plasma membrane', no evidence is provided for this statement. Does this statement refer to a

role in cellular immune response? Overall, the paper lacks functional data to statements with respect to biological function, let alone 'master function'.

Line 367-369, it is a (common) misunderstanding that C3(H₂O) is a biological active species of complement activation on a surface. The AP initiator is most likely C3, when loosely associated with a surface, that may adopt the active-like conformation of C3b or (what is referred to as) C3(H₂O), i.e. proteolytically uncleaved C3. No functional data given that indeed C3(H₂O) is bound and thereby stops AP initiation altogether.

Line 370-371, 'Due to the complexity of 369 the complement system and the sheer abundance of complement in the blood stream, a single mechanism is less likely to be successful than a concerted attack. Therefore ...' This statement and subsequent logic are unsubstantiated.

Line 373, 'Recruitment of native C3 to the cell surface has several consequences', this is an overstatement.

Reviewer #4 (Remarks to the Author):

The authors here present a novel characterization of the structure and binding conformers of ISG65 from *T. b. gambiense* using a variety of biophysical techniques. Structural elucidation is essential for a clear understanding of protein function, and this manuscript makes an important contribution to the mechanisms of this host-pathogen interaction and potential immune escape. The conclusions are reasonable and appear well-supported by the data.

The bulk of the findings rely on cryo-EM, SPR, SAXS, and alanine scanning data, and mass spectrometry (MS) plays a minor role in establishing the conclusions drawn. However, if these MS experiments remain in the manuscript, revisions and additional details are very much needed for clarity and reproducibility.

Overall, I recommend that the authors revise and refine the MS portions of the manuscript and the related figures for re-evaluation before publication. I also recommend minor clarifications to 3 non-MS related sections of the text (placed at the end of the comments).

Specific comments are listed below.

Gelband experiment - Main text lines 85-90 / Supplementary Figure 1: As presented, the experimental logic is that, of any proteins identified in the gelband, the highest scoring protein would be the specific enrichment target for ISG65. This is an inaccurate interpretation of protein scores in MS search algorithms. The score only reflects the quality and certainty of the identification, not the likelihood of this protein being most relevant to the experimental question. It is also unclear as presented whether gelbands were excised from both the control and the enriched condition and compared (as inferred from main text line 89), or whether the comparison was done visually from the gel and only the enriched condition was analyzed by MS. In order to show that C3 was uniquely enriched by ISG65, authors should have analyzed both the negative control serum and the ISG65-enriched serum and shown either that C3 appears uniquely in the enrichment condition, or that C3 appears at much greater abundance in the enrichment condition, and if this was done, the figure should reflect comparative data from these runs. A less-compelling but acceptable argument could be made on the basis of C3 being the most abundant protein detected in the enriched sample alone, which depends on either the number of PSMs (if using spectral counting for quantification) or peak intensity (if using area under the curve for quantification). The protein score, however, is irrelevant for proving that C3 is specifically enriched by ISG65. This figure should be changed to present supporting data as described above, and the main text revised to clarify the experiment.

Intact mass experiment - Main text lines 136-137 / Supp Fig 4: Figure is not easily interpretable as presented. The mass labels in the insets are cut off and cannot be fully read. The figure legend states that the highest intensity peak in panel A is 187,632 but the peak is labeled as 187,637.22 and a 5-Da difference is substantial – was this a typo or is 187,632 Da the expected theoretical mass? Authors should report the theoretical mass of each species, the measured mass, and ideally the mass accuracy in ppm. Masses should be reported to 2 or 3 decimal places, not rounded to the nearest integer, as the reported instruments are high-res accurate mass and capable of greater than unit resolution. The legend should mention what software is being used to generate the data.

Since the goal of this experiment is to confirm the identity of the C3 species used in the SPR experiment, presenting only intact mass measurements requires that those measurements be highly accurate and precise (which cannot be determined as presented). If any fragmentation data were generated, metrics such as the number of unique peptides, sequence coverage, and protein score could be included as in Supp Fig 1 to provide confidence in the identification. Further, since contaminating species would lower the confidence of the SPR experiments, the authors could further strengthen this figure by commenting on the relative purity of their target constructs in these MS runs, which has a significantly greater sensitivity than SDS-PAGE. I recommend that the authors consult with the mass spectrometrists associated with this study to generate clearer figures demonstrating an accurate Mw determination.

Disulfide mapping experiment - Main text lines 204-206 / Supp Fig 8: Cystamine can be used to promote the formation of disulfide bonds, and this experiment was conducted after an overnight incubation with excess cystamine. Were the authors using this reagent to map the general proximity of various cysteines to refine the cryo-EM model, or were they trying to locate native disulfide bonds? If the latter, what evidence do they have that the linkages observed are not artifacts of the sample prep? If the former, a clarifying comment on the experimental goal and the methods used, either in the main text or the methods, would be helpful.

In Supp Fig 8, panel A shows extracted ion chromatograms for three crosslinked peptides. The standard in the field for presenting peptide identifications is an annotated MS2 spectra, rather than an EIC of the MS1 mass, since intact mass alone cannot determine sequence information, and the authors should change the figure to provide this evidence instead. In addition, the peptide map of the unreduced protein form in panel B indicates that for each of the crosslinked pairs, one of the peptides is detectable in an uncrosslinked form. If this is true, the figure would be improved by including the relative quantification of the linked and unlinked peptide forms to obtain an estimate of the disulfide stoichiometry. Stoichiometry information is important for cryo-EM model refinement because if only 5% of the target peptide population was detected in a crosslinked form, then constraining the structural model to a conformation that allows that link to exist would not represent the majority conformer.

The third peptide in panel A is annotated as having a glycosylation event detected “by manual data inspection” but no MS evidence of this modification is provided. Modified peptides are typically a very low proportion of the overall population of that peptide (<10%). Were the authors able to find the unmodified form of this crosslinked peptide, or was the fucosylated paucimannosylated form the only identification possible? If the latter, this is very likely a false identification. If the former, the unmodified form should be presented instead, and any PTMs referenced in the manuscript should be evidenced by annotated MS2 spectra.

HDX experiment - Main text lines 264-268 / Supp data lines 25-34 and Supp Fig 11: Insufficient detail is provided for the actual exchange reaction, including reaction buffer components, pH, and temperature. The authors should comment on their choice to include reducing agent in the quench buffer if this was not present in the exchange reaction, as this would affect the conformation of the protein sample and create the potential for artifactual in-exchange. The authors should include basic MS acquisition method parameters. Data processing methods should be expanded to include details of any manual validation, filtering choices, and back-exchange corrections if applied or reasoning if not. Figure legend should include an explicit description of what the color scale in panel C is showing (is this the percent difference between free and complexed protein? Does red indicate an increase in exchange or protection? Does red indicate the free compared to the complex, or the complex compared to the free? Needs to be much more clear).

The authors mention only one residue specifically from this experiment, Arg1134, but multiple regions look to have equal amounts of protection. Can the authors comment on the other regions of protection? Did the authors have overlapping peptides covering residue 1134 to narrow down this residue as being specifically protected? (This information should be available in the peptide map in panel B but the residue numbers are not adjusted to match the numbering in panel C.) If not, conclusions have to remain at the peptide level, and should not be extrapolated to single residue level. Additionally, protection in C3d appears to be strongest at 20 minutes and largely insignificant at shorter or longer timepoints – does this correlate with the authors' understanding of the affinity and residence time of the binding interaction? The peptide containing Y293 in ISG65 shows the strongest protection at a different timepoint (2 min) than its proposed binding interface residues 1109-1119 in C3d (20 min) – can the authors comment on this?

I suggest the authors include uptake plots for all regions affected in the timescale plot, and include a structure in this figure to highlight where in the 3D conformer protection occurs, particularly as those sites relate to the binding interface residues detailed in the main text lines 227-234.

General notes on supplemental method text

Lines 97-99: If more than one run was acquired, authors should include the amount of material injected for runs (if measured) and whether injection amounts were normalized across samples.

Line 106: For reproducibility, authors need to specify the range of %B increase (5-35%? 5-40%?) and specific time period, as the 65 minutes described could refer to either the entire gradient including wash and re-equilibration steps, or just the active elution portion of the gradient.

Lines 106-108: Authors should note basic MS acquisition parameters including resolution of MS1 and MS2 scans, AGC settings, maximum injection times, whether the fragmentation data was obtained as data-dependent or data-independent acquisition, collision type and detector used for MS2 scans, loop count settings, and dynamic exclusion settings.

Line 110: Search algorithm is "MS Amanda", not just "Amanda" and algorithms should be cited wherever possible. Authors should specify what databases were used for Homo sapiens and common contaminants (likely UniProt? In which case, needs to include a UniProt reference identifier and the access date.) Authors should also note whether they included the ISG65 sequence in the database, as peptides in the sample mapping to this protein will be incorrectly assigned to a human protein if its own sequence is not available to search. Authors need to state general search settings

for data, including mass tolerances for both MS1 and MS2 scans, variable and fixed modifications, number of missed cleavages allowed, and false discovery rate thresholds.

Non-MS-related comments

Main text lines 92-101 / Fig 1A: Since the authors use Western blotting for verification later in the text, the SDS-PAGE analysis would be strongly improved by adding a Western blot here, as SDS-PAGE does not provide protein identity. Also recommend editing the figure to move the gel inset away from the y-axes of the chromatogram, as currently the units of the graph can be misread as the Mw markers for the gel, and adding Mw markers to the gel specifically.

Fig 1B: The authors interpret this blot to read that the ISG65:C3b complex exists in a 1:1 stoichiometry, but it is unclear how this conclusion was derived (band densitometry in ImageJ? By eye?). Since the methods state that ISG65 was added at 2x excess, the free ISG65 band should be of equal intensity with the complex band, but it appears substoichiometric from the image. More detail on this calculation would be helpful. If a Western blot of the sample is available, this would also help strengthen the figure.

Main text lines 106-109 / Supp Fig 2: What is the n of this experiment? Should be included in the legend.

REVIEWER COMMENTS

Reviewer #1 (Remarks to the Author):

The manuscript establishes ISG65 as a complement C3 binding protein. The primary binding site of ISG65 is shown to be on the thioester domain of C3, which enables its binding to C3, C3water, C3b and the C3d fragment. Two cryo-EM structures are presented that convincingly establishes the binding site of ISG65 on C3 and C3b. The structural work is solid although minor issues need to be addressed. In contrast, the functional characterization of how ISG65 interfere with complement activation and the downstream effector functions is incomplete and the description of complement biology is too simple minded.

Major issues:

Abstract “we provide evidence that ISG65 not only plays a crucial role in the inhibition of C3, but also interferes with phagocytosis and the activation of immune cells, thus bridging the gap to adaptive immunity”.

Authors perform a single assay, hemolysis, to show that ISG65 interfere with complement. This demonstrates that ISG interfere with either the C5 convertase or assembly of the MAC complex. It does not show that ISG65 interfere with C3 cleavage and C3b deposition by the alternative pathway (AP) or the classical pathway (CP) C3 convertase. It is throughout assumed that it is the AP that is inhibited, but this is not demonstrated in any way, e.g. with serum depleted for either of the components C4, C2 (CP) or C3, FB, FD, FP (AP). C3b plays a role in both C5 convertases beyond being the Bb binding subunit in the AP C5 convertase, the effect of ISG65 on hemolysis could be through interfering with the role of C3b in the C5 convertases.

We did indeed not investigate a role of ISG65 in the classical pathway because it was already shown convincingly that Trypanosoma brucei gambiense does not activate the classical pathway, but only the alternative pathway (Devine et. al, <https://doi.org/10.1128/iai.52.1.223-229.1986>). We apologise for not making this point sufficiently clear.

In this study no immunoglobulin was detected on the surface of T. b. gambiense which is prerequisite for activation of the classical pathway. C3 deposition was enhanced by fB, fD and magnesium. Further, fB could be detected on the cell surface. Finally, C3 deposition was inhibited by EDTA (which inhibits all pathways), but not EGTA (which inhibits the CP but not the AP by depleting free Calcium ions).

We have now clearly emphasised the findings of this study as they provided the basis for the rationale driving our own experiments. In line with this rationale, we used a hemolysis assay in which only the AP is activated due to the usage of non-sensitised rabbit erythrocytes (<https://www.haemoscan.com/products/ap50/>). In an earlier version of this manuscript, we falsely mentioned sheep erythrocytes (while still referencing the correct kit). We have rectified the issue and apologise for this mistake.

As we believe the cited study by Devine et al. to be sound and comprehensive, we did not attempt to recreate it. Particularly in absence of a T. b. gambiense knock-out strain this would have not resulted in additional insights.

However, the reviewer is right that in T. b. gambiense the (AP) C5 convertase is the likely point of intervention within the complement cascade. In the mentioned study by Devine et al. it was further shown that neither C5 nor C5b-9 could be detected on the surface of T. b. gambiense. This is further corroborated by the fact that C5 knockout (KO) mice show a normal parasitaemia progression when infected with different Trypanosoma strains (Jones & Hancock, [10.1128/iai.42.2.848-851.1983](https://doi.org/10.1128/iai.42.2.848-851.1983); Jarvinen & Dalmasso, [10.1128/iai.16.2.557-563.1977](https://doi.org/10.1128/iai.16.2.557-563.1977); Tabel et al., [10.1016/s1286-4579\(00\)01318-6](https://doi.org/10.1016/s1286-4579(00)01318-6); Greca et al., [10.1111/pim.12106](https://doi.org/10.1111/pim.12106)). If C5 is truly never activated in wild-type mice, its absence in KO mice would be of no consequence.

In the discussion it is stated “While the precise mechanism by which C3 convertase interacts with C3 remains unknown, Rooijackers et al., 2009³³ proposed a plausible structural model for this event which is incompatible with our structure of the ISG65:C3 complex as it likely presents itself on the cell surface.”

It is not true that it remains unknown how the C3 convertase recognizes the C3 substrate.

We apologise for the overstatement. What we meant is that there is no experimental structural data on the actual C3bBb:C3 enzyme-substrate complex. A model is available which relies on superimposition of C3 with C3b within an inhibitor-mediated dimer of the crystal structure of C3bBb (PDB: 2WIN; Rooijackers et al., [10.1038/nr1756](https://doi.org/10.1038/nr1756)).

The respective sentence has been changed to: “Moreover, binding of native C3 to ISG65 on the parasite surface which is likely presented in a VSG-like orientation with its long axis perpendicular to the membrane could sterically hinder the interaction of C3 substrate with C3bBb as proposed earlier (Rooijackers et al., [10.1038/nr1756](https://doi.org/10.1038/nr1756)) (Supplementary Fig. 13).”

Recent reviews on structural studies of the complement system may be inspected to get an improved understanding of C3 convertase models. In combination with the alphafold2 model of ISG65 available at uniprot and the figures in the manuscript, it is straightforward to predict the following:

1) ISG65 binding to C3b does not prevent binding of substrate C3 to the AP C3 convertase where C3b is the C3 binding subunit.

We agree with the reviewer. ISG65 does clearly not prevent this assembly (Supplementary Fig. 13c). We have added a comprehensive supplementary figure (Supplementary Fig. 13) in which we superimpose C3b-bound ISG65 with the known structures of C3b in complexes with its binding partners. This figure illustrates which C3(b) assemblies are structurally possible in presence of ISG65.

ISG65 may prevent assembly of the AP proconvertase C3b-FB, but that needs to be experimentally tested.

*The structure of C3b:fB (PDB: 2XWJ) when superimposed with C3b:ISG65 suggests that ISG65 does indeed not prevent assembly of the AP proconvertase. ISG65 does not produce any clashes with fB (Supplementary Figure 13b). Furthermore, the data by Devine et al. (<https://doi.org/10.1128/iai.52.1.223-229.1986>) shows that the C3 convertase is formed on the surface of *T. b. gambiense*.*

As suggested by the reviewer, we also tested C3 proconvertase formation in presence and absence of ISG65 experimentally by SPR and included this information in the supplementary data (Supplementary Fig. 14) as well as below. Binding of fB to immobilised C3b causes an identical increase in response when compared to binding to C3b, previously saturated with ISG65. This indicates that binding of fB is unimpaired in presence of ISG65 and that the AP proconvertase assembly is not inhibited.

Supplementary figure 14: AP pro-convertase formation in presence of ISG65. SPR sensorgrams showing the AP proconvertase formation in presence of ISG65. C3b was immobilised onto the chip and ISG65 and factor B subsequently injected (solid black). Control injections were performed with ISG65 (black dashes) or factor B (solid grey) only. Beginning and end of the injections are marked with arrows (ISG65: blue, factor B: red).

2) Whether ISG65 binding to native C3 prevents binding to CP or AP C3 convertase appears to depend completely on how the C3 binding part of ISG65 is oriented relative to the cell membrane and the VSG coat. There is no strong experimental evidence in the manuscript for how the C3/C3b binding part of ISG65 is oriented and accessible on the parasite, the SAXS data is at best indicative and overinterpreted. Interaction of trypanosome ISG65 bound C3 and C3b with their known interaction partners are crucial to make a more qualified model of how accessible C3 and C3b are when bound to ISG65 on the parasite.

We agree that a clarifying figure is needed to understand the interaction of C3- and C3b-bound ISG65 with the various interaction partners of the complement factor. We now provide this overview in supplementary figure 13. Our SAXS model must be viewed in context of the current model of the organization of the VSG layer. In this model VSGs, as well as interspersed receptors, are oriented upright with their head domains being located at the membrane-distal tip of the 3-helix bundle and thus pointing away from the membrane (Mehlert et al, <https://doi.org/10.1371/journal.ppat.1002618>; Bartossek et al., [10.1038/s41564-017-0013-6](https://doi.org/10.1038/s41564-017-0013-6); Schwede et al., [10.1371/journal.ppat.1005259](https://doi.org/10.1371/journal.ppat.1005259)). For ISG65 to allow C3 substrate binding to the C3 convertase (C3bBb) as proposed in the model by Rooijackers et al., [10.1038/ni.1756](https://doi.org/10.1038/ni.1756), the C3 binding domain of ISG65 would have to be oriented parallel to the plasma membrane, necessitating a 90° kink in the C-terminal linker. While theoretically possible, under physiological conditions it is highly unlikely for several reasons:

- i) *Head domains contain complex, variable loop structures and in order to either act as immune decoys (VSGs; Hsia et al. <https://doi.org/10.1046/j.1365-2958.1996.351878.x>) or for nutrient acquisition (Transferrin receptor, Tfr; Trevor et al. [10.1038/s41564-019-0589-0](https://doi.org/10.1038/s41564-019-0589-0)) they have to be accessible to the environment.*
- ii) *VSG homodimers that constitute the surface coat are packed at very high density to exert their shielding function (Manna et al., [10.1016/j.pt.2014.03.004](https://doi.org/10.1016/j.pt.2014.03.004)). An orientation of ISG65 parallel to the membrane, occupying a larger area, would therefore be disfavoured.*
- iii) *Even if ISG65-bound (substrate) C3 would be in an orientation to interact with membrane-bound C3 convertase, it is most likely that this complex cannot penetrate the VSG layer due to its size and the lack of conformational flexibility of the TED in C3 (see question: Line 320. This argument is unclear. Why can C3b perhaps bind while native C3 cannot bind?)*

This argumentation is provided also in a condensed form in the discussion as well as the figure legend of supplementary figure 13.

The prediction from the established C3 convertase model is that if only the fragment used for structural studies here, ISG65 may in fact not at all interfere with C3 cleavage by either of the two C3 convertases. This should be also tested experimentally in standard C3b deposition assay in vitro.

We agree with the reviewer's conclusion. Structural data strongly suggests that binding of fluid phase ISG65 (outside the context of the *Trypanosoma* VSG coat) to C3 substrate would neither interfere with binding of C3 substrate to C3bBb nor with its cleavage (Supplementary Fig. 13e). The ISG65 binding site on native C3 is far away from the scissile bond that is accessed by Bb and that leads to liberation of C3a and conversion of substrate C3.

As suggested, we also we confirmed this experimentally but with a C3a ELISA instead. Similarly to C3b release, C3a release indicates cleavage of the C3 substrate.

C3a enzyme-linked immunosorbent assay. Purified C3 substrate, alone or in complex with ISG65 was added to surface-immobilised, nickel-stabilised AP C3 convertase (using 2 different concentrations of C3b for convertase assembly). After completion of the reaction, C3a was detected in the supernatant using a monoclonal HRP-conjugated Anti-C3a antibody. No differences in C3 substrate conversion could be detected between free C3 and ISG65-bound C3 substrate. This confirmed that fluid-phase ISG65 does not prevent C3 cleavage by C3bBb.

We have emphasised this in the discussion: “However, neither does ISG65 binding to C3b and native C3 prevent assembly of the C3 convertase (C3bBb) and the enzyme-substrate complex (C3bBb:C3) (Supplementary Fig. 13 and 14) nor does binding to C3 substrate prevent its cleavage by C3bBb (Supplementary Fig. 13)”

Authors do show that ISG65 interfere with FH binding and therefore possibly prevents C3b->iC3b conversion, but this is not confirmed with a simple C3b/FI/FH degradation assay. This is actual crucially for the understanding of the biological consequences since iC3b is the receptor for CR3 and CR4 on phagocytes.

As suggested by the reviewer we conducted basic C3b/FI/FH degradation assays in absence and presence of ISG65 and visualised the formation of degradation fragments of the alpha chain by SDS-PAGE and Coomassie staining (Sahu and Lambris <https://doi.org/10.1034/j.1600-065x.2001.1800103.x>). These degradation assays were performed in solution. As can be seen in the gels below no differences in cleavage rate could be observed during a time frame of 6 hours at 37°C.

iC3b degradation assay. The degradation assay was conducted over a time period of 6 hours at 37°C solution using purified proteins (fH, fI, C3b and ISG65). ISG65 was added to C3b prior to addition of other components in 3-fold molar excess. Degradation products were visualized on an Coomassie-stained 8% SDS-gel.

Although ISG65 blocks the fH CCP19-20 binding site on C3b-TED, it does not abrogate fI activity and consequently iC3b formation in the fluid phase. This, however, does not rule out a role of ISG65 in inhibiting co-factor activity of fH in cleavage of cell surface bound C3b. In addition to binding to surface attached C3b-TED, CCP 19-20 and 7 of fH have also been shown to interact with carbohydrates on the cell surface (Loeven et al., [10.3389/fimmu.2021.676662](https://doi.org/10.3389/fimmu.2021.676662)), a step which is thought to be crucial for correct positioning of the co-factor bearing domains CCP1-4 onto C3b for subsequent cleavage by fI. Indeed, it has been shown that competition of recombinant CCP19-20 with full-length fH can inhibit iC3b formation on cell surfaces but not in solution (Ferreira et al., <https://doi.org/10.4049/jimmunol.177.9.6308>). In a similar manner this could be the case for ISG65 as well.

We meant to test the formation of *iC3b* in a realistic *in vivo* setting on the parasite cell surface by comparing the situation in the wild-type to the one in a knock-out strain. While we were pursuing the generating of a knock-line in all 3 ISG65 loci, we could not achieve a complete knock-out during the time of the revision process. The investigation of *iC3b* formation on the parasite surface in presence and absence of ISG65 will therefore be part of a separate study. A figure illustrating the presence of residual ISG65 copies in a CRISPR-Cas knock-out strain is shown below.

Whole Genome Sequencing to assess new ISG65 locus architecture. ISG65_6 and ISG65_9 refer to different clones.

If FI degradation is prevented by ISG65, and the C3b is active in forming an AP C3 convertase on the parasite also, rather than inhibiting C3 cleavage, C3b could actually be accumulated on the parasite! Apparently the C3b is taken up by the parasite, but a clear model for why it is advantageous for the parasite to express ISG65 and bind C3 and its degradation products is completely lacking and many fundamental experiments regarding how ISG65 interfere with complement are required to fully understand the implications of the finding that ISG65 interacts with both C3 and C3b.

We admit that here we only provide a first glimpse into a very complex mechanism that might take years of research to be fully understood. It is true that C3b could accumulate on the cell surface to a certain extent, but it is constantly removed by the shear forces, generated by vigorous swimming of the parasite, that act on surface-bound proteins. This process has been described for VSGs (Engstler et al., [10.1016/j.cell.2007.08.046](https://doi.org/10.1016/j.cell.2007.08.046)), the only other surface molecule that is found distributed across the entire surface in a manner similar to ISGs (Ziegelbauer and Overath, [10.1128/jai.61.11.4540-4545.1993](https://doi.org/10.1128/jai.61.11.4540-4545.1993)). Others surface receptors described so far are localized exclusively to the flagellar pocket (Field and Carrington, <https://doi.org/10.1038/nrmicro2221>).

*The advantage for the parasite to have a C3(b)-binding molecule on the surface is to have a specific, high affinity attachment point for a molecule which subsequently delivers its cargo to the 'sink hole' of the cell, the flagellar pocket, the only place for endo- and exocytosis for subsequent degradation in the lysosome. The high recycling rate that has been described for ISG65 (Koumandou et al., [10.1128/EC.00273-12](https://doi.org/10.1128/EC.00273-12)) would hereby serve to remove C3b with extremely high efficiency, possibly preventing a critical density of C3b on the surface that has been associated with C5 convertase formation (Zwarthoff et al., [10.3389/fimmu.2018.01691](https://doi.org/10.3389/fimmu.2018.01691)). We show surface removal of fluorescently labelled C3b by our improved uptake experiments in *T. b. gambiense* (Supplementary Figure 2).*

The main purpose of VSGs is to create immune decoys but their other purpose is to also remove toxic molecules such as antibodies from the surface. This is an immune defence mechanism in itself! Similarly, one purpose of ISG65, maybe even the primary one, could be to remove complement from the surface, a function that is not associated with VSGs.

*However, since the extracellular domain of ISG65 is also able to abrogate AP activity *in vitro*, we do speculate that there might be another mechanism of complement cascade inhibition associated with ISG65 binding to C3 and that such a mechanism could serve to prevent complement activation *before* it can be ultimately removed from the surface. As suggested by the reviewer, ISG65 could indeed interfere with assembly of the C5 convertase either through binding to C3b or C3 substrate that is converted to the second C3b of (C3b)₂Bb. Unfortunately, no structure of the C5 convertase is currently available which precludes drawing any conclusions about its assembly in presence of ISG65.*

In an attempt to answer this question, we tried to assemble the C5 convertase on an SPR chip in presence of ISG65. This attempt however failed due to constant dissociation of already assembled components during the injection of further components. There is a gap time between injection of 2 proteins while buffer is still continuously flown over the chip. This a technical limitation of SPR. BLI (biolayer interference), a technique similar to SPR, might be more suited to answer this question as it does not use a constant flow of buffer over the surface onto which proteins are immobilized. Therefore, it is less prone to unwanted dissociation effects during the assembly of multi-component complexes. Due to time constraints, we could however not establish such a setup during the revision period.

We have changed the discussion of our manuscript in a fashion similar to the formulation above.

Considering the binding constants of ISG65 for the various functional states and the *in vivo* concentrations of fluid phase C3, it is very likely that accessible ISG65 will be saturated with C3. The *in vivo* fluid phase concentration of C3water and other degradation fragments are very low, see below. The implication of this need to be discussed carefully.

*It has been shown that the binding capacity of the surface of *T. b. gambiense* for C3 is indeed 3-4 times lower than for C3b (Devine et al., <https://doi.org/10.1128/iai.52.1.223-229.1986>). It is true that native C3 has the highest serum concentration, but the VSG coat of *T. b. gambiense* might just be more accessible for C3b than for C3 (see question: This argument is unclear. Why can C3b perhaps bind while native C3 cannot bind?). The same could be true for C3(H₂O) which has been suggested to be related in architecture to C3b with a similarly flexible CUB/TED (Chen et al., <https://doi.org/10.1074/mcp.M115.056473>), but this was not investigated in the mentioned study by Devine et al. (<https://doi.org/10.1128/iai.52.1.223-229.1986>). Based on published values for serum concentrations of native C3, C3b and C3(H₂O) we consider the dissociation constants with ISG65, as measured, to be physiological.*

We have appended the following clarification to the results section:

“Dissociation constants for C3 and C3b are physiological given the high serum concentrations of both fragments (Liu et al., <https://doi.org/10.1007/s12288-020-01338-0>). For C3(H₂O) the exact serum concentration has been more difficult to determine, but it is substantially lower with reports ranging from 3-100 nM (Pangburn et al., <https://doi.org/10.1084/jem.154.3.856>; <https://doi.org/10.1111/j.1749-6632.1983.tb18116.x>; Elvington et al., <https://doi.org/10.3389/fimmu.2019.00703>). Thus, the affinity of ISG65 for C3(H₂O), as measured, is physiological, although ISG65 might not be fully occupied.”

Minor issues

Line 35-38. The AP of complement activation serves as a first line of defence against invading pathogens

This is a very simplified view of complement. The AP is mainly an amplification pathway after initiation in the CP and LP. There are ongoing discussions in the complement field how relevant initiation through AP by C3water is *in vivo*. The CP and LP should be introduced. There are multiple recent reviews on the complement cascade that could be referenced here, even reviews w focus on AP. The text regarding the terminal pathway is confusing and far too simplistic. A figure is needed to describe the multiple different states of C3 (C3, C3b, C3water, C3dg, iC3b) that are encountered in the Results.

*We agree that we have depicted the complexity of the complement system in a perhaps too simplistic way. We also agree with the reviewer that an additional figure summarising the AP and introducing its most important components is needed here. As suggested, we have now included such a figure in the main text (Figure 1) together with a more detailed description of the mechanism of complement activation. However, since *T. b. gambiense* is only activating the AP (Devine et al., <https://doi.org/10.1128/iai.52.1.223-229.1986>), we decided to omit the other pathways in the figure for clarity and only mention them briefly in the text.*

Line 140-141. “The high affinity for C3(H₂O), the initiator of the AP, is likely to be a response to its low abundance in serum. “ Unclear what is meant here. Please relate to PMID 31019515 for concentrations of C3water also in discussion.

*What we meant here is that the lower the concentration of a ligand in a system (e.g. C3(H₂O) in blood), the higher the binding affinity that is needed by a receptor to establish an interaction. In human blood C3 is the most abundant species, followed by C3b and C3(H₂O). At a simple level one could therefore imagine that ISG65 has specifically evolved the highest affinity for the rarest C3 fragment. *K_D* and reported serum concentration of C3(H₂O) (3 nM according to PMID 31019515, approx. 100 nM according to other sources, Pangburn et al. <https://doi.org/10.1084/jem.154.3.856>; <https://doi.org/10.1111/j.1749-6632.1983.tb18116.x>) are both in the low nanomolar range which means the measured concentration is in the physiological range. We have changed this statement in the manuscript. Please see the corresponding citation in the last response in the previous section (major issues).*

The slight discrepancy between the lowest reported serum concentration of C3(H₂O) (3 nM) and the measured affinity between C3(H₂O) and ISG65 might also be attributed to the fact that our protein is only near native as it was generated through methylamine treatment and modified with iodoacetamide (Chen et al., <https://doi.org/10.1074/mcp.M115.056473>, Pedersen et al., <https://doi.org/10.15252/emj.201696173>). A completely native protein for SPR measurements would have been desirable, but the yields for C3(H₂O) directly purified from serum are too low for these measurements and the unreacted protein with an unpaired cysteine is not stable.

Line 186-190. “. Should this interaction occur in a physiological context, the second attachment point could lock the TED domain in a conformation that further restricts its free movement. This lock would thereby also prevent the movement of the hydrolysed thioester bond of Gln1013 towards the membrane.”

Is there prior data showing that the C3b is NOT covalently linked to the parasite membrane or any other molecular structure on the surface of the parasite? As written here, it sounds like all C3b bound to ISG65 is noncovalently associated through ISG65. The role of the thioester in nascent C3b needs to be clarified, if C3 is turned over at the parasite membrane by C3 convertase, it could well become covalently linked through the thioester

*We thank the reviewer for this question. To our knowledge there is indeed no data showing that C3b cannot get covalently linked to the membrane. Although the estimated number of ISG65 molecules on the cell surface is in the same range as the number of C3b molecules detected on the surface of *T. b. gambiense* (Field and Carrington, [10.1038/nrmicro2221](https://doi.org/10.1038/nrmicro2221); Devine et al., <https://doi.org/10.1128/iai.52.1.223-229.1986>), this correlation does not rule out that some C3b “escapes” the interaction with ISG65 and docks to the membrane where it could form C3 convertase. Within the complex with ISG65, however, the movement of the CUB and TED domains is likely to be restricted in comparison to free C3b. Such a conformational lock could impair an interaction of Gln1013 of the former thioester with hydroxy groups on the membrane. We changed our statement in the manuscript, making it less ‘absolute’, and marked the position of the thioester in our model of the ISG65:C3b complex in Figure 3 as well as Supplementary Figure 13.*

The text has been changed to:

“While we cannot rule out that some C3b attaches to the cell membrane and forms C3 convertases, in complex with ISG65 the observed contact point with C3b-CUB might restrict free movement of the TED and thereby impair binding of Gln1013 of the former thioester to hydroxy groups on the membrane.”

Line 194. How different was the experimental ISG65 structure from the alphafold2 structure now at uniprot? Were the disulfides correctly predicted by AF?

The experimental structure and the alphafold2 (AF2) structure are very similar. We have included a structural alignment below. The predicted disulphides are however different from the experimentally determined positions. In a recent benchmarking test (Gulsevini and Meiler, doi: <https://doi.org/10.1101/2022.02.17.480937>) AF2 has been shown to have deficits in reliably predicting disulphide bonds and flexible regions such as present in the head domain of ISG65.

Comparison of ISG65 models. Overlay of the experimental ISG65 model obtained by cryo-EM (green) and a computational model obtained via AlphaFold (grey) [Uniprot: C9ZJ67]. Predicted parts of ISG65 not present in the experimental model are displayed translucent.

Line 227. How should the reader know what the top of the ANA is? Mention residues or use N-term/C-term of the four helices in ANA to guide the reader. Refer to Fig 3A here.

We completely agree. This has been changed in the text to:

“Positioned above the loop (Gly684-Lys692) connecting the two N-terminal helices of the ANA domain (C3a, when proteolytically liberated) and the N-terminal side of the longest helix in the domain (Ala719-His738) (Fig. 3, Fig. 4A), binding of ISG65 to C3 is predominantly mediated via hydrophobic contacts and four distinct hydrogen bonds with the thioester domain. Three hydrogen bonds are mediated by Tyr293 (Tyr293_{ISG65}-Glu1110_{C3}, Tyr293_{ISG65}-Glu1111_{C3} and Tyr293_{ISG65}-Glu1119_{C3}) and one by Arg77 (Arg77_{ISG65}-Leu1109_{C3}) (Supplementary Tab. 2).”

Line 248-260. Suppl table 3 and the SPR data in figure 2 should be redone. Why are the C3d profiles so different from WT for L1109A, E1110A and likewise some of the ISG mutants? Also, if steady state was used for K_d calculation, please show all the curves and the plot from which the K_d is estimated. An orthogonal biophysical interaction technique might be considered. Mass photometry might be a better technique in this case. A simple pulldown comparing WT and mutants may also be informative.

The shape of the sensorgrams is determined by the different on- and off-rates of the analytes. Some interactions are characterised by fast on- and slow off-rates, while sometimes the opposite is the case. If all curves looked the same, they would give rise to identical K_D values. Moreover, some interactions can trigger conformational changes. These can then result in 2-stage reactions (with 2 separable on- and off-rates) which change the shapes of the sensorgrams. Not all curves could be kinetically fitted (i.e. K_{off}/K_{on}), but the shapes of their sensorgrams are still informative. In such cases, we determined the K_D using steady state analysis. To make the figure (new Fig. 3) more uniform and easier to understand, we now only show the relative differences in binding affinities. Sensorgrams are shown in the supplement alongside steady state binding curves. Now, within each group of mutants, the same method for K_D calculation was employed (see updated supplementary table 3). This did not change the overall outcome of our measurements but we agree that this the correct way of presenting the data.

We have tried pull-downs initially to get a first idea about the effect of our mutations on the overall binding affinity. While we were able to see that they all bind less than the wt, we could not discern any differences between the individual mutants. The C3d pull-down is shown as an example below. Pull downs are an excellent method to identify novel protein:protein interactions, but, as they are not quantitative, they are ill-suited to dissect more subtle differences in relative contributions of individual residues to overall K_Ds. SPR is one of the most established and reliable techniques for determining K_Ds in protein:protein interactions with high precision, also taking the off-rates of the interaction into account. The popularity of SPR especially for characterisation of receptor:ligand interactions is demonstrated by countless publications. Below are some examples where SPR is used to characterise other parasite (surface) proteins with their serum ligands:

Lane-Serff et al., (doi.org/10.7554/eLife.13044)
MacLeod et al., ([10.1038/s41467-020-15125-y](https://doi.org/10.1038/s41467-020-15125-y))
Wright et al. ([10.1038/nature13715](https://doi.org/10.1038/nature13715))
Reichhardt et al. (doi.org/10.1073/pnas.1909973116)

Furthermore, the unique chemistry of the SPR sensor chip CAP allowed us to measure interactions with ISG65 immobilised onto a surface via its C-terminus, thus mimicking its orientation on the cell surface by maintaining its polarity. Mass photometry is an interesting alternative but it works in solution which does not render it the first choice for a system such as ours. Furthermore, mass photometry still is a very new technique and we do not readily have access to an instrument.

C3d pull-down. Purified ISG65 mutants were immobilised on IMAC spin-columns. C3d was added, the column washed and the bound complex finally eluted with buffer containing 500 mM imidazole. Y70A, Y211A and T297A showed clearly reduced binding to C3d compared to ISG65 wt, while almost no binding was detected for Y293A.

Line 262. Why was Arg1134 mutated if not a contact residues in the structure ? No rationale is given

We apologise for not providing a clear rationale for the reader. Arg1134 is located in a long, flexible loop that was found to be protected from deuterium uptake in HDX-MS. Although not a contacting residue in our structure, we assumed that residues from this loop may engage in transient interactions with C3. Among the residues that we found to be protected (sequence IGGLR) we reasoned that mutation of the largest side chain will yield the most

pronounced effect. Indeed, affinity was reduced by 70% upon mutation to alanine (Fig. 3). We have now clarified the rationale behind mutation of this residue in the text.

“Surprisingly, despite its localisation outside the interface, Arg1134 was centred in a ‘protection hotspot’, indicating limited solvent accessibility (Supplementary Fig. 9, 11). In line with this finding, mutation of Arg1134 on C3d did indeed cause a decrease in K_D to 30%. Considering that Arg1134 is located within a long (26 aa) loop close to the interface, it seems conceivable that a transient interaction with ISG65 during binding may occur.”

Line 281. C3a is a very charged protein and the SPR experiment may not reflect the interaction that maybe is present in the structure, although the evidence is very limited for this in the figures. A control experiment is needed. Mutate ISG65 or C3a at the putative interacting residues or use C5a as control. Otherwise, remove the paragraph

The reviewer is correct that C3a is a very charged protein, but we have to point out that all our SPR sensorgrams are reference-subtracted i.e. we have already subtracted the charge-driven unspecific binding of C3a to the chip matrix in our figures. The remaining binding is specific for ISG65.

As suggested, we have provided SPR data of ISG65 mutants to increase the confidence in our results. Mutations in both interacting residues in ISG65 resulted in a significantly reduced affinity compared to C3a in comparison to ISG65 wt. (Supplementary Fig.9 and Supplementary Table 3).

Line 309. It unclear from Fig 3B what is ligand and what is analyte in the FH experiment.

We agree that the figure and the corresponding description combined were insufficient to reconstruct the experimental setup. The description of the Figure (now Figure 5b) has been appended:
“SPR sensorgrams (right) show the effect of ISG65 on the interaction between C3b and fH. Immobilized C3b (ligand) binds CCP 15-20 (analyte) (upper panel), while CCP 15-20 binding to C3b is abrogated when C3b (analyte 1) and CCP 15-20 (analyte 2) are sequentially applied to immobilised ISG65 (ligand) (lower panel). Beginning and end of the fH15-20 injections are marked with red arrows.”

Line 320. This argument is unclear. Why can C3b perhaps bind while native C3 cannot bind?

We agree with the reviewer that our arguments have not been made sufficiently clear. The VSG coat forms a physical barrier between the environment and the trypanosome cell membrane, keeping large molecules such as antibodies (~150 kDa) away from receptors and channels that are located close to the membrane. While antibodies as a whole cannot reach down to the membrane, it has recently been shown that due to flexibility between the Fab-arms, these parts of an antibody can penetrate the VSG layer to a certain extent and recognize epitopes closer to the membrane (Hempelmann et al., [10.1016/j.celrep.2021.109923](https://doi.org/10.1016/j.celrep.2021.109923)). As C3 is even larger than antibodies (~180 kDa) it is likely that the whole molecule can similarly not reach the membrane. While in C3, CUB/TED are locked in an unreactive conformation, wedged between MG2, MG8 and CUB, it has been shown that in C3b these domains exhibit considerable flexibility (Rodriguez et al., [10.1074/jbc.M114.605691](https://doi.org/10.1074/jbc.M114.605691); Nishida et al., [10.1073/pnas.0609791104](https://doi.org/10.1073/pnas.0609791104)). Under physiological conditions the salt bridge between Arg102 and Glu1032 is broken, allowing the TED domain to separate from MG1 and swing out by 60-100 Å. The small size of CUB/TED combined with high flexibility could therefore allow C3b to behave analogous to an antibody and bind ISG65 in its down/resting position, while the same is unlikely for native C3 due to the lack of flexibility of these domains. High flexibility of CUB/TED, as described previously, is likely responsible for the lower local resolution of these domains in the cryo-EM structure of ISG65:C3b as compared to ISG65:C3.

A condensed version of the same argument is now given in the results section – Mechanisms of C3 inactivation at the plasma membrane.

Line 464-470. Purification of the proteins need to be much more detailed described. The reader should be able to reproduce what is done

We apologise for supplying insufficient detail in the purification protocol of the proteins. Where applicable, a more detailed description has been added.

Line 478. If C3 is purified from serum it may contain significant amount of C3water. Cation exchange chromatograms documenting that the native C3 used for functional studies is really native C3 should be presented.

Cation exchange chromatograms of native C3, C3b and C3(H₂O) have now been included as part of Supplementary Figure 4. For convenience they are also shown below.

Cation exchange chromatograms of C3 and its proteolytic fragments. Human complement C3 (purified from human serum), C3b (obtained by tryptic cleavage of C3) and C3(H₂O) (generated by incubation of C3 in presence of 0.2 M methylamine) were applied to a Mono S 5/50 GL and eluted over a NaCl gradient. The corresponding peaks are highlighted with an asterisk. The UV traces are depicted as solid black lines, the conductivity is displayed as grey dashes. Each species elutes at a distinct conductivity, no major contaminants were observed in the purified C3 and both C3b and C3(H₂O) could be separated well from other, undesired species. Each C3 fragment was subsequently further purified by size-exclusion chromatography.

Line 491. Why use the recombinant FH fragment and not full FH commercially available?

We have tried full-length fH initially, but we experienced high unspecific background binding in the reference channel. We therefore chose a shortened fH construct which entails the TED interacting domains CCP 19-20 (necessary for fH co-factor activity on cell surfaces) but did not exhibit this problematic behaviour.

Figures and tables

Nowhere is close-up of functionally interesting residues shown together with the experimental cryo-EM map. Examples from both the C3 and C3b complexes should be provided.

We agree that this should be included. We now show close-ups of interacting residues for ISG65:C3 in the supplementary information. This complex was built de novo due to a sufficiently high resolution. For ISG65:C3b the resolution was too low and therefore both proteins were only docked. The combination of cryoEM data together with HDX-MS allowed us to identify key residues which were mutated to alanines and characterised by SPR. These showed decreased binding, confirming the identification of the interface. We have now added a new supplementary figure "validation of the interaction interface" which concentrates the results from various techniques (Supplementary Fig. 9.) We have also added a paragraph with the same name to the results section.

An overlay of ISG65-TED from BOTH structures is needed, most likely the interface is similar but no information is given regarding this.

An overlay of both structures has been included (Supplementary Fig. 7). With an RMSD of 1.3Å ISG65-TED is indeed very similar in both complexes. It is noteworthy, that, as previously mentioned, the resolution of the ISG65:C3b reconstruction was too low to be modelled. The experimental models built in the ISG65:C3 reconstruction were thus docked and real-space refined. Although the overall domain arrangement suggests an identical interface, the lack of high resolution information does not allow for a detailed comparison of the two interfaces.

The manuscript needs a single figure with ISG65 sequence, secondary structure with names, disulfide bridges, contacts to C3/C3b etc. As it is now, this information is spread over multiple poorly designed suppl figures. This figure should be in the main text.

We agree with the reviewer's comment. As suggested, the information has been summarised and the resulting figure has now been included as part of the main text as Figure 4.

Suppl table 2 needs formatting. Two of the hydrogen bonds in the table are VERY short. It is worth inspecting the map at these hydrogen bonds, perhaps something needs to be adjusted or the weight on nonbonded interactions raised substantially during refinement?

As suggested Supplementary Table 2 has been reformatted.

We agree with the reviewer that the hydrogen bonds Tyr293-Lys1111 and Arg77-Leu1109 reported in Supplementary Table 2 are very short and we understand the concern raised. However, we would like to point out that the values reported have a higher coordinate error, given the local resolution of the affected residues. For example, unlike in a high-resolution X-ray structure, at the resolution of our reconstruction, we cannot always see the entire sidechain, let alone distinguish the occupancy of the various possible rotamers. The table is supposed to illustrate the interaction interface and no conclusions are drawn from the distances stated there. In case of Tyr293-Lys1111, the sidechain densities are relatively well defined, and the model does not appear to be incorrectly built (Supplementary Fig. 9b). Similarly, sidechain density for Leu1109 is clear, Arg77 on the other hand (not included our mutation analysis), due to its highly flexible nature, is poorly resolved. This information is now also being reported in form of the model vs map cross-correlation (Supplementary Fig. 9a) and in combination with the images of the respective residues in context of the electron density allows the reader to make a good judgement of the quality of the data. Although no true consensus has been formed in the field of cryoEM, whether or not it is appropriate at lower resolutions, we have indeed used hydrogen-atoms in the real-space refinement of both complexes to lower the number of clashes, especially in poorly resolved regions. To our opinion, the adjustment of weights in absence of experimental data, i.e. electron density, may lead to 'expected' hydrogen bonding distances but won't lead to a more realistic model. Further, the importance assigned to individual residues in the interaction interface is mostly based on the mutational analysis and affinity measurements by SPR, not by reported atomic coordinates of hydrogen atoms.

Suppl fig 1. Is the gel from reducing or non-reducing conditions? Mw of marker proteins should be added. The C3 peptides, were they from the beta-, alpha- or both chains? Were peptides in ANA present?

We apologise for the lack of information. Supplementary Figure 1 has been updated with the missing information. The new figure now shows the identified peptides mapped both on the structure of C3 as well as its sequence. Samples were reduced before SDS-PAGE. The majority of peptides originated from the alpha chain (including the ANA domain) since only this band was excised from the gel. The band for the beta chain is masked by contaminating serum proteins. Due to imprecise manual cutting of the alpha chain gel band some material was likely carried over from the beta chain band, giving rise to few peptide matches also for this chain.

Supp Fig 3. Needs to be better explained for the non-experts and results summarized in table or plots

We have added a detailed explanation of Supplementary Figure 3 (see below), which also describes the gating strategy applied:

Supplementary Fig. 3: Flow cytometry analysis of C3b surface binding and uptake.

T. b. gambiense cells were incubated with AF₅₉₄ labelled human transferrin or AF₅₉₄ labelled human C3b for 0 min at 4°C (surface binding conditions; see Methods part), for 2 min or 5 min at 37°C, respectively. Cells were then fixed and subjected to flow cytometry using an excitation wavelength of 560 nm in combination with a PE-Texas Red filter. **a** Bar graph plotting counts of AF₅₉₄ positive cells (mean with standard deviation, n=3) at 0, 2 and 5 min for C3b_{AF594} alone (black) and in the presence of ISG65 (light grey) or ISG75 (dark grey). Respective experiments with transferrin_{AF594} (white) are shown for comparison. Non-labelled C3b added at 4-fold molar excess (superimposed, hatched bars) competes with C3b_{AF594} surface binding and uptake. **b** A refined gate based on size versus granularity (FSC-A/SSC-A plot) was applied to exclude cell debris. Secondly singlets were gated (FSC-A/FSC-H plot). **c** Lastly, a gate ('labelled', blue) was defined for AF₅₉₄ positive cells (PE-Texas Red-A/FSC-A plot) based on exclusion of signals from non-treated control cells upper panel; cells only), to define AF₅₉₄ positive cells. Histogram plots of cell count versus fluorescence intensity (Count/PE-Texas Red-A) are shown, with the gate indicated as dashed line.

Further, we have added a new plot as Supplementary Figure 3a for flow cytometry analysis of C3b surface binding and uptake: This bar graph plots counts of AF594 positive cells at different timepoints and shows mean values with standard deviation, n=3 (stated in the legend) for transferrin_{AF594} and C3b_{AF594} alone/in the presence of ISG65. We have added 2 control experiments to further confirm specific, receptor mediated uptake of C3b: (1) addition of ISG75 and (2) addition of non-labelled C3b at 4-fold molar excess (that competes with C3b_{AF594} surface binding).

Supp table 1. Standard deviations on bonds is very large. No information is given regarding correlation between map and model anywhere for the two structures, this is needed. A correlation-by-residue plot would be useful, especially if it was indicated on such a plot where interacting residues were present.

We would like to clarify that the quality measure reported in Supplementary Table 1 is not the RMSD (root mean square deviation) of bond length and angles but the RMSZ (root-mean-square value of Z-scores).

The definition according to the PDB User guide to Validation reports is as follows:

“The root-mean-square value of the Z-scores (RMSZ) of bond lengths (or angles) is calculated for individual residues and then averaged for each chain and over the whole molecule. RMSZ scores are expected to lie between 0 and 1. For low-resolution structures, geometry should be tightly restrained and small values are expected. For very high-resolution structures, values approaching 1 may be attained. Values greater than 1 indicate over-fitting of the data. Individual bond lengths or angles with a Z-score greater than 5 or less than -5 merit inspection.” (<https://www wwptdb.org/validation/2017/XrayValidationReportHelp>)

With reported RMSZ values of ~0.25 for bond lengths and ~0.47 for bond angles we are well within the recommended values. Individual bond lengths or angles with extreme Z-scores (> 5, < -5) are not present in either model.

However, we do agree with the reviewer that not enough information was provided to the reader to reliably assess the quality of the structural data and the conclusions drawn. We have included a correlation-by-residue plot as well as a correlation-by-residue plot for the individual interacting residues (Supplementary Fig. 9a)

Furthermore, we would like to highlight once again that due to the reported resolutions of only 3.58 and 3.59 Å with partially significantly lower local resolutions, we did also carry out mutagenesis SPR and HDX-MS to increase the confidence in the interaction interface. (Supplementary Fig.9 – Validation of the interaction interface)

Supp 7+8+9 should be merged to a single figure.

MS data referring to disulphide mapping and images of the AF2 modelled, disulphide-constrained head domain including the surrounding electron density cloud are now provided in a single figure (Supplementary Figure 8).

Supp table 3. It is unclear why different models for fitting the data are chosen for fitting the data for the various data. This must be explained if the SPR data is kept

2-state fitting was chosen because ligand and analyte interact with 2 separable on- and off-rates. A first interaction can, for example, induce a conformational change which triggers a second interaction with a different K_{on} . 2-state reactions typically result in sensorgrams distinctly different from simple 1:1 binding. These interactions are more accurately fitted by a 2-state model. This approach is commonly found in literature. A different fitting algorithm was also used for the interaction between Efb-C and C3 while for the interaction of Efb-C with other C3 fragments the 1:1 Langmuir model was used (Hammel et al. [10.1038/ni1450](https://doi.org/10.1038/ni1450)) In principle all C3 profiles could be fitted with the same, more complex 2-state model, but the least complex model should be used wherever applicable. Both models are standard kinetic fitting algorithms provided by the BIAcore T200 software. Using the 2-state model is not a sign for lower quality but simply reflects the nature of this interaction. To our opinion a 2-state interaction in case of binding to native C3 makes sense, as the TED is less accessible in this C3 conformation (perhaps requiring structural rearrangements) and we also show evidence for a second interaction site at the ANA domain.

Steady state analysis was used when kinetic fitting of the data was not possible or unreliable (either k_{on} or k_{off} could not be determined). The reasons were either

- i) Technical: The data was recorded on a BIAcore T200 which acquires data with a maximum frequency of 10 Hz. This can however be insufficient for very rapid on- and off-rates. A 40 Hz collection rate as available in the BIAcore S200 would be necessary to reliably fit such challenging kinetic profiles as it increases the number of collection points in a certain time window.*
- ii) Biological: Some interactions are characterised by very slow off-rates that do not return back to baseline within a reasonable time frame. In such cases the k_{off} value, and, consequently, the K_D cannot be determined.*

Although kinetic analysis is preferred (as it also takes the off-rate into account), it is for such scenarios that the instrument's software provides the option to also perform steady state binding analysis.

For the interface mutants we were only interested in the relative affinities to their respective wildtype proteins C3d-Avi and ISG65-Avi. This is also true for the group of C3 fragments (Figure 2). While we used the 2-state model throughout group 1 (Ligand: C3d-Avi), for group 2 (Ligand: ISG65-Avi) a mixture of steady state and kinetic analysis was used due to the reasons stated above. We agree that this is a potential source for confusion

and inaccuracies. For greater comparability we now calculated the binding affinities for all proteins within group 2 using steady state analysis.

Reviewer #2 (Remarks to the Author):

Trypanosoma brucei gambiense are exclusively extracellular in their mammalian hosts and are therefore subject to a constant onslaught of both innate and adaptive immune responses. As a result they have developed multifaceted strategies of evasion and subversion in order to survive.

In this manuscript the authors identify the *Trypanosoma brucei gambiense* surface protein ISG65 to be a receptor of the human complement factor C3 and present data in support of this binding interfering with the killing of the parasite by the alternative pathway. They use an extensive array of different molecular and structural biology techniques to identify and verify binding and to define the binding interface. In particular, they present the cryo-EM structures of ISG65 in complex with C3 and C3b and use SAXS and AlphaFold modeling to augment the structure models. Mutagenesis analysis was employed to confirm involvement of specific residues in the binding interface. SPR experiments show binding of different C3 fragments to ISG65 suggesting the surface protein interferes at various stages of the complement cascade. Interestingly the authors also find by competition SPR experiments that binding of factor H to C3b can be abrogated by ISG65. This work reveals another layer to the multipronged evasion strategies used by this African trypanosome.

Minor points

1) For the reader to fully appreciate the extent of known immune evasion strategies used by African trypanosomes it might be helpful to add a few keywords to at least give a broad overview of these mechanisms (cited in references 2-11). A few other papers might warrant inclusion here (Macleod et al., Nat. Commun. 2020 – FHR receptor; Pinger et al., Nat. Microbiol. 2018 – immune evasion by O-glycosylation of the VSG)

We agree with the reviewer's suggestion. A more comprehensive overview of immune evasion mechanisms also citing the suggested papers is now included in the introduction.

2) In Fig. 1D the arrows labelled with ISG65/ISG75 concentration seem confusing. Might it not be better to add a legend to clearly state the ISG concentration corresponding to each curve? Alternatively add this information in the caption.

We agree with the reviewer that this representation is confusing. We have changed the symbol of the arrow and also mention the concentration range (7.5 μ M – 0 μ M) in the figure legend.

3) In line 109 you reference the FACS data with Supplementary Fig. 2 this data is shown in Supplementary Fig. 3 though. The fluorescence microscopy imaging shown in Supplementary Fig. 2 appears not to be mentioned in the text.

We apologise for this mistake. The error in labelling has been corrected. Fluorescence microscopy data can now be found in Supplementary Figure 2 while FACS data is shown in Supplementary Figure 3.

4) You appear to mean excess not access in line 559

We apologise for the spelling error. This has been corrected.

5) In the 'C3b surface binding and uptake analysis' paragraph in the supplementary Data section several volumes are stated as ml, but should surely be μ l.

Again, we apologise for another spelling error. This has also been corrected.

Reviewer #3 (Remarks to the Author):

This paper describes primarily the structure determinations of ISG65 in complex with C3 and C3b. By itself these are relevant data. However, the paper falls short in respect of functional data that demonstrates what is bound to the trypanosomal surface and what processes are at play on the surface. Since the primary data is only structural, the discussion should be phrased along the lines of possible or plausible structural interpretations and avoid attributing function to molecules.

The paper describes two protein complexes of ISG65 bound to C3 and C3b. It is unclear what the relative importance of either complex in the defense of the trypanosome against complement activation. At what stage complement activation efficiently inhibited is unclear.

Overall, the writing style is one characterized by bold statements or overstatements and is rather non-academic in my opinion. At least for me this manuscript is very hard to read. As an example, the first paragraph consists of

a set of bold statements of which only the last sentence was informative.

Line 61-64, "We demonstrate that ISG65 interferes with the release of anaphylatoxin C3a, as well as with the binding of fH to C3b, thereby inhibiting iC3b formation and consequently halting downstream opsonisation and phagocytosis of *T. b. gambiense*." These claims are not substantiated by the data. Neither C3a release nor effects on phagocytosis are directly experimentally shown.

We agree with the reviewer that we have overinterpreted our results. These claims were indeed not substantiated by our data. Therefore, we have removed both statements from this manuscript.

We could indeed show that ISG65 does not prevent release of C3a (ELISA data added to responses to reviewer #1 and Supplementary Figure 13). We could however support our claim that ISG65 interacts with ANA in native C3. Mutational analysis of residues in the binding interface resulted in significantly reduced binding to C3a. (Supplementary Figure 9 and Supplementary Table 3). Our interpretation of this additional, smaller binding site is that it might serve to position ISG65 next to TED in C3. We do not make any further claims.

While we are confident that ISG65 indeed blocks fH CCP 19-20 binding to C3b, in solution we could not show that this prevents generation of iC3b (see figure below). It has however been shown earlier that lack of CCP19-20 binding only affects co-factor activity of fH in fl mediated C3b cleavage on the cell surface but not in solution (Ferreira et al., <https://doi.org/10.4049/jimmunol.177.9.6308>).

iC3b

degradation assay. The degradation assay was conducted over a time period of 6 hours at 37°C solution using purified proteins (fH, fl, C3b and ISG65). ISG65 was added to C3b prior to addition of other components in 3-fold molar excess. Degradation products were visualized on an Coomassie-stained 8% SDS-gel.

We meant to test the formation of iC3b in a realistic in vivo setting on the parasite cell surface by comparing the situation in the wild-type to a knock-out strain. While we were pursuing the generating of a knock-line in all 3 ISG65 loci we could not achieve a complete knock-out during the time of the revision process (see genome sequencing data added to responses to reviewer #1). This will therefore be part of a separate study.

Line 76, cryo-EM applied to complement proteins is not new, as suggested by the authors. A great example is the structure of the MAC pore by Bubeck and co-workers

We apologise for the phrasing. We did not mean to extend the statement to the entire complement but were rather referring to prior structures, especially complexes of C3 and its proteolytic fragments, solved by X-ray crystallography to mostly low resolution. (Pedersen et al., 10.3389/fimmu.2019.02007 [6 Å]; Garcia et al., 10.1016/j.jmb.2010.07.029 [7.5 Å]; Forneris et al., 10.1126/science.1195821 [4 Å]).

*We have rephrased the sentence, it now specifically states: "The structures shown in this work represent the first structures of a *Trypanosoma* surface protein and of complement factor C3 determined by cryo-EM"*

Line 145, the work by Chen et al. and Rappsilber Mol Cell Proteomics 2016, 15, 2730-2743 revealed the structural features of C3(H₂O).

We have removed this statement. We now cite the work by Chen et al. in a different context:

"In C3(H₂O) the TED has been suggested to be more accessible or more flexibly tethered to the C3 scaffold than in the other fragments (Chen et al., <https://doi.org/10.1074/mcp.m115.056473>)"

Line 157, the phrase 'near-atomic range' or 'near-atomic resolution' is misleading for 3.5 Ang data.

Many papers refer to cryo-EM structures in the range of 3.2-4.2 Å as 'near-atomic range' (Worrall et al., 10.1038/nature20576; Galkin et al., 10.1016/j.str.2014.11.006; Bartesaghi et al., 10.1073/pnas.1402809111; Chua et al., 10.1093/nar/gkw708; von der Ecken et al., 10.1038/nature18295). We appreciate that this description might be better suited for structures around 2 Å, but we feel it is still appropriate to use this term to differentiate cryo-EM structures with side chain density from the bulk of lower resolution structures solved by cryo-EM that did not allow for de novo building as it was the case for ISG65:C3.

Line 358, the authors claim to have presented the function of ISG65. However, no knock-out data is available, thus the claim is unsubstantiated. The word 'function' needs to be replaced by 'role'.

We agree with the reviewer's suggestion. The word 'function' has been substituted for 'role'.

Line 361, 'manipulates innate immunity at several critical intervention points using a multipronged strategy to avoid complement mediated clearance' is unsubstantiated. The paper lacks functional assays to support this statement. Earlier papers (refs 14 and 15) state that complement activation does not progress beyond formation of the C3 convertase (line 52/53). It is not clear what fragments occur at the surface of trypanosomes and what defense mechanisms are at action. It is not shown that factor H binds to trypanosomes.

We agree that we still have a very incomplete picture of the innate immune escape of human-infective African trypanosomes. Native C3, C3b and factor B were found on the surface of T. b. gambiense, whereby factors B and D as well as magnesium enhance binding of C3b. No C5 or components of the terminal pathway were identified and T. b. gambiense does not activate the classic pathway (Devine et al., <https://doi.org/10.1128/iai.52.1.223-229.1986>). It is true that neither factor H nor other negative regulators have so far been identified on the surface of blood stage T. b. gambiense, but it is thinkable that fH could also play a role in immune evasion e.g. in the decay of C3 convertases. Factor H has so far only been identified on the surface of another developmental form of T. brucei (MacLeod et al. <https://doi.org/10.1038/s41467-020-15125-y>)

Trypanosomes employ redundant mechanisms for immune evasion. An example is the escape of the adaptive immune response (VSG transcriptional switching, homologous recombination between VSG loci, hydrodynamic flow mediated antibody clearance by VSGs). It is possible that also for evasion of the alternative pathway trypanosomes utilise several strategies, including, but not limited to C3/C3b/C3(H₂O) recruitment by ISG65. We also do not explicitly rule out that there might be another C3 receptor on the surface:

"Residual uptake of ISG65-saturated C3b can hereby either be attributed to fluid phase endocytosis, partial disintegration of the complex or the presence of another C3 surface receptor."

We have entirely removed our absolute and unsubstantiated statements and have considerably toned down our language. We have re-written the discussion along the evidence that is provided by our structural and biophysical data and we have been more careful in drawing conclusions that are unsubstantiated by data

Line 364, 'master function of ISG65-mediated complement inhibition is to keep the TED away from the plasma membrane', no evidence is provided for this statement. Does this statement refer to a role in cellular immune response? Overall, the paper lacks functional data to statements with respect to biological function, let alone 'master function'.

We apologise for the overstatement, we meant to say that the TED is the main interaction site of ISG65. The sentence describing a 'master function' has been removed in the new discussion.

We indeed have no data supporting that ISG65 is keeping TED/C3d away from the plasma membrane. As we explain in the discussion, we believe that ISG65 primarily has a role similar to VSGs, providing a high-affinity attachment point that serves to transport toxic molecules (driven by hydrodynamic shear forces) to the flagellar pocket at the posterior of the cell, delivering a cargo to the lysosome for degradation. (Engstler et al., 10.1016/j.cell.2007.08.046). This is the only place for endo- and exocytosis, the carrier, ISG65, can then return to the cell surface, regenerated for a new ligand. ISG65 recycling has been shown earlier (Koumandou et al., 10.1128/EC.00273-12).

Line 367-369, it is a (common) misunderstanding that C3(H₂O) is a biological active species of complement activation on a surface. The AP initiator is most likely C3, when loosely associated with a surface, that may adopt the active-like conformation of C3b or (what is referred to as) C3(H₂O), i.e. proteolytically uncleaved C3. No functional data given that indeed C3(H₂O) is bound and thereby stops AP initiation altogether.

We apologise if the previous phrasing was unclear. We did not attempt to make a statement that C3(H₂O) is a biologically active species on the plasma membrane of T. b. gambiense. We admit that the word 'surface' can be misleading when referring to African Trypanosomes. The word cell surface is often synonymously used for plasma membrane as well as the VSG coat, although both are functionally and spatially separate. We do propose that ISG65 can recruit complement C3 fragments (and therefore also C3(H₂O)) from the blood stream in vicinity of the VSG coat by extending beyond the boundaries of the latter. For this to happen C3 does not have to be attached to the coat or the plasma membrane (Figure 6).

*According to a contemporary review the AP is mainly initiated by spontaneous hydrolysis of C3 in solution (tick over) but can be enhanced by absorption on various (microbial) surfaces (Ricklin et al. <https://doi.org/10.1111/imr.12500>). We provide biophysical data (SPR) showing that ISG65 binds C3(H₂O) with a physiological dissociation constant, but the reviewer is right that we do not corroborate this finding with any functional cell-based assay e.g. binding of C3(H₂O) to *T. b. gambiense*. We have considered this option but to our knowledge there is no commercial C3(H₂O)-specific antibody that could be used to detect this fragment on the parasite grown in human serum.*

We have toned down our statement about the consequences of ISG65 binding C3(H₂O) in the discussion and removed the sentence stating that binding of C3(H₂O) inhibits the cascade at its inception. Binding of C3(H₂O) is indeed very unlikely to stall the AP altogether as a single event, but more likely part of a general surface removal of any complement C3 fragment that carries the thioester domain (the main interaction site of ISG65)

Line 370-371, 'Due to the complexity of 369 the complement system and the sheer abundance of complement in the blood stream, a single mechanism is less likely to be successful than a concerted attack. Therefore ...' This statement and subsequent logic are unsubstantiated.

We agree that this is a very unclear formulation. What we meant is the following:

As trypanosomes do employ redundant mechanisms for immune evasion (e.g. VSGs, see above), binding and removal of C3(H₂O) by ISG65 alongside native C3 and C3b might represent a similar approach to effectively interfere with progression of the alternative pathway at several stages. Such a comprehensive strategy has also been proposed for staphylococcal Efb-C which also interacts with native C3, C3b and C3(H₂O) (Hammel et al. [10.1038/ni1450](https://doi.org/10.1038/ni1450)).

Line 373, 'Recruitment of native C3 to the cell surface has several consequences', this is an overstatement.

We agree with the reviewer. This sentence has been removed. In a manner similar to C3b binding, ISG65 might mainly serve as an attachment point for subsequent surface removal of native C3. Binding of C3 substrate to ISG65 in an 'upright' orientation (with the head domain in a membrane-distal orientation, similarly to VSGs) might however also sterically prevent an interaction with AP C3 convertase (C3bBb) that could form on the plasma membrane (Supplementary Figure 13).

Reviewer #4 (Remarks to the Author):

The authors here present a novel characterization of the structure and binding conformers of ISG65 from *T. b. gambiense* using a variety of biophysical techniques. Structural elucidation is essential for a clear understanding of protein function, and this manuscript makes an important contribution to the mechanisms of this host-pathogen interaction and potential immune escape. The conclusions are reasonable and appear well-supported by the data.

The bulk of the findings rely on cryo-EM, SPR, SAXS, and alanine scanning data, and mass spectrometry (MS) plays a minor role in establishing the conclusions drawn. However, if these MS experiments remain in the manuscript, revisions and additional details are very much needed for clarity and reproducibility.

Overall, I recommend that the authors revise and refine the MS portions of the manuscript and the related figures for re-evaluation before publication. I also recommend minor clarifications to 3 non-MS related sections of the text (placed at the end of the comments).

Specific comments are listed below.

Gelband experiment - Main text lines 85-90 / Supplementary Figure 1: As presented, the experimental logic is that, of any proteins identified in the gelband, the highest scoring protein would be the specific enrichment target for ISG65. This is an inaccurate interpretation of protein scores in MS search algorithms. The score only reflects the quality and certainty of the identification, not the likelihood of this protein being most relevant to the experimental question. It is also unclear as presented whether gelbands were excised from both the control and the enriched condition and compared (as inferred from main text line 89), or whether the comparison was done visually from the gel and only the enriched condition was analyzed by MS. In order to show that C3 was uniquely enriched by ISG65, authors should have analyzed both the negative control serum and the ISG65-enriched serum and shown either that C3 appears uniquely in the enrichment condition, or that C3 appears at much greater abundance in the enrichment condition, and if this was done, the figure should reflect comparative data from these runs. A less-compelling but acceptable argument could be made on the basis of C3 being the most abundant protein detected in the enriched sample alone, which depends on either the number of PSMs (if using spectral counting for quantification) or peak intensity (if using area under the curve for quantification). The protein score, however, is irrelevant for proving that C3 is specifically enriched by ISG65. This figure should be changed to present supporting data as described above, and the main text revised to clarify the experiment.

We thank the reviewer for pointing out this mistake. We have only analysed the enriched sample, but this was done independently for 3 of the elution fractions. For all three samples complement 3 ranked on position 1 based on peptide spectrum matches (PSMs). Tables for samples 2 and 3 have now been included in supplementary figure 1 and the text has been revised accordingly.

Intact mass experiment - Main text lines 136-137 / Supp Fig 4: Figure is not easily interpretable as presented. The mass labels in the insets are cut off and cannot be fully read. The figure legend states that the highest intensity peak in panel A is 187,632 but the peak is labeled as 187,637.22 and a 5-Da difference is substantial – was this a typo or is 187,632 Da the expected theoretical mass? Authors should report the theoretical mass of each species, the measured mass, and ideally the mass accuracy in ppm. Masses should be reported to 2 or 3 decimal places, not rounded to the nearest integer, as the reported instruments are high-res accurate mass and capable of greater than unit resolution. The legend should mention what software is being used to generate the data.

We apologise for a poor figure design, clipping of numbers and the mentioned typo. We have prepared a new figure and figure legend (Supplementary Figure 4) in which we also mention the used program. As requested by the reviewer, we now report both measured and theoretical masses.

The ppm error is calculated by comparing the measured to the expected, theoretical mass. Posttranslational modifications however can make this task challenging. Large proteins, such as C3, can acquire many modifications resulting in a wide range of differences in mass, from deamidation of asparagine (-1 mu), oxidation of methionine (+16 mu) to O- and N-glycosylation sites (not all of which are known). Furthermore, since C3 was purified from serum pooled together from multiple donors this likely gave rise to a mixed population of natural C3 isoforms. Exact theoretical mass values may therefore differ and thus are difficult to determine. As it was beyond the scope of our experiment, the precise source of the apparent mass differences was not further investigated.

We therefore decided to measure the samples in parallel, rendering the data consistent and comparable. Here, we were focusing on the relative differences only. A mass difference of 86.7 Da was measured between C3 and C3(H₂O). This difference agrees with the predicted mass according to the reaction scheme in supplementary figure 4 within a 7.5 ppm error of measurement.

	Highest peak in the spectrum	Theoretical mass (without modifications)
C3	187550.54	184326.59
C3(H ₂ O)	187637.24	184414.69
C3b	178549.36	175249.96

Since the goal of this experiment is to confirm the identity of the C3 species used in the SPR experiment, presenting only intact mass measurements requires that those measurements be highly accurate and precise (which cannot be determined as presented). If any fragmentation data were generated, metrics such as the number of unique peptides, sequence coverage, and protein score could be included as in Supp Fig 1 to provide confidence in the identification.

The purification of all 3 C3 species was carried out based on published protocols (Pangburn, MK; [https://doi.org/10.1016/0076-6879\(88\)62106-9](https://doi.org/10.1016/0076-6879(88)62106-9)) which already provides a good level of certainty about their identity. In order to further increase confidence, we carried our intact mass measurements in conjunction with Western-blotting or N-terminal sequencing.

C3 and C3(H₂O) can easily be distinguished from C3b by presence of the ANA domain. ANA or its absence can be detected by i) mass difference of 9 kDa ii) western blotting using an anti-C3a antibody (this antibody does not detect C3b) and iii) N-terminal sequencing (after cleavage of ANA a new N-terminus is created in the alpha chain of C3b).

C3(H₂O) can be distinguished from C3 by a broken thioester. To render the free cysteine unreactive (and prevent unintended dimer formation) we reacted it with iodoacetamide using a published protocol (Pederson et al., [10.15252/embj.201696173](https://doi.org/10.15252/embj.201696173)). This gives rise to a theoretical mass difference of 88.1 Da. The measured difference as reported here is in good agreement with this value.

Additionally, all three protein species can be distinctly separated using ion-exchange chromatography, giving rise to the same chromatograms as previously described (Pangburn, MK, [https://doi.org/10.1016/0076-6879\(88\)62106-9](https://doi.org/10.1016/0076-6879(88)62106-9)). This further increases confidence in the protein identity (Supplementary Fig. 4).

Altogether, processing of the C3 protein resulted in distinct molecules that have distinct molecular weights as reflected by their intact masses. The relative differences in masses are easily interpretable by the explanation we provide here. Although fragmentation data of peptides after protease digestion could in principle confirm these

results, we do not see a compelling reason to include it here. The intact mass measurement is showing the difference to our opinion better than potential sequence coverage of each molecular species.

Furthermore, the potential absence of peptides for a certain C3 domain (e.g. ANA) due to inefficient cleavage or differential ionization properties would not necessarily be proof for its absence in the measured protein. This means it would not aid our goal to distinguish between C3 and C3b.

Further, since contaminating species would lower the confidence of the SPR experiments, the authors could further strengthen this figure by commenting on the relative purity of their target constructs in these MS runs, which has a significantly greater sensitivity than SDS-PAGE. I recommend that the authors consult with the mass spectrometrists associated with this study to generate clearer figures demonstrating an accurate Mw determination.

Largely different amounts of contaminations in each of the C3 species could indeed lead to a wrong estimation of analyte concentration in SPR experiments and hence inaccuracies in absolute K_D determination. All C3 species were however of equally high purity (>95%). In addition to SDS-PAGE analysis, which we believe in principle to be sufficient to judge relative purity for the purpose intended, we have now also added a figure panel which further demonstrates the monodispersity of our samples by SEC-SAXS analysis. Supplementary Figure 4e now also shows scattering traces of native C3, C3(H₂O) and C3b. No additional peaks indicative of contaminations that would invalidate or compromise the accuracies of the concentration measurements of the samples used for our SPR measurements could be identified.

Disulfide mapping experiment - Main text lines 204-206 / Supp Fig 8: Cystamine can be used to promote the formation of disulfide bonds, and this experiment was conducted after an overnight incubation with excess cystamine. Were the authors using this reagent to map the general proximity of various cysteines to refine the cryo-EM model, or were they trying to locate native disulfide bonds? If the latter, what evidence do they have that the linkages observed are not artifacts of the sample prep? If the former, a clarifying comment on the experimental goal and the methods used, either in the main text or the methods, would be helpful.

The goal was to locate native disulphide bonds. The protein that was used for mapping was produced in *D. melanogaster* (insect) S2 cells with a signal sequence for the secretory pathway. This closely resembles native conditions and is a widely used method for producing extracellular, disulphide containing proteins recombinantly (e.g. Wright et al. (10.1038/nature13715)). Both oxidizing conditions and the presence of disulphide isomerases within the endoplasmic reticulum of eukaryotic cells ensure the formation of disulphide bonds (Feige and Hendershot, 10.1016/j.ceb.2010.10.012). Below we also provide a CD spectrum of ISG65, showing that it is correctly folded with minima at 208 and 222 nm that is characteristic for alpha-helical proteins.

Circular dichroism spectrum of ISG65 used for disulphide mapping. The protein is correctly folded showing minima at 208 and 222 nm characteristic for alpha-helical proteins.

In this study, the cystamine was present in the sample to prevent disulphide bond scrambling during all steps of its processing. The previously published data showed suppression of disulphide bond scrambling when cystamine was used. However, some level of scrambling could not be avoided (Pompach et al., 10.1002/jms.1609).

In Supp Fig 8, panel A shows extracted ion chromatograms for three crosslinked peptides. The standard in the field for presenting peptide identifications is an annotated MS2 spectra, rather than an EIC of the MS1 mass, since intact mass alone cannot determine sequence information, and the authors should change the figure to provide this evidence instead.

We apologise for providing insufficient information. MS and MS/MS spectra with annotated fragments of all three disulphide linked peptides are now shown in Supplementary Figure 8.

In addition, the peptide map of the unreduced protein form in panel B indicates that for each of the crosslinked pairs, one of the peptides is detectable in an uncrosslinked form. If this is true, the figure would be improved by including the relative quantification of the linked and unlinked peptide forms to obtain an estimate of the disulfide stoichiometry. Stoichiometry information is important for cryo-EM model refinement because if only 5% of the

target peptide population was detected in a crosslinked form, then constraining the structural model to a conformation that allows that link to exist would not represent the majority conformer.

Information on the positions of disulphides was not used in refinement of the cryo-EM model. This part of the model, i.e. the loop-rich head domain, is not based on traceable electron density and was appended to the final model later. The head domain in which the disulphides are present is largely disordered giving rise to an electron density cloud (Supplementary Fig. 8f). Therefore, we aimed to create a realistic model of the head domain using AlphaFold2 modelling constrained by the positions of the experimentally determined disulphides as well as the boundaries of the density cloud. We deemed this approach more likely for yielding a correct result than relying on AF2 alone. AF2 has been shown in a recent benchmarking test (Gulsevicius and Meiler bioRxiv 2022.02.17.480937; doi: <https://doi.org/10.1101/2022.02.17.480937>) to have deficits in reliably predicting disulphide bonds and flexible regions such as present in the head domain of ISG65. Still, our model of the head domain is meant to be illustrative and merely informs about the extent of the head domain. We do not make any claims regarding the specific function of any of the loops in the head domain.

Semi-quantification of disulphide peptides

The disulphide linked form of NCNIGQSSSEHPCTMTEEWQTPYK peptide represents the majority part compared to its reduced form. For the EGAAALCK – VLNWYCITK peptide and the SAIDCSSTS YEENYDWSANALQVALNSWEDVKPK – HNATPNC DGIQFR peptide, intensities of each peptide differ most likely due to different ionization efficiency. The data therefore has to be considered an estimation. Since the data was not used in refinement but rather served to build a more realistic computational model of the head region, we believe this level of confidence to be sufficient.

The third peptide in panel A is annotated as having a glycosylation event detected “by manual data inspection” but no MS evidence of this modification is provided. Modified peptides are typically a very low proportion of the overall population of that peptide (<10%). Were the authors able to find the unmodified form of this crosslinked peptide, or was the fucosylated paucimannosylated form the only identification possible? If the latter, this is very likely a false identification. If the former, the unmodified form should be presented instead, and any PTMs referenced in the manuscript should be evidenced by annotated MS2 spectra.

The fragmentation spectrum of glycosylated, disulphide linked peptide is now provided including annotation of glycopeptide fragments (Supplementary Fig. 8d). The observed parent mass of the glycosylated disulphide linked peptide was 6328.72 (1055.628 6+) and the theoretical mass of unmodified disulphide linked peptide is 5290.38. The mass difference corresponds to fucosylated paucimannose. In the fragmentation spectrum, oxonium ions corresponding to Hex-HexNAc at m/z 366.1374, HexHex-HexNAc at 528.1883 and HexHex-HexNAc at 690.2554 were observed. Several doubly charged y ions were matched for the peptide SAIDCSSTS YEENYDWSANALQVALNSWEDVKPK. No peptide backbone fragments were observed for HNATPNC DGIQFR. This is likely caused by CID fragmentation as it provides intensive oxonium and glycopeptide fragments and less intense peptide backbone fragments.

The figure below shows extract ion chromatograms of glyco disulphide linked peptide at m/z 1266.5514 (5+) and extract ion chromatogram of non-glyco disulphide linked peptide at m/z 1058.8800 (5+). Ion intensities show three time less signal for the unmodified form compared to the modified form.

References

- 1) *Semi-automated identification of N-Glycopeptides by hydrophilic interaction chromatography, nano-reverse-phase LC-MS/MS, and glycan database search. Pompach P, Chandler KB, Lan R, Edwards N, Goldman R [10.1021/pr201183w](https://doi.org/10.1021/pr201183w)*

- 2) *Ultrasensitive Characterization of Site-Specific Glycosylation of Affinity-Purified Haptoglobin from Lung Cancer Patient Plasma Using 10 μ m i.d. Porous Layer Open Tubular Liquid Chromatography–Linear Ion Trap Collision-Induced Dissociation/Electron Transfer Dissociation Mass Spectrometry.* Dongdong Wang, Marina Hincapie, Tomas Rejtar, and Barry L. Karger [10.1021/ac102825g](https://doi.org/10.1021/ac102825g)
- 3) *Quantitative liquid chromatography-mass spectrometry-multiple reaction monitoring (LC-MS-MRM) analysis of site-specific glycoforms of haptoglobin in liver disease.* Miloslav Sanda, Petr Pompach, Zuzana Brnakova, Jing Wu, Kephher Makambi, Radoslav Goldman [10.1074/mcp.M112.023325](https://doi.org/10.1074/mcp.M112.023325)

HDX experiment - Main text lines 264-268 / Supp data lines 25-34 and Supp Fig 11: Insufficient detail is provided for the actual exchange reaction, including reaction buffer components, pH, and temperature.

The required information has been added:

“Hydrogen deuterium exchange (HDX) was initiated by a 10-fold dilution of 160 μ M ISG65: C3d complex into a deuterated buffer (20 mM HEPES pH 7.5, 150 mM NaCl). The hydrogen deuterium exchange reaction was performed at RT.”

The authors should comment on their choice to include reducing agent in the quench buffer if this was not present in the exchange reaction, as this would affect the conformation of the protein sample and create the potential for artifactual in-exchange.

Exchange and quench buffer differ in their composition. We have added the composition of the exchange buffer in the method. The reason for addition of TCEP to the quench buffer was to reduce the disulphide bonds in the protein and thereby increase the sequence coverage and peptide overlapping. The same approach was successfully applied to another Trypanosoma surface protein with disulphide bonds: (Zoll, S. et al. [10.1038/s41564-017-0085-3](https://doi.org/10.1038/s41564-017-0085-3))

The authors should include basic MS acquisition method parameters.

The required information has been added:

“Mass spectrometer was operated in positive MS mode in the mass range 350 – 2000 m/z. Data were measured with 1 Mb acquisition with two selective accumulations. The capillary voltage was set at 3900 V, drying gas temperature was 180°C and nebulizer gas was 2.0 bar. Spectra of partially deuterated peptides were processed by Data Analysis 4.2 (Bruker Daltonics, Billerica, MA) and by in-house program DeutEx (Kavan, D. and Man, P. “MSTools - Web based application for visualization and presentation of HXMS data” Int. J. Mass Spectrom. 2011, 302: 53-58. <http://dx.doi.org/10.1016/j.ijms.2010.07.030>.”

Data processing methods should be expanded to include details of any manual validation, filtering choices, and back-exchange corrections if applied or reasoning if not.

The required information has been added:

“In the DeutEx software calculation the envelope width intensity was set at 25, error for deuteration was 10 ppm and error for isotopes was 10 ppm. An intensity filter was set at value 25 and rate filter was set at 5. No back-exchange correction was applied, because quantification of the absolute amount of exchange was not desired.”

Figure legend should include an explicit description of what the color scale in panel C is showing (is this the percent difference between free and complexed protein? Does red indicate an increase in exchange or protection? Does red indicate the free compared to the complex, or the complex compared to the free? Needs to be much more clear).

We agree with the reviewer that this might lead to confusion. The figure legend has been updated (Supplementary Figure 11).

*ISG65 - The red colour represents level of protection in ISG65/C3d complex compared to ISG65.
C3d - The red colour represents level of protection in ISG65/C3d complex compared to C3d.*

The authors mention only one residue specifically from this experiment, Arg1134, but multiple regions look to have equal amounts of protection. Can the authors comment on the other regions of protection? Did the authors have overlapping peptides covering residue 1134 to narrow down this residue as being specifically protected?

(This information should be available in the peptide map in panel B but the residue numbers are not adjusted to match the numbering in panel C.) If not, conclusions have to remain at the peptide level, and should not be extrapolated to single residue level.

Several patches in C3d show significant protection levels comparable to Arg1134. All, with the exception of a patch at the N-terminus of C3d, are located within or in close proximity to the interface. Arg1134 is located within a large flexible loop as part of a larger patch. In the cryo-EM structure this part of the patch is further away from the interface, but it is conceivable that due to its length it could transiently contact ISG65, thus explaining the observed protection from deuterium uptake. As a role in complex formation for a residue that is not directly located in the binding interface is less obvious, we decided to confirm it by site-directed mutagenesis. It is because of this and not a particularly high protection level that we specifically mention it.

While we have many overlapping peptides for this region (Supplementary Figure 11), our data does indeed not allow us to comment on the protection of a single residue such as R1134 based on HDX-MS data alone. Within the peptide 1130-1135 with the sequence IGGLR we reasoned that mutation of the largest side chain would yield the most pronounced effect. The R1134A mutant shows 70% reduced binding affinity as compared to the wt. This suggests a role of this loop in the interaction between ISG65 and C3 even though not visible in the structure. We have now provided a clearer statement in the text:

“Surprisingly, despite its localisation outside the interface, Arg1134 was centred in a ‘protection hotspot’, indicating limited solvent accessibility (Supplementary Fig. 9, 11). In line with this finding, mutation of Arg1134 on C3d did indeed cause a decrease in K_D to 30%. Considering that Arg1134 is located within a long (26 aa) loop close to the interface, it seems conceivable that a transient interaction with ISG65 during binding may occur.”

We apologise for incorrect and inconsistent numbering. The corresponding figure (Supplementary Figure 11) has been updated and the error corrected.

Additionally, protection in C3d appears to be strongest at 20 minutes and largely insignificant at shorter or longer timepoints – does this correlate with the authors’ understanding of the affinity and residence time of the binding interaction? The peptide containing Y293 in ISG65 shows the strongest protection at a different timepoint (2 min) than its proposed binding interface residues 1109-1119 in C3d (20 min) – can the authors comment on this?

We thank the reviewer for pointing this out. We suggest that this observation is the result of different deuteration rates in the un-complexed, free proteins due to differences in local fold architecture. The region 1109-1119 in C3d alone shows very little to almost no deuterium uptake over time (~10% uptake after 7200s). Consequently, any additional protection from deuterium uptake as a result of complex formation will only be measurable after a longer period of time. The corresponding region in ISG65 (290-300) on the other hand deuterated quickly in the free protein (~30% uptake already after 20s). Interaction with C3d then rapidly slowed down deuterium uptake in this region of ISG65, already after the first time point (<10% after 20s) giving rise to pronounced levels of protection.

While we do not think that this data allows us to speculate about affinity and residence time, it might however provide clues about local protein structure. Interestingly, residues 290-300 are part of alpha-helix 4 in the complex structure of ISG65. While well-structured elements such as alpha-helices typically show slow deuteration rates this is not the case for the C-terminal, kinked part of alpha-helix 4. This is in contrast to expectedly slow deuteration rates in the larger, N-terminal part of the same helix. It is therefore conceivable that the kinked, C-terminal part of helix 4 is adopting a different, less structured conformation in free ISG65 as compared to the complex with C3d. More experiments would be necessary to provide further evidence for this hypothesis.

I suggest the authors include uptake plots for all regions affected in the timescale plot, and include a structure in this figure to highlight where in the 3D conformer protection occurs, particularly as those sites relate to the binding interface residues detailed in the main text lines 227-234.

As suggested, we have now included uptake plots for all peptides of ISG65 and C3d (Supplementary Figure 11). We have also added a new supplementary figure showing deuterium protection mapped onto the structures of ISG65 and C3d and marked the interaction interface as identified (Supplementary Figure 9c).

General notes on supplemental method text

Lines 97-99: If more than one run was acquired, authors should include the amount of material injected for runs (if measured) and whether injection amounts were normalized across samples.

In total 3 gel bands (corresponding to 3 elution fractions) were analysed. In each measurement 1/5th of the dissolved sample was injected onto the column and analysed by MS/MS. Bands were cut out manually, resulting in worse reproducibility compared to automatic excision. Therefore no normalisation was carried out:

“Single gel bands (3 in total) as depicted in supplementary figure 1 were cut out, chopped into small pieces, reduced with dithiothreitol, alkylated with iodoacetamide and digested with trypsin overnight. Peptides were extracted from each gel piece, lyophilized in speed-vac, dissolved in 0.1% formic acid and 1/5th of the dissolved sample was separated on an UltiMate 3000 RSLCnano system (Thermo Fisher Scientific) coupled to a Mass Spectrometer Orbitrap Fusion Lumos (Thermo Fisher Scientific). No normalisation was carried out since bands were cut out manually.”

Line 106: For reproducibility, authors need to specify the range of %B increase (5-35%? 5-40%?) and specific time period, as the 65 minutes described could refer to either the entire gradient including wash and re-equilibration steps, or just the active elution portion of the gradient.

The requested information has been included in the supplementary information.

“Using a constant flow rate of 300 nL/min a 65 min elution gradient was started at 5% B (0.1% formic acid in 99.9% acetonitrile) and 95% A (0.1% formic acid). The gradient reached 30% B at 52 min, 90% B at 53 min, and was then kept constant until 57 min before being reduced to 5% B at 58 min.”

Lines 106-108: Authors should note basic MS acquisition parameters including resolution of MS1 and MS2 scans, AGC settings, maximum injection times, whether the fragmentation data was obtained as data-dependent or data-independent acquisition, collision type and detector used for MS2 scans, loop count settings, and dynamic exclusion settings.

The requested information has been included in the supplementary information.

“For the first minute nanospray was set to 1600V, 275°C source temperature, measuring the scans in the range of m/z 350-2000. Orbitrap detector was used for MS with the resolution 120000, AGC target value was set custom as well as the maximum injection time, MSMS was acquired also using orbitrap with resolution 30000, the data was acquired in data dependent manner, ions were fragmented by HCD with dynamic exclusion set to 60s.”

Line 110: Search algorithm is "MS Amanda", not just "Amanda" and algorithms should be cited wherever possible. Authors should specify what databases were used for Homo sapiens and common contaminants (likely UniProt? In which case, needs to include a UniProt reference identifier and the access date.) Authors should also note whether they included the ISG65 sequence in the database, as peptides in the sample mapping to this protein will be incorrectly assigned to a human protein if its own sequence is not available to search. Authors need to state general search settings for data, including mass tolerances for both MS1 and MS2 scans, variable and fixed modifications, number of missed cleavages allowed, and false discovery rate thresholds.

The requested information has been included in the supplementary information:

“The software Proteome Discoverer 2.3 (Thermo Fisher Scientific) was used for peptide and protein identification using Sequest and MS Amanda as search engines and databases of *Homo sapiens* (downloaded from Uniprot combining reviewed and unreviewed sequences on November 12th 2018) and common contaminants (downloaded from Proteome Discoverer software on December 21th 2018). The protein sequence of ISG65 was included in database for the searching. Mass tolerance for MS was 10 ppm and MSMS 0.6 Da for Sequest, tolerance in MS 5ppm and MSMS 0.06 Da in MS Amanda. The fixed modification included carbamidomethylation on cystein, variable modification was set to methionine oxidation and deamidation of asparagine and glutamine, 2 missed cleavages were allowed, FDR target set to 1%.”

Non-MS-related comments

Main text lines 92-101 / Fig 1A: Since the authors use Western blotting for verification later in the text, the SDS-PAGE analysis would be strongly improved by adding a Western blot here, as SDS-PAGE does not provide protein identity. Also recommend editing the figure to move the gel inset away from the y-axes of the chromatogram, as currently the units of the graph can be misread as the Mw markers for the gel, and adding Mw markers to the gel specifically.

As suggested by the reviewer, we have added a Western blot of the complex peak (Figure 1a) whereby we used an anti-His HRP conjugated antibody to specifically detect His-tagged ISG65 (which was purified by IMAC). The identity of purified C3b that was used to prepare the complex was already confirmed by intact MS measurement, N-terminal sequencing as well the purification procedure itself resulting in a specific ion-exchange retention time characteristic for C3b (Pangburn, MK, [https://doi.org/10.1016/0076-6879\(88\)62106-9](https://doi.org/10.1016/0076-6879(88)62106-9)).

As requested, we have also moved the marker away from the y-axis of the chromatogram to avoid confusion.

Fig 1B: The authors interpret this blot to read that the ISG65:C3b complex exists in a 1:1 stoichiometry, but it is unclear how this conclusion was derived (band densitometry in ImageJ? By eye?). Since the methods state that ISG65 was added at 2x excess, the free ISG65 band should be of equal intensity with the complex band, but it appears substoichiometric from the image. More detail on this calculation would be helpful. If a Western blot of the sample is available, this would also help strengthen the figure.

The stoichiometry presents as a 1:1 complex based on the size of the C3b:ISG65 band according to the molecular weight standard on the left hand side. It is true that for complex preparation ISG65 was added in 2x molar excess to saturate all free C3b. It is also true that since C3b and ISG65 interact with 1:1 stoichiometry (which we also show by our cryo-EM structure) we do expect equimolar amounts of complex and free ISG65 on the gel (when ISG65 is added in 2x molar excess). It is however not true that this would give rise to identical

band intensities. Band intensity depends on the number of residues that bind Coomassie (R250 or G250 bind to basic and hydrophobic amino acids) and thereby on the size, molecular weight or chain length of the protein. Thus, the larger the protein, the more of these residues are statistically present and more stain can be retained by the protein.

We have decided to not add another Western blot to this figure as we believe that this information would be redundant. The identity of both proteins used for crosslinking was already confirmed by several independent means.

To additionally ease the reviewer's concern about the 1:1 stoichiometry (which, as previously mentioned, is confirmed by our cryo-EM structure) we would also like to share a molecular weight estimation for the C3b:IGS65 complex from small angle X-ray scattering (screenshot below). MW estimation was carried out by 5 different methods resulting in an average MW of 233.4 kDa. This is within the acceptable 5-10% error of this method and it agrees with the molecular weight estimation on the gel. We did not include the scattering data in the manuscript since we believe that the gel showing the crosslinked complex is more accessible for the reader.

Molecular weight estimation of the C3b:IGS65 complex by small angle X-ray scattering following size exclusion chromatography (SEC-SAXS). MW estimation was carried out by 5 independent calculation methods resulting in an average of 233.4 kDa. This agrees well with the theoretical mass of 217 kDa and confirms a 1:1 stoichiometry. MW estimation was carried out using ATSAS: (<https://www.embl-hamburg.de/biosaxs/software.html>)

Main text lines 106-109 / Supp Fig 2: What is the n of this experiment? Should be included in the legend.

We have added a new plot as Supplementary Figure 3a for flow cytometry analysis of C3b surface binding and uptake: This bar graph plots counts of AF₅₉₄ positive cells at different timepoints and shows mean values with standard deviation, n=3 (stated in the legend) for transferrin_{AF594} and C3b_{AF594} alone/in the presence of ISG65. We have added 2 control experiments to further confirm specific, receptor mediated uptake of C3b: (1) addition of ISG75 and (2) addition of non-labelled C3b at 4-fold molar excess (that competes with C3b_{AF594} surface binding).

REVIEWER COMMENTS

Reviewer #1 (Remarks to the Author):

General comments

The presentation has been somewhat improved and important functional data has been added. However, authors still lack to pinpoint exactly how ISG65 interferes with the alternative pathway of complement activation and how *T. brucei* benefits from expressing ISG65. There are relatively simple experiments that need to be performed. Also, the fact that ISG65 in the fluid phase does NOT prevent FI degradation should be presented to the reader rather than hidden in a rebuttal.

There are two strategies for improving the paper. Authors can perform additional functional experiments that allows them to build a more qualified model for the function of ISG65. This does not necessarily require an ISG65 knockout as claimed in the rebuttal and the required reagents are readily available. The alternative is to focus much more on the important and well performed integrative structural biology work and present the structures in more details and then down weight the functional implications. There is no doubt that authors have captured relevant complexes in cryo-EM, which is also confirmed by the excellent agreement with the recent paper by Mcleod on ISG65. The structural work deserves a more detailed presentation, in the current manuscript it is to some degree buried in the supporting information.

Major issues

Authors base their elimination of the CP as initiator on a single paper from 1986 by Devine et al. There are two problems with this. First, the lectin pathway was not discovered in 86. Second, in table 1 of the paper by Devine, there is a single data point for anti-C3c in the presence of EGTA, that would inhibit both CP and LP. In fact, there is quite a difference between NHS and EG-NHS. But, since the standard deviation is not provided, one cannot judge whether this is noise or reflects a true inhibition of a pathway. This paper may have been acceptable in 1986, but cannot form the basis for rejecting a contribution from the LP or CP in 2022.

In a modern paper, experiments with serum lacking selected complement components would be expected. Even a minor C3b deposition through LP/CP could be amplified in the AP. Authors must provide own evidence that CP/LP C3 convertase is not relevant for complement activation on trypanosomes or find additional evidence in the literature. It is not difficult to generate such data

using commercially available serum depleted for components of CP, AP and terminal pathway (TP), eg C2, FB and C5. If such experiments are not performed or more convincing data from the literature is presented, they should adjust their model and state that the actual initiation mechanism remains to be settled.

The antibody used in the study by Devine for quantitation does not distinguish between C3, C3b, C3water, iC3b. Trypanosomes were incubated with 3 mg purified C3 in 500 ul. If this purified C3 contains or during the assay ticked over to 0.5% C3water (which is a conservative estimate), the concentration would be 30 ug/ml or 160 nM. The experimental data presented in Devine 1986 therefore cannot form the only basis for discussion on whether native C3 binds to trypanosomes. Authors should perform their own C3 binding experiments using an antibody that is specific to C3 and does not react with C3water. Perhaps better, they could use native C3 in a competition experiment where labelled C3b binding is studied, fig 2b would be an excellent starting point. Native C3 could be used as competitor in flow and phago experiment. This would eliminate the influence of C3 tickover during the experiment and answer the crucial question of whether Tb bound ISG65 actually can bind native C3. If such experiments are not performed, the discussion must reflect that it is currently unknown whether native C3 can bind to ISG65 presenting trypanosomes.

The discussion regarding “Mechanisms of C3 inactivation at the plasma membrane” should be more clear on where covalently deposited C3b could be present. Also it should be made clear that C3b could be covalently (the membrane or anything embedded in the membrane) and non-covalently (through ISG65 binding) attached. Perhaps this can be included in an updated figure 6.

The data presented in the rebuttal letter on FH assisted FI degradation of C3b should be presented in the manuscript. Authors show in the rebuttal letter that ISG65 has no effect of fluid phase FH assisted FI degradation but they suggest the opposite in the text. This is contradictory. If they hypothesize that FH will disrupt the activity of FH on a trypanosome, they should provide experimental evidence for this. If in fact ISG65 does prevent FH cofactor activity it totally contradicts that ISG65 blocks progression of the alternative pathway as stated in the abstract. If FH activity is suppressed, the prediction is uncontrolled AP amplification. There are recent reviews on FH that authors may want to consult that provide excellent overviews of FH biology and FH focused complement evasion PMID: 33717083 PMID: 35469595.

Abstract Line 10-11. “and its activation fragments and that it is sufficient to block progression of the alternative pathway in vitro”

This is not shown in the manuscript. In fact, authors fail to identify inhibition of the AP at any step. They show inhibition of the terminal pathway.

Line 59. C5 convertase could be assembled but not able to bind or cleave substrate. Devine et al 1986 is severely overinterpreted in the manuscript.

Line 206: Supp fig 7 should include 7pi6, the C3d:ISG65 crystal structure. Although released during revision, it should be properly presented, compared to the cryo-EM structures and discussed in this manuscript. A comparison between ISG65 bound C3/C3b and their unbound reference structures could also be placed in supp fig 7.

Line 220. The disulfide bridge residue numbering is not the numbering used in sup fig 8 and figure 4A. In pdb entry 7pi6, the disulfide Cys43-Cys200 is present that contradicts the mapping presented here. The electron density for 7pi6 confirms this disulfide. This should be addressed.

Line 225-30. Panels showing the quality of 3D reconstruction for the regions mentioned here should be added to main figure 4 which currently only has 2 panels. This is just one example, authors need to provide multiple examples of map quality in the main figures. Currently there are only low magnification views of map in figure 3A. This is not proper for a manuscript that is so much based on two important cryo-EM structures. The sup fig 8 has many excellent panels with magnified views containing important residues in the structures with the map. These panels should go into the main figures instead.

Figure 6. Unless the authors provide experimental evidence that ISG65 on *T. brucei* can bind native C3, they should omit a model detailing when native C3 can be bound. The figure is also misleading if molecules are drawn to scale. As ISG65 is shown here to the right, native C3 would penetrate significantly into the VSG layer whereas it is more reasonable that C3b can bind to the “active” ISG65. C3 binding should be eliminated from model unless supported by experimental evidence

Line 440. The fact that there is no structure of a C5 convertase should not prevent authors from testing how ISG65 interferes with C5 convertase. Authors could separate AP or CP C3 deposition from MAC assembly by using e.g. C5 deficient serum in the first step, wash, and then add the “missing” factor together with ISG65 to study the effect of ISG65 on the CP and AP C5 convertase and MAC assembly. These are not very demanding experiments

Comments to the revised manuscript

Minor points

Line 31. Spell out abbreviations for Tb proteins

Line 45. MAC formation occurs in the terminal pathway, not the AP. It is possible to have MAC formation e.g in FB depleted serum in vitro. Authors must include the CP/LP C5 convertase in their analysis of mechanism.

Line 116. Add legend letters

Line 136. Add ref to sup fig 3 here

Line 146. Add reference Fig 2D

Line 149. Add ref to figure 1

Line 160-2. "physiologically relevant" instead ?. Mention the C3 and C3b concentrations in ref 23 for comparison with Kd here.

Line 189: TED is not in contact with ANA in native C3. It is the association with CUB and MG8 that fixes TED in native C3. Authors should clarify.

Line 199: Quantitate the resemblance between C3b structures by rmsd number

Line 241. Why is Fig 4a relevant here? 5a appear to be relevant

Line 244-45. A panel presenting the quality of the map underlying the identification of these hydrogen bonds must be displayed

Line 265. Show the density of Trp211 in main figure

Line 330-8. The discussion here is an over-interpretation. SAXS data for C3b has shown that TED can release from the MG1 domain but not to the extent suggested here. C3b is also not smaller than antibodies, it has a higher molecular weight and antibodies are more flexible. So the argument is invalid. Without further evidence, the distinction between C3 and C3b with respect to VSG penetrance is not justified. Also, the conformation of ISG65 could be hugely influenced by the VSG coat as compared to the SAXS experiment conducted here. That being said, the SAXS models are likely to present two possible conformations of the C-terminal regions in solution well.

Line 452. Ref 56 is out of context

Line 723. What tool in phenix is used for this docking? Perhaps coot or chimera instead? Authors broadly mention "phenix" as reference for multiple steps. Phenix is a collection of programs written by many different contributors. It would be more informative to specify the actual program in, e.g phenix.real_space_refine. To compare, REFMAC is a CCP4 program in the same manner as phenix.real_space_refine is a phenix program. Also molprobity used for validation was perhaps the phenix version?

Figure 1. CP/LP C3 and C5 convertase are not depicted. They cannot be excluded from the mechanistic analysis given the weak basis presented by the authors for excluding the CP/LP. FP binds to C3bB, not only to C3bBb. C3dg does not bind CR4, it binds the C3c moiety within iC3b, PMID: 28292891.

Figure 3. Open book views of the C3-ISG65 and C3b-ISG65 views could be added here.

Figure 4. Perfect place for adding magnified views of model with map in important regions.

Figure 5. Panel b could be much smaller and still provide the message. Show the map quality for the C3a interaction.

Supp fig 12. Panel B, unit on horizontal axis is nm not Å

Reviewer #4 (Remarks to the Author):

Overall, the authors have satisfactorily addressed most of my previous comments. There are a few minor edits still needed and a few major concerns on the disulfide crosslinking experiment.

Supplemental method text changes

The revised methods are much improved and satisfactorily detailed. Specific final edit suggestions below:

HDX section:

- Please include manufacturer info for nepenthesin-2 column.
- Acidity of deuterated solutions should be reported as pD, not pH.

In-gel section:

- AGC and MIT values are reported as “custom”, which means the user enters their own percent value; text should include the percent value entered.
- Fragmentation method is reported as HCD; please also include collision energy setting for completeness.
- The authors state that the mass tolerances for Sequest were 10 ppm and 0.6 Da for MS1 and MS2 scans, respectively, but in MS Amanda these were set to 5 ppm and 0.06 Da. Because the MS2 scans

were acquired in the Orbitrap, the 0.6 Da mass tolerance in Sequest is pretty broad, and a 0.06 Da setting would make more sense for both search engines. Can the authors confirm whether the 0.6 Da setting is correct vs. a typo? No change needed if these values are correct.

Supplemental figure changes

SF1-B: Looks good. Legend text reads that “complement C3 was identified with the highest PSM score” – please amend to “highest number of PSMs” for accuracy.

SF1-C: Nice addition. Minor suggestion: a color legend in the figure itself, in addition to the legend text, would be helpful, though not strictly necessary.

Intact mass experiment changes

My apologies to the authors – I was inappropriately requesting a standard of precision common to peptide experiments from an intact mass experiment. I appreciate that intact mass is much more complicated than peptide mass, given the variation from potential proteoforms and the lack of a detectable monoisotopic peak in most intact measurements. I thank the authors for the changes implemented and the time invested in their detailed response. The figure is much clearer and the added blot evidence is helpful.

Disulfide mapping experiment

The authors have helpfully clarified that they are looking for native disulfide linkages as part of a general characterization but not for use in refining the cryo-EM model. The figure is still difficult to parse, and the numbering in the main text (which specifies bridges between C12-C137, C149-C168, and C209-C220) does not agree with the peptide map provided in SF8-A,B. In the peptide map, the sequence picks up at residue 32, which would make detection of a bridge with C12 impossible, and of the remaining specified cysteines, only residue 168 is actually a Cys in the map. The peptide sequences listed in SF8-C,D,E suggest the authors mean to specify bridges between C43-C168, C181-C200, and an internal link of C240-C251. If these are accurate, the main text needs to be revised. If these are not correct, the figure needs to be revised.

I thank the authors for providing MS2 spectra of the three disulfide linkages. The first and third peptides (EGAAALCK-VLNWYCITK and NCNIGQ....) are credibly presented. I am skeptical of the second peptide (HNATPN....) as presented, however.

- The methods for this section (main text lines 625-627) are well-detailed for the LC portion of the experiment but not for the MS acquisition or search. The authors should expand this text to include parameters of both the instrument acquisition (either reference a publication detailing "the PASEF method for standard proteomics" or listing specific parameters here) and the data search, and specifically what sequences were searched and what variable modifications were included (presumably fucosylated paucimannosylation, but what others?).

- The annotated sequence in Supp Fig 8d shows the modification present on N176, but with only 4 fragment ions from the other peptide in the crosslink, this mod cannot be localized to that specific Asn residue based on the spectral data. The modification would need to be presented as attached agnostically to the cross-linked pair. Alternatively, N-glycosylation often follows sequence motif patterns in human cells – did the authors assign the modification based on NATP being a known sequence motif in this species? If so, that could justify the localization if accompanied by explanatory text and a reference establishing the motif.

- The modification is shown on the peptide sequence with the green dot moieties being branched, but on the spectra, shown as linear. The presentation should be consistent between the two.

- The response to the authors includes an EIC for the glyco-crosslinked form and the non-glyco-crosslinked form. The glyco form does appear as the more abundant form, but have the authors investigated the MS2 data for the non-glyco form? The unmodified crosslinked peptide may provide more backbone fragment ions and if so, I recommend including it side-by-side with the glyco form to raise confidence in this peptide identification.

HDX experiment

I thank the authors for including the additional detail requested and for the thoughtful response to the differences in timepoints for strongest protection. I have no further comments on this section.

Other comments

The authors have sufficiently addressed previous concerns/suggestions.

REVIEWER COMMENTS

Reviewer #1 (Remarks to the Author):

The presentation has been somewhat improved and important functional data has been added. However, authors still lack to pinpoint exactly how ISG65 interferes with the alternative pathway of complement activation and how *T. brucei* benefits from expressing ISG65. There are relatively simple experiments that need to be performed. Also, the fact that ISG65 in the fluid phase does NOT prevent FI degradation should be presented to the reader rather than hidden in a rebuttal.

There are two strategies for improving the paper. Authors can perform additional functional experiments that allows them to build a more qualified model for the function of ISG65. This does not necessarily require an ISG65 knockout as claimed in the rebuttal and the required reagents are readily available. The alternative is to focus much more on the important and well performed integrative structural biology work and present the structures in more details and then down weight the functional implications. There is no doubt that authors have captured relevant complexes in cryo-EM, which is also confirmed by the excellent agreement with the recent paper by McLeod on ISG65. The structural work deserves a more detailed presentation, in the current manuscript it is to some degree buried in the supporting information.

*Here, we would like to honestly thank the reviewer for the thoughtful and comprehensive review of our manuscript and the many useful comments and suggestions. By incorporating most of the suggested experiments, we believe we could significantly improve our manuscript, particularly with regard to which complement pathways that play a role in *T. b. gambiense* and how ISG65 interferes with them.*

As a side note –Transport of cargo to the flagellar pocket by hydrodynamic forces does play a role in trypanosomes and is well-described in the literature as an immune-evasion mechanism and should not be discounted. It has just not been described for ISG65. Several references have been provided in the manuscript as well as in the last rebuttal letter.

Major issues

Authors base their elimination of the CP as initiator on a single paper from 1986 by Devine et al. There are two problems with this. First, the lectin pathway was not discovered in 86. Second, in table 1 of the paper by Devine, there is a single data point for anti-C3c in the presence of EGTA, that would inhibit both CP and LP. In fact, there is quite a difference between NHS and EG-NHS. But, since the standard deviation is not provided, one cannot judge whether this is noise or reflects a true inhibition of a pathway. This paper may have been acceptable in 1986, but cannot form the basis for rejecting a contribution from the LP or CP in 2022.

In a modern paper, experiments with serum lacking selected complement components would be expected. Even a minor C3b deposition through LP/CP could be amplified in the AP. Authors must provide own evidence that CP/LP C3 convertase is not relevant for complement activation on trypanosomes or find additional evidence in the literature. It is not difficult to generate such data using commercially available serum depleted for components of CP, AP and terminal pathway (TP), eg C2, FB and C5. If such experiments are not performed or more convincing data from the literature is presented, they should adjust their model and state that the actual initiation mechanism remains to be settled.

*We fully agree with the reviewer, particularly on the fact that the LP was not discovered in 1986 (and therefore not investigated). A crucial fact that we were not aware of. We have performed binding experiments with pathway specific, fluorescently labelled antibodies and *T. b. gambiense* cells incubated in component-depleted human sera. In C2-depleted serum we could observe Bb and C3 surface binding (accumulating in the FP in *T. brucei*), while in fB-depleted serum no C4 and C3 binding could be detected. The CP/LP is therefore either not triggered or restricted upstream of C4b deposition.*

Furthermore no binding of C5b could be detected indicating that the terminal pathway is not initiated and thus the AP is restricted. In cases where a negative result was obtained erythrocyte binding was used as a positive control. For C4 binding, erythrocytes were incubated in C6-depleted serum to avoid cell lysis and allow imaging. For C5 binding, erythrocytes were incubated in C7-depleted serum (No signal was obtained in C6-depleted serum as C6 is needed for membrane docking of C5b). DOI: [10.1038/s41467-018-07653-5](https://doi.org/10.1038/s41467-018-07653-5)

Also, unlike in the AP50 test, where we can see a significant ISG65-mediated inhibition of the terminal pathway (i.e. reduced haemolysis) following activation of the AP, no such inhibition could be observed following the specific activation of the CP in the CH50 test.

The antibody used in the study by Devine for quantitation does not distinguish between C3, C3b, C3water, iC3b. Trypanosomes were incubated with 3 mg purified C3 in 500 ul. If this purified C3 contains or during the assay ticked over to 0.5% C3water (which is a conservative estimate), the concentration would be 30 ug/ml or 160 nM. The experimental data presented in Devine 1986 therefore cannot form the only basis for discussion on whether

native C3 binds to trypanosomes. Authors should perform their own C3 binding experiments using an antibody that is specific to C3 and does not react with C3water. Perhaps better, they could use native C3 in a competition experiment where labelled C3b binding is studied, fig 2b would be an excellent starting point. Native C3 could be used as competitor in flow and phago experiment. This would eliminate the influence of C3 tickover during the experiment and answer the crucial question of whether Tb bound ISG65 actually can bind native C3. If such experiments are not performed, the discussion must reflect that it is currently unknown whether native C3 can bind to ISG65 presenting trypanosomes.

We thank the reviewer for pointing this out! We agree to the criticism raised regarding Ab specificity and native C3 stability in the paper mentioned. We are also grateful for the suggestions on how to improve native C3 binding experiments, but after careful consideration we have decided to not perform these experiments for the following reasons:

- i) C3b and native C3 have quite different dissociation constants (C3 binds more weakly) and also binding kinetics which considerably complicates their use in competition assays.*
- ii) Accessibility of ISG65 is likely to be different for C3b and native C3. Devine et al. reported >3 times less native C3 bound to the cell surface than C3b. In reality it might have even been less, since, as pointed out by the reviewer here, their C3 prep likely contained a relatively high amount of C3water which likely behaved similarly to C3b and also bound to the surface.*

Differential affinity and accessibility of ISG65 for the two C3 species might therefore result in only a very small reduction in fluorescent signal when unlabelled C3 is used in competition with labelled C3b. To the authors' opinion this would be insufficient to make a reliable enough claim.

- iii) For direct Ab detection of native C3 one would need a conformation-specific antibody. The antibody that was suggested by the reviewer (communication via the editor) does however also detect C3water via binding to C3a/ANA, a motif in both C3 species. Despite thorough investigations, no tangible evidence for the existence of a conformation-specific, native C3 antibody could be found, neither in the literature nor on the websites of known vendors. According to the authors' opinion such an antibody would be very difficult to produce indeed since native C3 as an immunogen would also "tick" and thereby also raise antibodies against C3water. The same problem would present itself at the stage of selection against native C3.*
- iv) It was further suggested to use the anti-C3a antibody for detection of native C3 after removal of C3water via IEX chromatography and act as quickly as possible to avoid further pollution of the preparation via tick-over. While in principle this could alleviate the unintended binding of the antibody to ANA as mentioned here, such a setup is unfortunately logistically not feasible for us. The fluorescence microscopy shown in this study is carried out at another research institute without the actual possibility to perform the required C3 purification step on site. In addition to the sample preparation time during microscopy, the overall time span would be too long, allowing for further "tick over" and thus unreliability of the measurements.*

As suggested by the reviewer, we have therefore removed any claims from our manuscript stating that native C3 binds to ISG65 on the surface of trypanosomes. This is also reflected in our updated model of C3b binding on the cell surface (Figure 9). We do however not rule out that ISG65, which can be released from the surface during differentiation, may react with C3 fragments in solution.

The discussion regarding "Mechanisms of C3 inactivation at the plasma membrane" should be more clear on where covalently deposited C3b could be present. Also it should be made clear that C3b could be covalently (the membrane or anything embedded in the membrane) and non-covalently (through ISG65 binding) attached. Perhaps this can be included in an updated figure 6.

As suggested, we now clearly state that not all C3b is attached to ISG65 in the 'up' position and that it could also covalently attach to the plasma membrane (Results – 'Mechanism of C3b binding on the parasite surface' and Discussion). We also show this in an updated version of figure 9 (previously figure 6).

The data presented in the rebuttal letter on FH assisted FI degradation of C3b should be presented in the manuscript. Authors show in the rebuttal letter that ISG65 has no effect of fluid phase FH assisted FI degradation but they suggest the opposite in the text. This is contradictory. If they hypothesize that FH will disrupt the activity of FH on a trypanosome, they should provide experimental evidence for this. If in fact ISG65 does prevent FH cofactor activity it totally contradicts that ISG65 blocks progression of the alternative pathway as stated in the abstract. If FH activity is suppressed, the prediction is uncontrolled AP amplification. There are recent reviews on FH that authors may want to consult that provide excellent overviews of FH biology and FH focused complement evasion PMID: 33717083 PMID: 35469595.

We understand the reviewer's concern, but due to time constraints, the relative complexity of the necessary experiments (for a structural biology lab) and the fact that we are still not in possession of a complete knock-out,

we have decided to remove this aspect (i.e. inhibition of factor H co-factor activity) from the manuscript altogether and analyze it in due depth as part of a separate study. Omission of this aspects also helps to provide a better focus of the manuscript on the role of ISG65 in complement inhibition.

Abstract Line 10-11. "and its activation fragments and that it is sufficient to block progression of the alternative pathway in vitro"

This is not shown in the manuscript. In fact, authors fail to identify inhibition of the AP at any step. They show inhibition of the terminal pathway.

The sentence has been changed. It now reads "(...) that it takes over a role in selective inhibition of the alternative pathway C5 convertase and thus abrogation of the terminal pathway"

Line 59. C5 convertase could be assembled but not able to bind or cleave substrate. Devine et al 1986 is severely overinterpreted in the manuscript.

We agree with the reviewer on this, although this interpretation was initially made by Devine et al. in their publication and not by the authors of this study. We have removed this sentence from the introduction and re-phrased it in the results section:

"(...) inhibition via binding to C3b must take place at the stage of the AP C5 convertase, either by preventing its assembly or by diminishing the ability of the assembled convertase to cleave C5 substrate."

In fact, we have found that soluble ISG65 can bind to pre-assembled AP C5 convertase in vitro and reduce its activity which we detect as a decrease in erythrocyte lysis.

Line 206: Supp fig 7 should include 7pi6, the C3d:ISG65 crystal structure. Although released during revision, it should be properly presented, compared to the cryo-EM structures and discussed in this manuscript. A comparison between ISG65 bound C3/C3b and their unbound reference structures could also be placed in supp fig 7.

Structural alignments of ISG65:C3 – 7PI6, ISG65:C3b – 7PI6 as well as C3b (cryo) – C3b (crystal, 2I07), C3 (cryo) to C3 (crystal, 2A73) are now included in the new supplementary fig. 9. Similarities and differences between our complexes and 7PI6 are described and discussed in the figure legend of supp. fig. 9 and the manuscript discussion.

Line 220. The disulfide bridge residue numbering is not the numbering used in sup fig 8 and figure 4A. In pdb entry 7pi6, the disulfide Cys43-Cys200 is present that contradicts the mapping presented here. The electron density for 7pi6 confirms this disulfide. This should be addressed.

The reviewer is right here. We have therefore conducted a new disulphide mapping under acidic condition using nepenthesin-2 as an enzyme. As we detail in our response to reviewer 4, during our previous mapping at neutral pH using trypsin, disulphide scrambling might have occurred. This phenomenon has been described before (e.g. DOI: [10.1016/j.bbapap.2016.05.011](https://doi.org/10.1016/j.bbapap.2016.05.011)). It does however not occur at acidic pH. Using this new approach we could identify and confirm the disulphide bond presented in 7pi6. Our ISG65 hybrid model has been updated accordingly. This is the new pairing: C43-C200, C168-C181 and C240-C251 (also identified before).

Line 225-30. Panels showing the quality of 3D reconstruction for the regions mentioned here should be added to main figure 4 which currently only has 2 panels. This is just one example, authors need to provide multiple examples of map quality in the main figures. Currently there are only low magnification views of map in figure 3A. This is not proper for a manuscript that is so much based on two important cryo-EM structures. The sup fig 8 has many excellent panels with magnified views containing important residues in the structures with the map. These panels should go into the main figures instead.

As suggested, multiple panels showing close-up views of model and map are now shown in the new main figure 7. We do however believe that the former figure 4 (now figure 8, the schematic illustration) is better kept separate.

Figure 6. Unless the authors provide experimental evidence that ISG65 on *T. brucei* can bind native C3, they should omit a model detailing when native C3 can be bound. The figure is also misleading if molecules are drawn to scale. As ISG65 is shown here to the right, native C3 would penetrate significantly into the VSG layer whereas it is more reasonable that C3b can bind to the "active" ISG65. C3 binding should be eliminated from model unless supported by experimental evidence

*Since we do not provide experimental evidence for native C3 binding on the cell surface of *T. b. gambiense* we have omitted it from our model as suggested. Native C3 and C3water are still depicted in figure 9 (previously*

figure 6), but only as serum components in the fluid phase without any contact to the surface. This is also made clear in the figure legend and the discussion.

"(...) C3(H₂O) and native C3 are depicted in solution as it is currently unknown whether they bind to the trypanosome surface." (figure legend)

Line 440. The fact that there is no structure of a C5 convertase should not prevent authors from testing how ISG65 interferes with C5 convertase. Authors could separate AP or CP C3 deposition from MAC assembly by using e.g. C5 deficient serum in the first step, wash, and then add the "missing" factor together with ISG65 to study the effect of ISG65 on the CP and AP C5 convertase and MAC assembly. These are not very demanding experiments

We thank the reviewer for this very useful suggestion. We have performed the experiment and could show that soluble ISG65 can bind to pre-assembled AP C5 convertase (CP/LP C5 convertase was not further tested as no inhibition was observed in the CH50 assay) and reduce its activity. We would however like to point out that simply adding the missing factor C5 does not work, as the washing step is removing all other complement factors necessary in the terminal pathway. C5 has to be added back (together with ISG65) as part of C3-deficient serum which provides the factors need downstream.

Comments to the revised manuscript

Minor points

Line 31. Spell out abbreviations for Tb proteins

Abbreviations are now spelled out.

Line 45. MAC formation occurs in the terminal pathway, not the AP. It is possible to have MAC formation e.g in FB depleted serum in vitro. Authors must include the CP/LP C5 convertase in their analysis of mechanism.

The mistake has been corrected. We now mention and show MAC formation as part of the terminal pathway. We now also discuss the role of the CP/LP C5 convertase in the introduction.

Line 116. Add legend letters

A legend letter has been added.

Line 136. Add ref to sup fig 3 here

A reference to supplementary figure 3 has been added.

Line 146. Add reference Fig 2D

This sentence has been removed. New referencing has been introduced where appropriate.

Line 149. Add ref to figure 1

A reference to figure 1 has been added.

Line 160-2. "physiologically relevant" instead ?. Mention the C3 and C3b concentrations in ref 23 for comparison with K_d here.

We fully agree with the suggestion. "Physiologically" has been changed to "physiologically relevant". Concentrations for C3 and C3b have been added.

Line 189: TED is not in contact with ANA in native C3. It is the association with CUB and MG8 that fixes TED in native C3. Authors should clarify.

We agree that this formulation is misleading. TED does interact with MG2, MG8 and CUB. It has been changed to "(...) with TED wedged between MG2, MG8 and CUB(...)"

Line 199: Quantitate the resemblance between C3b structures by rmsd number

The resemblance between C3/C3b and their ISG65-bound counterparts is now quantitated by RMSD. This is mentioned in Supp. Fig. 9 as well as the main text in the results section.

Line 241. Why is Fig 4a relevant here? 5a appear to be relevant

This is correct. The figure reference has been changed.

Line 244-45. A panel presenting the quality of the map underlying the identification of these hydrogen bonds must be displayed

Panels showing the map quality for interacting residues have now been moved from the supplemental part of the manuscript to the new main figure 7. We have however to point out once again, that, in cases where due to the low local resolution, side chain density was ambiguous, we validated the importance of these residues for the overall binding affinity by alanine mutagenesis and SPR analysis.

Line 265. Show the density of Trp211 in main figure

The electron density for Trp211 is now shown in the main figure 7 alongside electron densities of other key residues.

Line 330-8. The discussion here is an over-interpretation. SAXS data for C3b has shown that TED can release from the MG1 domain but not to the extent suggested here. C3b is also not smaller than antibodies, it has a higher molecular weight and antibodies are more flexible. So the argument is invalid. Without further evidence, the distinction between C3 and C3b with respect to VSG penetrance is not justified. Also, the conformation of ISG65 could be hugely influenced by the VSG coat as compared to the SAXS experiment conducted here. That being said, the SAXS models are likely to present two possible conformations of the C-terminal regions in solution well.

We did not say that C3b is smaller than an antibody. This is a misunderstanding. What we said was that CUB/TED as part of C3b is smaller than an antibody. Thus CUB/TED of C3b could penetrate the VSG layer similar to the Fab arm of an antibody. This was shown recently: doi:10.1016/j.celrep.2021.109923.

However, since we do not claim anymore that native C3 binds to the parasite surface, the paragraph discussing the structural differences of both C3 conformations with regard to VSG penetrance has been removed.

Line 452. Ref 56 is out of context

The corresponding sentence is not present anymore in the new version of the manuscript.

Line 723. What tool in phenix is used for this docking? Perhaps coot or chimera instead? Authors broadly mention "phenix" as reference for multiple steps. Phenix is a collection of programs written by many different contributors. It would be more informative to specify the actual program in, e.g phenix.real_space_refine. To compare, REFMAC is a CCP4 program in the same manner as phenix.real_space_refine is a phenix program. Also molprobity used for validation was perhaps the phenix version?

We used the programs dock_in_map and real_space_refine implemented in Phenix. This information has now been added to the respective passage in the text. For validation the molprobity website (<http://molprobity.biochem.duke.edu>) was used.

Figure 1. CP/LP C3 and C5 convertase are not depicted. They cannot be excluded from the mechanistic analysis given the weak basis presented by the authors for excluding the CP/LP. FP binds to C3bB, not only to C3bBb. C3dg does not bind CR4, it binds the C3c moiety within iC3b, PMID: 28292891.

We have included the classical and lectin pathway in a new version of figure 1. We also show fP binding to C3bB and not only C3bBb. Opsonization, host cell binding and negative regulation of the cascade have been omitted from the figure for clarity since the corresponding, relevant data has been removed from the manuscript. The figure now focuses on CP,LP, AP and TP and the complement complexes mentioned in the manuscript.

Figure 3. Open book views of the C3-ISG65 and C3b-ISG65 views could be added here.

Open books views of both complexes have been added in the new figure 6.

Figure 4. Perfect place for adding magnified views of model with map in important regions.

Close-up views of the suggested regions of model and map are now presented in the new figure 7.

Figure 5. Panel b could be much smaller and still provide the message. Show the map quality for the C3a interaction.

Panel b and c have been removed from the manuscript.

We have added a panel showing the map quality for the ANA interaction in figure 7. As we point out in the text, side chain density for interacting residues is poor in this region due to intrinsic flexibility of both the residues (lysins and glutamates) and the loops they are located in. Because of this, SPR analysis was conducted using the isolated domains and mutational analysis of the relevant residues in ISG65 (as suggested by the reviewer before).

Supp fig 12. Panel B, unit on horizontal axis is nm not Å

We apologize for the mistake. The unit is indeed nm not Å. This has been corrected (now supp. fig. 14).

Reviewer #4 (Remarks to the Author):

Overall, the authors have satisfactorily addressed most of my previous comments. There are a few minor edits still needed and a few major concerns on the disulfide crosslinking experiment.

Supplemental method text changes

The revised methods are much improved and satisfactorily detailed. Specific final edit suggestions below:

HDX section:

- Please include manufacturer info for nepenthesin-2 column.
- Acidity of deuterated solutions should be reported as pD, not pH.

The manufacturer of the nepenthesin-2 column has been included (Affipro). The acidity of the deuterated solution is now reported as pD.

In-gel section:

- AGC and MIT values are reported as "custom", which means the user enters their own percent value; text should include the percent value entered.
- Fragmentation method is reported as HCD; please also include collision energy setting for completeness.
- The authors state that the mass tolerances for Sequest were 10 ppm and 0.6 Da for MS1 and MS2 scans, respectively, but in MS Amanda these were set to 5 ppm and 0.06 Da. Because the MS2 scans were acquired in

the Orbitrap, the 0.6 Da mass tolerance in Sequest is pretty broad, and a 0.06 Da setting would make more sense for both search engines. Can the authors confirm whether the 0.6 Da setting is correct vs. a typo? No change needed if these values are correct.

The missing information has been added.

The 0.6 Da setting is correct.

"AGC target value was set as custom with normalized AGC target 250 %. Maximum injection time was set to 50 ms."

"(...)ions were fragmented by HCD collision energy set to 30% with dynamic exclusion set to 60s."

Supplemental figure changes

SF1-B: Looks good. Legend text reads that "complement C3 was identified with the highest PSM score" – please amend to "highest number of PSMs" for accuracy.

The legend text has been changed as suggested.

SF1-C: Nice addition. Minor suggestion: a color legend in the figure itself, in addition to the legend text, would be helpful, though not strictly necessary.

We agree with the reviewer's suggestion. A colour legend has been added to the figure panel.

Intact mass experiment changes

My apologies to the authors – I was inappropriately requesting a standard of precision common to peptide experiments from an intact mass experiment. I appreciate that intact mass is much more complicated than peptide mass, given the variation from potential proteoforms and the lack of a detectable monoisotopic peak in most intact measurements. I thank the authors for the changes implemented and the time invested in their detailed response. The figure is much clearer and the added blot evidence is helpful.

Disulfide mapping experiment

The authors have helpfully clarified that they are looking for native disulfide linkages as part of a general characterization but not for use in refining the cryo-EM model. The figure is still difficult to parse, and the numbering in the main text (which specifies bridges between C12-C137, C149-C168, and C209-C220) does not agree with the peptide map provided in SF8-A,B. In the peptide map, the sequence picks up at residue 32, which would make detection of a bridge with C12 impossible, and of the remaining specified cysteines, only residue 168 is actually a Cys in the map. The peptide sequences listed in SF8-C,D,E suggest the authors mean to specify bridges between C43-C168, C181-C200, and an internal link of C240-C251. If these are accurate, the main text needs to be revised. If these are not correct, the figure needs to be revised.

We apologize for this error. The peptide map is correct and the numbering as presented in the main text is incorrect.

I thank the authors for providing MS2 spectra of the three disulfide linkages. The first and third peptides (EGAAALCK-VLNWYCITK and NCNIGQ....) are credibly presented. I am skeptical of the second peptide (HNATPN....) as presented, however.

While this manuscript was under review the structure of a homologous protein of ISG65 was published. In the crystal structure of the homologue the disulphide bridge Cys43-Cys200 was identified which contradicts the mapping as presented here. No other disulphide bridges were visible in the crystal structure of the homologue. Most likely the differently linked disulphide peptides identified in our sample were formed due to disulphide bond scrambling although cystamin was present during the sample processing. To eliminate disulphide bond scrambling, ISG65 was online digested by Nepenthesin-2 under acidic conditions. The nonspecific peptides were measured using a timsToF mass spectrometer operated in PASEF mode. Here, new disulphide bridges Cys43-Cys200 and Cys168-Cys181 were found together with the previously identified internal bridge Cys240-Cys251. The corresponding MS/MS spectra are now shown in supplementary figure 10. Interestingly, no N-linked glycoform of disulphide linked peptide was found in this data. This might be caused by different cleavage specificity of the Nepenthesin-2 enzyme.

Disulphide linked peptides VLNWYC(168)ITK – HNATPNC(181)DGIQFR and EGAAALC(43)K – SAIDC(200)SSTSYEENYDWSANALQVALNSWEDVKPK were detected in the trypsin treated sample too. For the former also the N-linked glycan was detected. As the insect cell-derived glycosylation is not relevant for our analysis and the disulphide-linked peptide identified under acidic conditions are generally more credible due to a lack of scrambling, these are now presented in the manuscript. We also decided to omit the peptide coverage map as it does not add essential information and

unnecessarily inflates an already large figure. Information about the glycosylation in the trypsin-treated sample is shown below for the interested reviewer.

Trypsin VLNWYC(168)ITK – HNATPNC(181)DGIQFR

Trypsin VLNWYC(168)ITK – HNATPNC(181)DGIQFR **Non-glyco**

Trypsin

VLNWYC(168)ITK – HNATPNC(181)DGIQFR

Glyco

- The methods for this section (main text lines 625-627) are well-detailed for the LC portion of the experiment but not for the MS acquisition or search. The authors should expand this text to include parameters of both the instrument acquisition (either reference a publication detailing "the PASEF method for standard proteomics" or listing specific parameters here) and the data search, and specifically what sequences were searched and what variable modifications were included (presumably fucosylated paucimannosylation, but what others?).

We thank the reviewer for the comment, we have added a detailed description of MS parameters and search parameters. The internally linked disulphide linked peptides and glycopeptides were identified by manual curation of the raw data.

- The annotated sequence in Supp Fig 8d shows the modification present on N176, but with only 4 fragment ions from the other peptide in the crosslink, this mod cannot be localized to that specific Asn residue based on the spectral data. The modification would need to be presented as attached agnostically to the cross-linked pair. Alternatively, N-glycosylation often follows sequence motif patterns in human cells – did the authors assign the modification based on NATP being a known sequence motif in this species? If so, that could justify the localization if accompanied by explanatory text and a reference establishing the motif.

We agree with the reviewer that based on the 4 fragments, the modification cannot be localized. We have assigned the glycan modification based on the NXS/T motif on which N-glycosylation occurs in insect cells (which were used for protein expression). As no N-linked glycans are present in the new spectra that we included in the manuscript no further explanation was added.

- The modification is shown on the peptide sequence with the green dot moieties being branched, but on the spectra, shown as linear. The presentation should be consistent between the two.

We have corrected this in the figure shown above.

- The response to the authors includes an EIC for the glyco-crosslinked form and the non-glyco-crosslinked form. The glyco form does appear as the more abundant form, but have the authors investigated the MS2 data for the non-glyco form? The unmodified crosslinked peptide may provide more backbone fragment ions and if so, I recommend including it side-by-side with the glyco form to raise confidence in this peptide identification.

We have investigated the non-glycoform of the old disulphide linked peptide. The ion intensity of the parent ion was very low and did not pass the criteria for CID fragmentation in the timsToF mass spectrometer.

The presence of paucimannose was confirmed by detection of another disulphide linked peptide VLNWYC(168)ITK – HNATPNC(181)DGIQFR, where both glyco- and nonglyco-forms were observed (shown above).

HDX experiment

I thank the authors for including the additional detail requested and for the thoughtful response to the differences in timepoints for strongest protection. I have no further comments on this section.

Other comments

The authors have sufficiently addressed previous concerns/suggestions.

REVIEWERS' COMMENTS

Reviewer #1 (Remarks to the Author):

The manuscript has been vastly improved. I congratulate the authors with a very comprehensive set of interdisciplinary data that describes an important aspect of *T. brucei* biology and opens for many new experiments. Consideration of the minor points below may improve the manuscript further.

Main text

Line 49. Change to “the serine protease C2a” to make more clear to the general reader

Line 64. The role of the second C3b molecule is not clarified, possibly C3b only primes C5 for cleavage rather than being a part of the C5 convertase. Perhaps authors wants to add this.

Line 153. Start the sentence with “Modelling suggests..” since sup fig 5 does not contain experimental data. Related to that, I suggest to merge sup fig 5 and suP fig 15, thereby it is also possible to refer to the data in sup fig 15 under Results rather than only bringing up this information in the Discussion

Line 188. The value referenced for C3b appears high. First of all, most of it is likely to be iC3b. Authors may also want to consult PMID: 19196712

Line 221. Here and in many other places, the number of atoms contributing to the RMSD should be specified. The RMSD only is meaningful if number of atoms is given for comparison, small structures will tend to have lower RMSD than larger structures.

Line 240. Supp fig 10 could be specified to sup fig 10d. Similar problem is present in multiple other places, where mentioning the exact panel rather than just the figure number could be helpful for the reader

Line 258. There is only one structure of native C3 in ref 28, so either give more references or use “structure”

Line 390. Explain what is meant with “natural receptor”

Figure 4, line 930. It states that incubation was C3 depleted followed by C5 depleted. In text it is the other way around, which must be correct.

Figure 5. It appears to be C3MA and not C3(H₂O) that was used for this experiment and the MS analysis conducted in the supporting material. C3MA is a widely used and fully acceptable mimic of C3(H₂O), but the correct name should be displayed.

Figure 7. Add labels C3 and C3b to the upper and lower panel, respectively, and add label ISG65 to both.

Line 992. It is confusing to use both TED and C3d. Explain once how C3d relates to C3 and C3b and then use the term TED in the context of C3 and C3b. C3d is a proteolytic fragment of iC3b, it is not the same as TED

Fig 8B. The location of the membrane relative to helix 6 is misleading if compared to the sequence. Also the width of the membrane could reflect what is expected to be covered on helix 6.

Line 1051. C3b in the blood stream is mentioned. But this could perhaps also be C3b bound covalently to the VSG layer. There is no need to assume that covalent bound C3b is only present inside the VSG as sketched here.

Supporting material

P13. See my comment above on not using C3(H₂O) if C3MA was actually used

P24. electron density is the word for a map obtained by crystallography. For EM, it is Coulomb potential

P25. use "." instead of "," in table of distances

P47, panel B. The pair distance distribution shown here was most likely calculated without a model, only from the data. It is therefore not relevant to mention ensemble here. The descent at the high distance is also very steep, possibly the D_{max} was underestimated

REVIEWERS' COMMENTS

Reviewer #1 (Remarks to the Author):

The manuscript has been vastly improved. I congratulate the authors with a very comprehensive set of interdisciplinary data that describes an important aspect of *T. brucei* biology and opens for many new experiments. Consideration of the minor points below may improve the manuscript further.

Main text

Line 49. Change to “the serine protease C2a” to make more clear to the general reader

The suggested description has been added.

Line 64. The role of the second C3b molecule is not clarified, possibly C3b only primes C5 for cleavage rather than being a part of the C5 convertase. Perhaps authors wants to add this.

Indeed. A sentence and a reference mentioning this alternative model have been added.

“The role of the second C3b molecule has however not been fully clarified. Recently, it was also suggested that surface-bound C3b only primes C5 for cleavage by C3bBb rather than being part of a trimolecular complex²⁰.”

Line 153. Start the sentence with “Modelling suggests..” since sup fig 5 does not contain experimental data. Related to that, I suggest to merge sup fig 5 and suP fig 15, thereby it is also possible to refer to the data in sup fig 15 under Results rather than only bringing up this information in the Discussion

We fully agree with the reviewer’s suggestion. The two supplementary figures 5 and 15 have been merged and the text has been changed to:

“Modelling suggests that ISG65 does not interfere with assembly or activity of the AP C3 convertase (Supplementary Fig. 5a). Additionally, AP C3 pro-convertase formation in presence of ISG65 was shown by SPR (Supplementary Fig. 5b).”

Line 188. The value referenced for C3b appears high. First of all, most of it is likely to be iC3b. Authors may also want to consult PMID: 19196712

We could not find any information on more “realistic” C3b concentrations in the referenced article. In fact such values are difficult to find in the literature. In absence of a better number we would like to keep the current range as an approximation amended by the following statement:

“Notably, the amount of C3b depends on the complement activation status. Together with a short half-life of C3b, actual concentrations may be lower than reported values.”

Line 221. Here and in many other places, the number of atoms contributing to the RMSD should be specified. The RMSD only is meaningful if number of atoms is given for comparison, small structures will tend to have lower RMSD than larger structures.

The number of atoms used in each alignment has been appended to main text and Supplementary Figure legends as needed.

Line 240. Supp fig 10 could be specified to sup fig 10d. Similar problem is present in multiple other places, where mentioning the exact panel rather than just the figure number could be helpful for the reader

Where possible, the references to specific panels in the figures have been added.

Line 258. There is only one structure of native C3 in ref 28, so either give more references or use “structure”

This has been corrected. Only one structure is referenced and “structures” has been changed to “structure”.

Line 390. Explain what is meant with “natural receptor”

Only T. b. gambiense, from which the ISG65 used in this study is derived, is able to infect humans (this is mentioned in the introduction). Therefore only ISG65 from this parasite but not from T. b. brucei will ever come in contact with human C3 and its fragments in nature. This has been clarified in the text:

“Tbg ISG65, the receptor that encounters human C3 under natural infection conditions, showed a 2 magnitudes (...)”

Figure 4, line 930. It states that incubation was C3 depleted followed by C5 depleted. In text it is the other way around, which must be correct.

The mistake in the figure legend has been corrected.

Figure 5. It appears to be C3MA and not C3(H₂O) that was used for this experiment and the MS analysis conducted in the supporting material. C3MA is a widely used and fully acceptable mimic of C3(H₂O), but the correct name should be displayed.

This has been corrected.

Figure 7. Add labels C3 and C3b to the upper and lower panel, respectively, and add label ISG65 to both.

Labels for C3, C3b and ISG65 have been added to Figure 7. Additionally, a previously missing label for the ANA domain has been added as well.

Line 992. It is confusing to use both TED and C3d. Explain once how C3d relates to C3 and C3b and then use the term TED in the context of C3 and C3b. C3d is a proteolytic fragment of iC3b, it is not the same as TED

This has been corrected.

Fig 8B. The location of the membrane relative to helix 6 is misleading if compared to the sequence. Also the width of the membrane could reflect what is expected to be covered on helix 6.

We agree with the reviewer that the previous illustration may have been misleading. We have changed both the placement and the width of the illustrated cell membrane to more closely resemble the architecture outlined in panel a.

However, we would like to point out that this Figure serves only as an illustration of the overall architecture of the protein, as clearly stated in the Figure legend, and is thereby not technically to scale, especially with respect to the disordered termini as well as the disordered loops/linkers. We would like to point out that ISG65 in context of the membrane is illustrated to scale in Fig. 9.

Line 1051. C3b in the blood stream is mentioned. But this could perhaps also be C3b bound covalently to the VSG layer. There is no need to assume that covalent bound C3b is only present inside the VSG as sketched here.

While covalent C3b binding to membranes is well documented (as this where the lytic pore is being formed) there is to our knowledge no evidence, at least not in vivo, that covalent bonds can also be formed with membrane-associated proteins. As this is to our opinion quite speculative in this context, we do not want to depict it in the same way as we depict covalent membrane

binding (with a red line to TED). On the other hand it can also not be ruled out. We have therefore decided to add an explanatory sentence in the figure legend (as well as the corresponding section in results) stating both options.

Supporting material

P13. See my comment above on not using C3(H2O) if C3MA was actually used

The figure has been changed accordingly.

P24. electron density is the word for a map obtained by crystallography. For EM, it is Coulomb potential

We apologize for the inaccuracy of the statement. The figure legend as been updated an now states:

"ISG65 hybrid model and locally sharpened cryo-EM density map around the head domain. The boundaries of the density cloud and disulphide positions were used as constraints in model selection."

P25. use "." instead of "," in table of distances

We apologise for the oversight. This has been corrected.

P47, panel B. The pair distance distribution shown here was most likely calculated without a model, only from the data. It is therefore not relevant to mention ensemble here. The descent at the high distance is also very steep, possibly the Dmax was underestimated

The mention of the ensemble has been removed. The Dmax was indeed slightly underestimated. This has been corrected.